# Domain walls from SPT-sewing

Yabo Li,[1, *] Zijian Song,[2, †] Aleksander Kubica,[3] and Isaac H. Kim[4]

[1]*Center for Quantum Phenomena, Department of Physics,*
*New York University, 726 Broadway, New York, New York 10003, USA*
[2]*Department of Physics and Astronomy, University of California, Davis, California 95656, USA*
[3]*Yale Quantum Institute & Department of Applied Physics,*
*Yale University, New Haven, CT 06520, USA*
[4]*Department of Computer Science, University of California, Davis, California 95616, USA*

We introduce a systematic method for constructing gapped domain walls of topologically ordered systems by gauging a lower-dimensional symmetry-protected topological (SPT) order. Based on our construction, we propose a correspondence between 1d SPT phases with a non-invertible $G \times \mathrm{Rep}(G) \times G$ symmetry and invertible domain walls in the quantum double associated with the group $G$. We prove this correspondence when $G$ is Abelian and provide evidence for the general case by studying the quantum double model for $G = S_3$. We also use our method to construct *anchoring domain walls*, which are novel exotic domain walls in the 3d toric code that transform point-like excitations to semi-loop-like excitations anchored on these domain walls.

## CONTENTS

---

* liyb.poiuy@gmail.com
† zjsong@ucdavis.edu

## I. INTRODUCTION

Most of our understanding of topological order stems from various exactly solvable models. The quantum double [1] and string-net models [2] provided concrete examples to study universal properties of topologically ordered phases; their extensions to three dimensions are also known [3]. It is believed that they constitute all gapped phases of matter with a gappable boundary [4, 5]. These models can also host a variety of lower-dimensional defects, which can make their physics richer [6–21]. The codimension-1 defects are called as *domain walls*, which are the main subject of our work.

Domain walls are often understood by folding the system along them and, subsequently, turning the domain walls into gapped boundaries [9]. In two dimensions, if the underlying anyon model is Abelian, the gapped boundary can be classified in terms of the set of anyons that can be condensed on the boundary, known as the Lagrangian subgroup [16]. More generally, Refs. [9, 17, 18, 22–24] proposed to use higher category theory to classify the boundaries. The corresponding mathematical object is a generalization of the Lagrangian subgroup, called the Lagrangian algebra. Beyond studying the condensation of Abelian anyons (0-dimensional topological defects), this framework examines the condensation of non-Abelian anyons and higher-dimensional topological excitations [24]. This approach has been employed in the study of gapped boundaries and domain walls of 2d non-Abelian quantum doubles [21] as well as 3d topological orders [25, 26].

However, not everything about the domain walls is well-understood, even in two spatial dimensions. For instance, Ref. [27] reported a set of superselection sectors and fusion rules at the domain wall whose categorical description is not understood at the moment; domain walls in higher dimensions are even less understood. In general, in order to study the physical properties of the system it is desirable to have a solvable model whose ground state can be described exactly.

The unifying approach to the problem of understanding domain walls that underpins our work is gauging. Recent studies have shown that gauging gives rise to a method of preparing a variety of topological phases [28–37]; gauging has also been explored in the context of quantum error-correcting codes [38–43]. The gauging procedure can be implemented using constant-depth adaptive quantum circuits and has been experimentally demonstrated [44–47]. Very recently, such an approach was extended to the more general setting of solvable anyon models [48].

Our work employs a similar approach, but by gauging lower-dimensional defects. For instance, we extensively study the effects of gauging a lower-dimensional symmetry-protected topological (SPT) order embedded in a trivial bulk, symmetric under the same symmetry group. We remark that there are prior works creating defects in a similar way [49]. However, the way in which we gauge the lower-dimensional SPT is different and leads to the domain walls that, to the best of our knowledge, cannot be obtained by the method from Ref. [49]. In order to differentiate the two approaches, we refer to the method from Ref. [49] as gauged-SPT domain wall (or more generally, defect). Our method is referred to as *SPT-sewing*. The origin of this name will become clear once we explain our method.

Although the bulk of our work focuses on gauging the standard SPT (based on a group symmetry), we also consider nontrivial SPT phases with a generalized notion of symmetry [50, 51]. These generalized symmetries are known as non-invertible symmetries [52–54], and lattice models for various non-invertible SPT phases have been studied [55–60]. We also discuss such models because there are domain walls that can only be created by gauging an SPT with a non-invertible symmetry.

The gauging approach allows us to obtain a variety of domain walls. In fact, in two spatial dimensions, we believe our approach is general enough to construct *all* gapped domain walls (insofar as the underlying topological order in the bulk can be obtained by gauging some SPT). The main evidence for this conjecture comes from Abelian anyon models, for which we prove that all gapped domain walls can be obtained via our method. The evidence for the non-Abelian case is more scarce; nonetheless, we provide some nontrivial examples, such as the quantum double model for the symmetric group $S_3$.

In three spatial dimensions, we obtain conceptually novel domain walls, which we call *anchoring domain walls*. Anchoring domain walls transform point-like excitations into semi-loop-like excitations anchored on them, and vice versa. We find that there are two types of such domain walls, which exhibit different physics. Namely, one type hosts an Abelian anyon model at the domain wall whereas the other one hosts a non-Abelian anyon model. To the best of our knowledge, such domain walls have not been studied before. How they fit with the existing categorical classification of lower-dimension defects [24] remains unclear. We simply discuss their constructions and pertinent properties, leaving the classification question for future work.

The rest of this article is structured as follows. In Section II, we describe our method with a simple example of the 2d toric code [1]. In Section III, we introduce the convention used throughout our work and review how gauging works. In Section IV, we construct all possible gapped domain walls for Abelian quantum doubles in two dimensions. In Section V, we apply our method to the non-Abelian quantum double model for $S_3$. In Section VI, we discuss various exotic domain walls in the 3d toric code and some simplified models are presented in Section VII. We end with a discussion in Section VIII.

## II. EXAMPLE: 2D TORIC CODE

In this section, we explain our method with a simple example of the 2d toric code. First, we briefly review this model and the gauging map we use. We then explain how gauging can be used to construct domain walls of the 2d toric code.

### A. Toric code and the gauging map

The toric code [1] is defined on a square lattice, wherein each edge hosts one qubit. The Hamiltonian contains commuting four-body interaction terms, often referred to as the vertex ($A_v$) and plaquette ($B_p$) terms

$$H = -\sum_{v \in V} A_v - \sum_{p \in F} B_p = -\sum_{v \in V} \prod_{v \in e} X_e - \sum_{p \in F} \prod_{e \in p} Z_e, \tag{1}$$

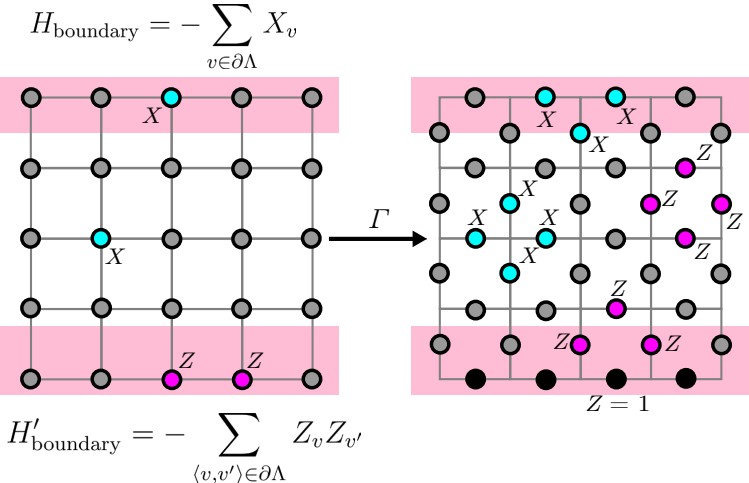

Figure 1. Schematic description of the gauging map. Local symmetric operators are mapped to local operators on the right lattice. For instance, $X_v$ is mapped to $\prod_{v\in e} X_e$, and $Z_v Z_{v'}$ on adjacent vertices is mapped to $Z_{\langle v,v'\rangle}$ on the edge. As a result, two different boundary Hamiltonians become the smooth and rough boundary.

where $V$ and $F$ are the sets of vertices and plaquettes, respectively. The ground states are thus the simultaneous $+1$-eigenstates of $A_s$ and $B_p$. If a quantum state $|\Psi\rangle$ is a $-1$-eigenstate of a certain vertex (plaquette) term, we say that $|\Psi\rangle$ has a point-like $e$ ($m$) excitation on that vertex (plaquette). A pair of such excitations can be created by applying a string of Pauli $Z$ or $X$ operators on the ground state.

A ground state of the toric code can be obtained via a *gauging map* applied to the trivial fixed-point state of the $\mathbb{Z}_2$ symmetry, i.e., $|+\rangle^{\otimes V}$ [61]. The gauging map is a bijective map between states with global symmetry $\mathbb{Z}_2$ and states with a gauge symmetry $\mathbb{Z}_2$ on the same lattice, but with the Hilbert space associated with the vertices for the former and with the edges for the latter. It is convenient to describe the action of the gauging map on local operators. The map converts a Pauli $X$ operator on a vertex $v$ to $\prod_{v\in e} X_e$. It also converts a Pauli operator of the form $Z_v Z_{v'}$ to $Z_e$, where $e$ is an edge connecting $v$ and $v'$. Upon gauging, the $\mathbb{Z}_2$ global symmetry becomes the $\mathbb{Z}_2$ gauge symmetry. This implies that the state $|+\rangle^{\otimes V}$ gets mapped to a ground state of the toric code; see Fig. 1.

For the lattice with open boundary conditions, the gauging map yields gapped boundaries of the toric code. Let $\Lambda$ be a square lattice with open boundary conditions and denote its boundary as $\partial\Lambda \subset \Lambda$. Different boundaries of the toric code can be obtained by choosing different boundary Hamiltonians with respect to the $\mathbb{Z}_2$ global symmetry. There are only two allowed possibilities, the symmetry and spontaneously symmetry breaking phases. After gauging, they become different toric code boundaries.

The smooth boundary [12] can be obtained from the product state $|+\rangle^{\otimes \Lambda}$, which can be thought of as the ground state of a $\mathbb{Z}_2$ symmetric Hamiltonian

$$H_1 = H_{\text{bulk}} + H_{\text{boundary}} = -\sum_{v\in\Lambda\setminus\partial\Lambda} X_v - \sum_{v\in\partial\Lambda} X_v. \tag{2}$$

Upon gauging, $H_{\text{boundary}}$ becomes a sum of stabilizers that define the smooth boundary; see the top right of Fig. 1.

The rough boundary [12] can be obtained by changing the boundary term to a $\mathbb{Z}_2$-symmetry breaking term

$$H'_{\text{boundary}} = -\sum_{\langle v,v'\rangle\in\partial\Lambda} Z_v Z_{v'}. \tag{3}$$

After gauging, we obtain the rough boundary; see the bottom right of Fig. 1.

On the smooth (rough) boundary, a single $m$ ($e$) particle can be annihilated. When that happens, we say that $m$ ($e$) is condensed on that boundary [7, 62]. These two boundaries are the only possible boundaries for the 2d toric code [8, 9, 13–15, 17–19].

| Subgroup $K$ | 2-cocycle | Domain wall |
|---|---|---|
| $\mathbb{Z}_2^{diag}$ | trivial | $S_1$ |
| $\mathbb{Z}_2^{(1)} \times \mathbb{Z}_2^{(2)}$ | nontrivial | $S_\psi$ |
| $\mathbb{Z}_2^{(1)} \times \mathbb{Z}_2^{(2)}$ | trivial | $S_m$ |
| $\mathbb{Z}_2^{(1)}$ | trivial | $S_{me}$ |
| $\mathbb{Z}_2^{(2)}$ | trivial | $S_{em}$ |
| $1$ | trivial | $S_e$ |

Table I. Correspondence between the 1d gapped phases under $\mathbb{Z}_2^{(1)} \times \mathbb{Z}_2^{(2)}$ and the domain walls in the $\mathbb{Z}_2$ toric code. $K$ is the unbroken subgroup, and the different 2-cocycles correspond to the different SPT phases under the unbroken $K$ symmetry.

## B.  Examples of SPT-sewing

Gapped phases of matter can host a *gapped domain wall*, which is a codimension-1 defect. For the toric code, there are six types of domain walls. Some of these domain walls are *invertible*, meaning that all the anyons can pass through. There are two invertible domain walls, the trivial one and the one exchanging the $e$ and $m$ particles. We refer to them as $S_1$ and $S_\psi$, respectively. The other domain walls are *non-invertible* and are labeled by the anyons that can condense on them, i.e., $S_e, S_m, S_{em}$, and $S_{me}$.

The non-invertible domain walls can be obtained via the method in Section II A. We thus focus on the construction of invertible domain walls, in particular the $S_\psi$ domain wall. There are many different ways of constructing $S_\psi$, e.g., via a lattice defect [6, 9]. Recently, Ref. [49] showed that $S_\psi$ can be constructed from gauging a 1d Kitaev Majorana fermion chain. Ref. [63] showed from topological field theory that $S_\psi$ can be constructed from higher-gauging on a codimension-1 surface in the spacetime. Ref. [64] showed that the domain walls can be obtained from a toric code ground state by applying a linear-depth unitary sequential circuit.

Here we take a different approach to constructing the $S_\psi$ domain wall by considering two disconnected lattices with close-by boundaries; see Eqs. (4)-(5) for illustration. Note that there is a $\mathbb{Z}_2$ global symmetry on each side. We can then obtain an invertible domain wall by decorating a nontrivial $\mathbb{Z}_2 \times \mathbb{Z}_2$ SPT state on the close-by boundaries and gauging. (For a trivial SPT, gauging would yield $S_m$.) We can view this procedure as sewing together two boundaries using a 1d SPT state, hence the name *SPT-sewing*.

An emblematic example of a $\mathbb{Z}_2 \times \mathbb{Z}_2$ SPT is the one-dimensional cluster state [65], where each $\mathbb{Z}_2$ symmetry acts separately on the set of even and odd sites. By placing the even and odd sites on the left and right close-by boundaries, we obtain an SPT compatible with the global symmetry we imposed. After applying the gauging map to this setup, the stabilizers of the SPT are transformed as follows

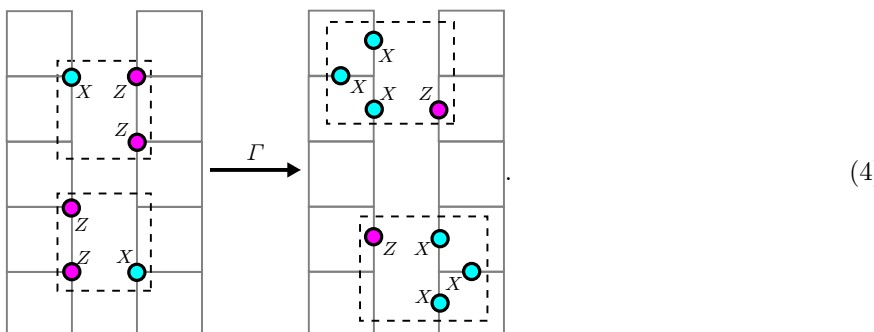

$$\tag{4}$$

This domain wall is invertible because the $e$ and $m$ particles from one side can pass through. Upon passing, $e$ is exchanged with $m$, and vice versa. Therefore, we realized the $S_\psi$ domain wall by gauging a 1d SPT under the $\mathbb{Z}_2 \times \mathbb{Z}_2$ symmetry.

Because there is only one nontrivial 1d SPT under the $\mathbb{Z}_2 \times \mathbb{Z}_2$ symmetry, one cannot hope to sew a $\mathbb{Z}_2 \times \mathbb{Z}_2$ SPT to obtain the transparent domain. However, we can instead consider a lattice defect line with three global $\mathbb{Z}_2$ symmetries. The following is a 1d $\mathbb{Z}_2 \times \mathbb{Z}_2 \times \mathbb{Z}_2$ SPT, which after gauging returns the trivial

domain wall

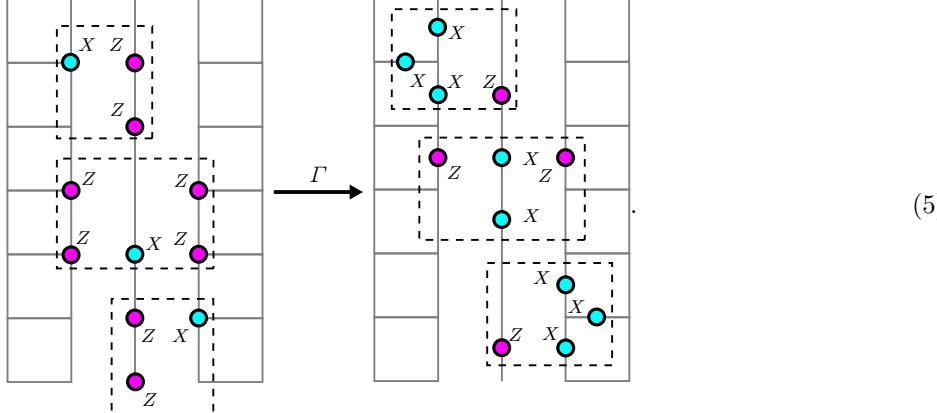

$$\tag{5}$$

The SPT wavefunctions in Eqs. (4)-(5) can be constructed from the topological action [35, 66–68]; see Appendix A 3 for a pedagogical introduction. For each $\mathbb{Z}_2$ symmetry, we introduce a $\mathbb{Z}_2$-valued background gauge field, denoted as $A_i, i \in \{1, 2, 3\}$. Then, the topological action for $S_\psi$ is

$$S_{\text{top}} = \int \frac{1}{2} A_1 \cup A_2. \tag{6}$$

The topological action for the $S_1$ domain wall is

$$S_{\text{top}} = \int \frac{1}{2} A_1 \cup A_2 + \frac{1}{2} A_2 \cup A_3. \tag{7}$$

In both cases, the term $\frac{1}{2} A_i \cup A_j$ can be understood as a domain wall decoration between the $i$'th and the $j$'th $\mathbb{Z}_2$ symmetry [66, 69]. Importantly, this approach for constructing SPT wavefunctions readily generalizes to any Abelian symmetry group.

Attentive readers might notice a similarity between SPT-sewing and gauged-SPT defects [49], since both approaches construct domain walls by gauging a trivially symmetric state decorated by lower-dimensional SPTs. However, there are important differences between the two, as we now explain.

First, for gauged-SPT defects the gauging map is applied throughout the bulk. Consequently, the gauged-SPT defects are always invertible. However, in SPT-sewing the gauging map is applied on two disconnected lattices and the obtained domain walls may or may not be invertible.

Second, the gauged-SPT defects arising from bosonic SPTs are always flux-preserving domain walls. On the other hand, the $S_\psi$ domain wall in the toric code is not flux-preserving. Thus, one should consider decorating a Kitaev Majorana fermion chain with a $\mathbb{Z}_2$ fermion parity symmetry, such that after gauging $S_\psi$ is obtained [49]. However, in a generic topological order, it is not clear if all the invertible domain walls can be constructed this way. On the other hand, the SPT-sewing constructions of invertible domain walls only use bosonic SPTs, and should be readily generalizable to domain wall constructions in generic topological orders. We now turn to such generalizations.

## III. GAUGING MAP

In this section, we review a generalization of gauging from Section II. Without loss of generality, we consider oriented graphs. We use the convention where the degrees of freedom prior to and after gauging live on the vertices and edges, respectively.

The local Hilbert space $\mathbb{C}(G)$ is assumed to be finite-dimensional and its basis is labeled by the elements of a finite group $G$. The action of the gauging map $\Gamma$ is defined on the basis states as follows

$$\tag{8}$$

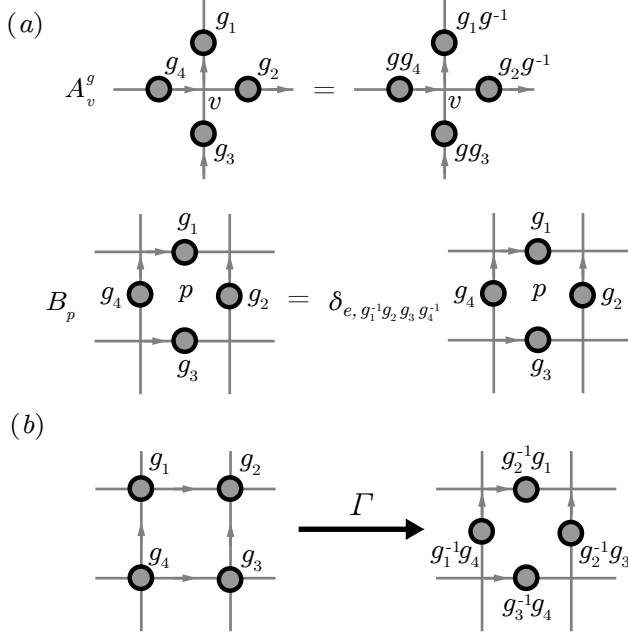

Figure 2. (a) The action of the operators $A_v^g$ and $B_p$. (b) Gauging maps any state to a zero-flux state.

It is convenient to study the action of the gauging map on local operators. We introduce the following standard left- and right-multiplication operators

$$L_+^g|h\rangle = |gh\rangle, \quad L_-^g|h\rangle = |hg^{-1}\rangle. \tag{9}$$

Prior to gauging, the action of $L_{-,v}^g$ can be described diagrammatically as follows

$$L_{-,v}^g \quad \begin{array}{c} g_1 \\ g_2 \quad g_0 \quad g_4 \\ v \\ g_3 \end{array} = \begin{array}{c} g_1 \\ g_2 \quad g_0 g^{-1} \quad g_4 \\ v \\ g_3 \end{array}. \tag{10}$$

Upon gauging, this state becomes

$$\begin{array}{c} g_1^{-1}g_0 g^{-1} \\ gg_0^{-1}g_2 \quad g_4^{-1}g_0 g^{-1} \\ v \\ gg_0^{-1}g_3 \end{array} = A_v^g \begin{array}{c} g_1^{-1}g_0 \\ g_0^{-1}g_2 \quad g_4^{-1}g_0 \\ v \\ g_0^{-1}g_3 \end{array}, \tag{11}$$

where $A_v^g$ is an appropriate product of $L_+^g$ and $L_-^g$ depending on the orientations [1]. Therefore, upon gauging we obtain

$$L_{-,v}^g \xrightarrow{\Gamma} A_v^g. \tag{12}$$

The same approach works also at the boundary of the lattice.

Under gauging, the symmetric product state gets mapped to the ground state of the quantum double model [1]. The symmetric product state

$$|+\rangle^{\otimes V} \equiv \bigotimes_{v \in V} \frac{1}{\sqrt{|G|}} \sum_g |g\rangle_v \tag{13}$$

is a $+1$-eigenstate of the operator $L^g_{-,v}$ on the vertex $v$. After gauging, this operator becomes $A^g_v$, and the state becomes a $+1$-eigenstate of $A^g_v$. By construction, the resulting state also satisfies the zero-flux condition; see Fig. 2(b). That is, it is the $+1$-eigenstate of $B_p$ defined in Fig. 2(a). The resulting state is the ground state of the following Hamiltonian

$$H = -\sum_{v \in V} A_v - \sum_{p \in F} B_p,$$

(14)

which is exactly the parent Hamiltonian of the quantum double model [1]. Thus, gauging maps the symmetric product state to a ground state of the quantum double model. A similar conclusion holds for open boundary conditions. In that case, the resulting boundary of the quantum double is analogous to the smooth boundary of the toric code.

## IV. DOMAIN WALLS IN THE 2D ABELIAN QUANTUM DOUBLE

In this section, we generalize SPT-sewing from Section II to the 2d quantum double model for any Abelian group $G$. The main result of this section is that all the domain walls can be obtained from $G \times G \times G$ SPT-sewing.

### A. Gauged-SPT domain walls

To study the general SPT-sewing construction of domain walls, we first review the domain walls in the $G$ quantum double obtained by gauging $G$ symmetric SPT phases. These domain walls are extensively studied in Ref. [49, 70, 71]; we refer to them as the *gauged-SPT* domain walls. Using the gauged-SPT domain walls and the folding trick, we will be able derive the properties of the general SPT-sewn domain walls.

Let us first pick a 1d line on the lattice, where there is a natural global $G$ symmetry action. We then put a symmetric product state $|+\rangle^{\otimes V}$ on the lattice, and decorate the line with a 1d SPT with a $G$ symmetry. The gauging map will take this decorated state into a ground state of the quantum double model, with a gapped domain wall on the line.

If we fuse together two such domain walls obtained by gauging $\text{SPT}_1$ and $\text{SPT}_2$, the resulting domain wall should be equivalent to the one obtained from gauging the stacked SPTs (usually denoted as $\text{SPT}_1 \boxtimes \text{SPT}_2$). For all the SPT states under finite symmetry group, stacking certain number of copies of the SPT state results in a trivial SPT, meaning the set of domain walls obtained this way should have a group structure. Therefore, the gauged-SPT domain walls are always invertible, i.e., all the anyons can pass through them.

Similarly to the invertible $S_\psi$ in the toric code, there could be an anyon exchange after passing through an invertible domain wall. Ref. [49] shows that the gauged-SPT domain walls correspond to some "flux-preserving" maps on anyons. When $G$ is Abelian, any anyon is specified by a pair $(e, m)$, where $e$ is the charge and $m$ is the flux, and this map is given by the slant product of the 2-cocycle[1]

$$(e, m) \to (e \cdot i_m \nu, m),$$

(15)

where $e$ and $i_m \nu$ are representations of $G$, and $m$ is an element in $G$.

Every finite Abelian group is isomorphic to a product of cyclic groups, hence $G = \prod_i \mathbb{Z}_{n_i}$. For such an Abelian group, we can understand the 2-cocycle $\nu$ via its corresponding SPT topological action; see Appendix A 3 for more details. The general form of the topological action is given by a sum over terms that couple between two cyclic subgroups

$$S_{\text{top}} = \int \sum_{i<j} \frac{k_{i,j}}{\gcd(n_i, n_j)} A^{(i)} \cup A^{(j)},$$

(16)

---

[1] In general, a 1d SPT state under a $G$ symmetry corresponds to a 2-cocycle representative $\nu$, where $\nu$ is a map from $G \times G$ to $U(1)$ satisfying the cocycle condition. Its slant product $i_g \nu$ is a representation of $G$ defined as $i_g \nu(h) := \frac{\nu(g,h)}{\nu(h,g)}$ for $g, h \in G$.

where $A^{(i)}$ is the $\mathbb{Z}_{n_i}$-valued background gauge field, and $k_{i,j} = 0, 1, \cdots, \gcd(n_i, n_j) - 1$. This SPT has a decoration of $k_{i,j}$-th power of the fundamental charge of the $\mathbb{Z}_{n_i}$ symmetry on the $\mathbb{Z}_{n_j}$ symmetry defect.

An anyon of the Abelian quantum double is specify by $(e_1 e_2 \cdots, m_1 m_2 \cdots)$, where $e_i \in \{0, 1, ..., n_i - 1\}$ is a representation of the $\mathbb{Z}_{n_i}$ subgroup and $m_j \in \{0, 1, ..., n_j - 1\}$ is an element of $\mathbb{Z}_{n_j}$. According to Eq. (15), the invertible gauged-SPT domain wall given by the topological action in Eq. (16) corresponds to the following flux-preserving map

$$(e, m) \to (e'_1 e'_2 \cdots, m), \quad e'_i = (e_i + \sum_j k_{i,j} m_j) \bmod n_i. \tag{17}$$

Let us discuss a simple example of $G = \mathbb{Z}_2 \times \mathbb{Z}_2$. The quantum double model corresponds to the two copies of the toric code. Therefore, the anyons are the compositions of $e_1, m_1, e_2, m_2$ particles, where the index $i = 1, 2$ denotes the copy. The only nontrivial gauged-SPT domain wall is given by the topological action in Eq. (6), and the corresponding flux-preserving map is

$$e_i \leftrightarrow e_i, \quad m_1 \leftrightarrow m_1 e_2, \quad m_2 \leftrightarrow m_2 e_1. \tag{18}$$

## B. $G \times G$ SPT-sewing

We start the SPT-sewing construction by taking two disconnected square lattices with close-by boundaries and putting the product state $|+\rangle^{\otimes V}$ on them. As we have already demonstrated, the gauging map will take this state to a quantum double ground state, with two adjacent smooth boundaries that condense all the flux anyons. This realizes a non-invertible domain wall of a general quantum double model. To SPT-sew this domain wall into an invertible one, we can consider putting a $G \times G$ SPT on the lattice defect.

The topological action (2-cocycle) of the group $G \times G$ is composed of terms that couple two cyclic subgroups. These terms can be divided into (i) the terms that couple two cyclic groups within each copy of $G$ and (ii) the terms that couple cyclic groups from different copies of $G$; these are called type-I and type-II actions, respectively [72]. We denote the two type-I 2-cocycles as $\omega_1$ and $\omega_2$ (corresponding to the first and second copy of $G$) and the type-II cocycle as $\eta$. Given any $G \times G$ 2-cocycle representative $\nu$, we can uniquely decompose it as $\nu = \omega_1 \eta \omega_2$.

We can use the folding trick to study the SPT-sewn domain walls with a $G \times G$ symmetry. Let us consider folding the lattice along the lattice defect, resulting in a square lattice where each edge is associated with the Hilbert space $\mathbb{C}(G) \otimes \mathbb{C}(G)$; see Fig. 3(a). The SPT-sewing construction after folding is essentially decorating the boundary with a $G \times G$ SPT. The gauging map takes the bulk into a ground state of two copies of the $G$ quantum double, and the boundary can be regarded as a gauged-SPT domain wall with the $G \times G$ symmetry, fusing with a smooth boundary. After folding, the $G \times G$ 2-cocycle is given by $\nu_f = \omega_1 \eta \overline{\omega_2}$.

According to Eq. (15), anyons passing through this gauged-SPT domain wall get mapped as follows

$$((e, e'), (m, m')) \to ((e, e') \cdot i_{mm'} \nu_f, (m, m')), \tag{19}$$

where $e, e'$ (respectively, $m, m'$) are charges (fluxes) in the first and second copy. Using the decomposition of the folded 2-cocycle $\nu_f = \omega_1 \eta \overline{\omega_2}$ we can write

$$(e, e') \cdot i_{mm'} \nu_f = \left( e(i_m \omega_1)(i_{m'} \eta), e'(i_m \eta)(i_{m'} \overline{\omega_2}) \right), \tag{20}$$

for any group elements $m, m' \in G$. The smooth boundary itself condenses all the flux anyons. Therefore, this boundary (viewed in the folded picture as a gauged-SPT domain wall followed by a smooth boundary) condenses the following anyons

$$\left( \left( \overline{i_m \omega_1} \cdot \overline{i_{m'} \eta}, \overline{i_m \eta} \cdot i_{m'} \omega_2 \right), (m, m') \right), \tag{21}$$

where we use $\bar{c}$ to denote the anti-particle of $c$ anyon.

If we unfold this condensation picture, we conclude that the anyon

$$\left( \overline{i_m \omega_1} \cdot i_{m'} \eta, m \right), \tag{22}$$

coming from the left of an SPT-sewn domain wall can be condensed together with another anyon

$$\left( i_m \eta \cdot i_{m'} \omega_2, m' \right), \tag{23}$$

coming from the right, for any $m, m' \in G$.

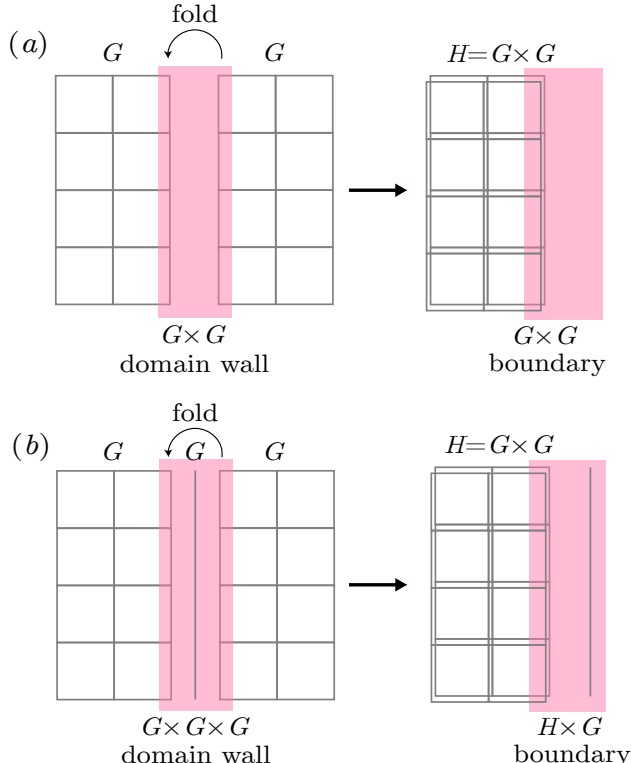

Figure 3. (a) After folding the system along the domain wall, the bulk becomes an $H = G \times G$ quantum double, and an SPT-sewn domain wall with a $G \times G$ symmetry becomes a gapped boundary, which is an $H$ gauged-SPT domain wall followed by a smooth boundary. (b) After folding, an SPT-sewn domain wall with a $G \times G$ symmetry becomes a $H \times G$ gauged-SPT domain wall followed by a smooth boundary. Note that by adding extra unentangled $G$ degrees of freedom along the boundary, we can reduce this scenario to the one in (a).

**Lemma 1.** *In a 2d quantum double (gauge theory) of a finite Abelian group $G$, an SPT-sewn domain wall constructed from a $G \times G$ SPT is invertible if and only if the type-II part of the 2-cocycle $\nu$ is non-degenerate.*

*Proof.* We can further regard this SPT-sewn domain wall as a fusing of a gauged-SPT domain wall characterized by $\omega_1$ with an SPT-sewn domain wall characterized by $\eta_{12}$ and another gauged-SPT domain wall characterized by $\omega_2$. Since the two gauged-SPT domain walls are always invertible, and their corresponding anyon maps are known, we can set $\omega_1 = \omega_2 = 1$ and focus on the contribution from the SPT-sewing part.

From above, this domain wall condenses an anyon $(i_{m'}\eta, m)$ coming from the left of an SPT-sewn domain wall can be condensed together with another anyon $(i_m\eta, m')$ coming from the right, for any $m, m' \in G$. Therefore, this domain wall is always invertible, and corresponds to some charge-flux exchange map, unless there exists an element $g \in G$, such that $i_g\eta \equiv 1$. $\qquad\square$

The anyon map given by such an invertible SPT-sewn domain wall characterized by a 2-cocycle $\nu = \omega_1\eta\omega_2$ can be understood as follows. Suppose an anyon $(e, m)$ is coming from the left bulk. It will be mapped sequentially as

$$(e, m) \xrightarrow{\omega_1} (e', m) \xrightarrow{\eta} (e'', m'') \xrightarrow{\omega_2} (e''', m''), \tag{24}$$

where the first and third arrows are some flux-preserving maps given by Eq. (15), and the second arrow is a charge-flux exchange map given above.

## C. $G \times G \times G$ SPT-sewing

The invertible domain walls constructed above are richer than the gauged-SPT domain walls. However, there is still a large number of invertible domain walls that cannot be constructed from sewing with $G \times G$ SPTs. Thus, we now consider SPT-sewing with $G \times G \times G$ symmetry, and show that this construction gives rise to all the invertible domain walls for Abelian quantum double models.

Let us first consider $G = \mathbb{Z}_2^n$. The quantum double model for $G$ is therefore $n$ copies of the toric code. The anyons are compositions of the charge $e_i$ and flux $m_j$, which are their own anti-particles. The braiding between $e_i$ and $e_j$ (or $m_i$ and $m_j$) is trivial, whereas the braiding between $e_i$ and $m_j$ gives rise to a phase $(-1)^{\delta_{ij}}$, where $i, j = 1, \cdots, n$. This is the same as the algebra of the Pauli operators in an $n$-qubit system, by equating $e_i$ with Pauli $Z_i$, $m_j$ with Pauli $X_j$, and equating the braiding between anyons with the commutator between two operators $UVU^\dagger V^\dagger$.

Furthermore, as we pointed out, the invertible domain walls are in a one-to-one correspondence with the maps on the anyons that are compatible with the anyon fusion and braiding, and are thus called braided autoequivalences [20, 21, 73]. It is straightforward to see that these anyon maps correspond to automorphisms of the $n$-qubit Pauli group. To be more precise, the group form by the invertible domain walls in $\mathbb{Z}_2^n$ quantum double is the quotient of the Clifford group by the Pauli group and phases (which is isomorphic to the symplectic group of degree $2n$ over $\mathbb{Z}_2$).

**Lemma 2.** *In a 2d quantum double (gauge theory) of $G = \mathbb{Z}_2^n$, all the invertible domain walls (invertible defects) can be constructed from SPT-sewing with $G \times G \times G$ symmetry.*

We sketch the proof idea; see Appendix B for details. Let us start from the conjugation action on the Pauli $Z_i$ and $X_j$ operators by any $n$-qubit Clifford group element. According to Theorem 1 in Ref. [74], with some adjustments, any action on the Pauli operators is the conjugation by the following canonical unitary operator $U = \Omega \Pi \Omega'$, where

$$\Omega = \prod_{i,j=1}^n CZ_{i,j}^{\Gamma_{i,j}}, \quad \Omega' = \prod_{i,j=1}^n CZ_{i,j}^{\Gamma'_{i,j}}, \quad \Pi = \left( \prod_{i \neq j} CX_{i,j}^{\Delta_{i,j}} \right) S \left( \prod_{i=1}^n H_i^{h_i} \right) \left( \prod_{i \neq j} CX_{i,j}^{\Delta'_{i,j}} \right), \quad (25)$$

$h_i, \Gamma_{i,j}, \Gamma'_{i,j} = 0, 1$, and $S$ is a permutation operator for $n$ qubits. The matrices $\Delta, \Delta'$ are upper-triangular and unit-diagonal, i.e., $\Delta_{i,j} = \Delta'_{i,j} = 0$ for $i > j$ and $\Delta_{i,i} = \Delta'_{i,i} = 1$ for all $i$. The product of $CX$ gates is ordered, such that the control qubit index increases from the left to the right. For example, when $n = 4$, $\prod_{i \neq j} CX_{i,j}^{\Delta_{i,j}} = CX_{1,2}^{\Delta_{1,2}} CX_{1,3}^{\Delta_{1,3}} CX_{1,4}^{\Delta_{1,4}} CX_{2,3}^{\Delta_{2,3}} CX_{2,4}^{\Delta_{2,4}} CX_{3,4}^{\Delta_{3,4}}$.

As we argued, the conjugation by the Clifford operators on the Pauli $Z_i$ and $X_j$ operators corresponds to the braided autoequivalences of anyons $e_i$ and $m_j$. Thus, the unitaries $\Omega$ and $\Omega'$ correspond to the flux-preserving maps given by the gauged-SPT domain walls. We claim that the following SPT-sewn domain wall with $G \times G \times G$ symmetry gives rise to the corresponding anyon map

$$
\begin{aligned}
S_{\text{top}} = \int \sum_{i,j} \frac{\Gamma_{i,j}}{2} C_i \cup C_j + \sum_{i,j} \frac{\Gamma'_{i,j}}{2} A_i \cup A_j + \sum_{\substack{i,j,k, \\ h_j=0}} \frac{\Delta'_{i,j}}{2} A_i \cup B_{s(j)} \\
+ \frac{\overline{\Delta}_{k,s(j)}}{2} B_{s(j)} \cup C_k + \sum_{\substack{i,j,k, \\ h_j=1}} \frac{\Delta'_{i,j} \overline{\Delta}_{k,s(j)}}{2} A_i \cup C_k,
\end{aligned}
\tag{26}
$$

where $\overline{\Delta}$ is the inverse matrix. $A_i$, $B_i$, and $C_i$ are the background gauge fields for the $i$-th $\mathbb{Z}_2$ subgroup in the first, second, and third $G$ symmetry.

To study the anyon map given by the above topological action, let us fold the topological order along the domain wall, such that the bulk becomes a quantum double of a group $G \times G$, and the domain wall becomes a gapped boundary, as shown in Fig. 3(b). The set of anyons condensing on this boundary can be derived similarly as in Section IV B. After unfolding, the anyon map is essentially from the component of a condensed anyon that is in the first copy, to the component that is in the second copy of the $G$ quantum double.

In Appendix B 1, we derive the anyon map from the condensed anyons on the folded boundary of $G \times G$ quantum double, and show that it exactly corresponds to the conjugation of the canonical Clifford operator

in Eq. (25). In fact, we can generalize the above result to an arbitrary finite Abelian group. We present it as the following theorem and defer the proof to Appendix B 2.

**Theorem 1.** *In a 2d quantum double (gauge theory) of a finite Abelian group $G$, all the invertible domain walls can be constructed from SPT-sewing with $G \times G \times G$ symmetry.*

### D.    Example: $G = \mathbb{Z}_2 \times \mathbb{Z}_2$

The $\mathbb{Z}_2 \times \mathbb{Z}_2$ quantum double is essentially the two copies of the $\mathbb{Z}_2$ toric codes. It is also equivalent to the 2d color code by introducing ancilla qubits and applying local unitary transformations [75–77]. The 2d color code has been extensively studied in the quantum information community due to its capability to support fault-tolerant quantum operations, in particular transversal logical gates [75]. Recently, it was shown that by applying a non-destructive sequence of measurements (anyon condensations), fault-tolerant logical gates in the second level of the Clifford hierarchy can be realized [78, 79]. In addition to its computational advantages, the classification of gapped boundaries, domain walls, and twist defects of the 2d color code has also been studied [80, 81]. In this subsection, we provide a new perspective on the gapped domain walls in this topological order through SPT-sewing.

The correspondence between anyons in the two copies of the toric code and the anyons in the 2d color code are listed in the following table.

| $e_1$ | $e_1 e_2$ | $e_2$ | | $r_x$ | $g_x$ | $b_x$ |
|---|---|---|---|---|---|---|
| $e_1 m_2$ | $f_1 f_2$ | $m_1 e_2$ | $\longleftrightarrow$ | $r_y$ | $g_y$ | $b_y$ |
| $m_2$ | $m_1 m_2$ | $m_1$ | | $r_z$ | $g_z$ | $b_z$ |

The braided auto-equivalences in the model are given by the permutations of the rows and columns, corresponding to $S_3 \times S_3$, and the transpose of the table, corresponding to $\mathbb{Z}_2$. Thus, the braided auto-equivalences together form the group $(S_3 \times S_3) \rtimes \mathbb{Z}_2$, with 72 elements. We now list some examples of gauged-SPT and SPT-sewing constructions of the domain walls, and their corresponding anyon maps.

As we showed in Eq. (18), the gauged-SPT domain wall with a $\mathbb{Z}_2 \times \mathbb{Z}_2$ symmetry given by the topological action

$$S_{\text{top}} = \frac{1}{2} \int A_1 \cup A_2 \tag{27}$$

gives rise to a flux-preserving map

$$e_i \leftrightarrow e_i, \quad m_1 \leftrightarrow m_1 e_2, \quad m_2 \leftrightarrow m_2 e_1. \tag{28}$$

The SPT-sewn domain walls with the $(\mathbb{Z}_2 \times \mathbb{Z}_2)^2$ topological actions

$$S_{\text{top}} = \frac{1}{2} \int A_1 \cup B_1 + A_2 \cup B_2;$$
$$S_{\text{top}} = \frac{1}{2} \int A_1 \cup B_2 + A_2 \cup B_1, \tag{29}$$

give rise to the following charge-flux exchange maps

$$e_1 \leftrightarrow m_1, \quad e_2 \leftrightarrow m_2;$$
$$e_1 \leftrightarrow m_2, \quad e_2 \leftrightarrow m_1. \tag{30}$$

The above domain walls, and all the other invertible domain walls can be constructed from SPT-sewing with $(\mathbb{Z}_2 \times \mathbb{Z}_2)^3$ symmetry. For instance, from the topological actions

$$S_{\text{top}} = \frac{1}{2} \int A_1 \cup B_2 + A_2 \cup B_1 + \sum_i B_i \cup C_i;$$
$$S_{\text{top}} = \frac{1}{2} \int A_1 \cup B_2 + \sum_i A_2 \cup B_i + B_i \cup C_i, \tag{31}$$

we can obtain the domain walls with the following anyon maps

$$e_1 \leftrightarrow e_2, \quad m_1 \leftrightarrow m_2;$$
$$m_1 \to m_2, \ m_2 \to m_1 m_2, \ e_2 \to e_1, \ e_1 \to e_1 e_2. \tag{32}$$

## V.  NON-ABELIAN EXAMPLE: $S_3$ QUANTUM DOUBLE

In this section, we discuss the construction of the domain walls in the non-Abelian quantum double models by taking $G = S_3$ as an example. In Section V A, we provide a brief introduction to the $S_3$ quantum double, including the anyonic excitations and the corresponding ribbon operators. Then, we discuss the gauged-SPT and SPT-sewn domain wall constructions for the $S_3$ quantum double. Specifically, in Section V B, we show that there is no nontrivial gauged-SPT domain wall with an $S_3$ symmetry. In Section V C, we study the SPT-sewn domain wall with an $S_3 \times S_3$ symmetry, and show that it is not invertible. In Section V D, we show that the invertible domain walls, especially the $C \leftrightarrow F$ domain wall [7, 21], can be constructed from SPT-sewing with an $S_3 \times \mathrm{Rep}(S_3) \times S_3$ symmetry. Based on these results, we conjecture that the invertible domain walls in a generic quantum double model can be constructed via $G \times \mathrm{Rep}(G) \times G$ SPT-sewing.

### A.  $S_3$ quantum double

The group of permutations on a set of three elements $S_3$ is isomorphic to the dihedral group $D_3$, the symmetry of an equilateral triangle. There are two generators $c$ and $t$ satisfying the following relations

$$c^3 = t^2 = e, \quad tct = c^2, \tag{33}$$

where $e$ is the identity element.

The local Hilbert space $\mathbb{C}(S_3)$ has an orthonormal basis $\{|g\rangle : g \in S_3\}$. Furthermore, we define the following operators

$$L_+^g |h\rangle = |gh\rangle, \qquad T_+^g |h\rangle = \delta_{g,h} |h\rangle,$$
$$L_-^g |h\rangle = |hg^{-1}\rangle, \quad T_-^g |h\rangle = \delta_{g^{-1},h} |h\rangle, \tag{34}$$

where $g, h \in S_3$. $L_\pm^g$ operators can be understood as generalized Pauli $X$ operators, while the linear combinations of $T_\pm^g$ can be understood as generalized Pauli $Z$ operators. For the convenience of later discussion, we define the following operators

$$\widetilde{Z}_L = \sum_{j,k} e^{2\pi i j/3} T_+^{c^j t^k},$$
$$\widetilde{Z}_R = \sum_{j,k} e^{2\pi i j/3} T_+^{t^k c^j}, \tag{35}$$
$$Z = \sum_{j,k} (-1)^k T_+^{c^j t^k}.$$

For more details about the operators in $\mathbb{C}(S_3)$, see Appendix D 2.

The $S_3$ quantum double model can be defined on a square lattice by associating each edge with the Hilbert space $\mathbb{C}(S_3)$ and taking the Hamiltonian

$$H = -\sum_{v \in V} A_v - \sum_{p \in F} B_p, \tag{36}$$

where $A_v = \frac{1}{|G|} \sum_g A_v^g$. The commuting local projectors $A_v^g$ and $B_p$ are defined in Fig. 2. The anyons in this non-Abelian quantum double are given by the pair $([g], \rho)$, where the flux $[g] = \{hgh^{-1} | h \in S_3\}$ is a

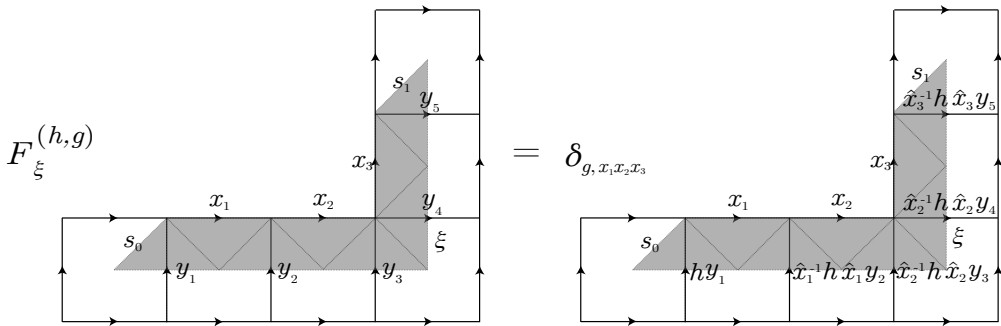

Figure 4. Ribbon $\xi$ starts from a site $s_0$ and ends at a site $s_1$. The ribbon operator $F_\xi^{(h,g)}$ on this ribbon is composed of some shift operators on the edges and a Kronecker delta.

conjugacy class, and the charge $\rho$ is an irreducible representation of the centralizer $Z_g = \{h|hg = gh\}$. They are shown in the following table [7]

|  | $A$ | $B$ | $C$ | $D$ | $E$ | $F$ | $G$ | $H$ |
|---|---|---|---|---|---|---|---|---|
| flux | $e$ | $e$ | $e$ | $[t]$ | $[t]$ | $[c]$ | $[c]$ | $[c]$ |
| charge | $\mathbf{1}$ | $s$ | $\pi$ | $\mathbf{1}$ | $-\mathbf{1}$ | $\mathbf{1}$ | $\omega$ | $\omega^*$ |

where $s$ denotes a one-dimensional sign representation

$$\rho_s(c) = 1, \quad \rho_s(t) = -1, \tag{37}$$

and $\pi$ is a two-dimensional representation

$$\rho_\pi(c) = \begin{pmatrix} e^{\frac{2\pi i}{3}} & 0 \\ 0 & e^{\frac{-2\pi i}{3}} \end{pmatrix}, \quad \rho_\pi(t) = \begin{pmatrix} 0 & 1 \\ 1 & 0 \end{pmatrix}. \tag{38}$$

Moreover, $-\mathbf{1}$ denotes the nontrivial representation of $Z_t = \{1, t\}$, and $\omega, \omega^*$ denote the nontrivial representations of $Z_c = \{1, c, c^2\}$.

These anyonic excitations can be created on the lattice by the so-called ribbon operators, generalizing the Pauli $X$ and $Z$ string operators in the toric code. On an oriented lattice, a site is given by a pair $s = (v, f)$, where $v$ is one of the vertices of a face $f$. A ribbon is a "path" of neighboring sites. Given a ribbon $\xi$, the action of the ribbon operator $F_\xi^{(h,g)}$ in the group basis is shown in Fig. 4 [7, 62].

The multiplication of ribbon operators on the same ribbon $\xi$ is

$$F_\xi^{h_1,g_1} F_\xi^{h_2,g_2} = \delta_{g_1,g_2} F_\xi^{h_1 h_2,g_2}. \tag{39}$$

If the end of a ribbon $\xi_1$ is the start of another ribbon $\xi_2$, we can denote the composition of the two ribbons as $\xi = \xi_1 \xi_2$. The ribbon operator $F_\xi^{h,g}$ on ribbon $\xi$ obeys the co-multiplication rule

$$F_\xi^{h,g} = \sum_{k \in S_3} F_{\xi_1}^{h,k} F_{\xi_2}^{k^{-1}hk,k^{-1}g}. \tag{40}$$

The ribbon operators that create a pair of certain anyons $([g], \rho)$ at the ends of the ribbon $\xi$ are given by a linear combination of $F_\xi^{h,g}$ for different $h, g \in S_3$. For example, both $C$ and $F$ in $S_3$ quantum double have two internal degrees of freedom, so they have quantum dimensions $d_C = d_F = 2$. Correspondingly, their ribbon operators creating $C$ and $F$ can be viewed as $2 \times 2$ matrices, with their entries given as follows

$$\begin{aligned} (F_\xi^C)_{ij} &= \frac{1}{3} \sum_{g \in S_3} \rho_\pi^{-1}(g)_{ij} F_\xi^{e,g}, \\ (F_\xi^F)_{ij} &= \frac{1}{3} \sum_{k \in Z_c} F_\xi^{c^{-i}, t^i k t^j}, \end{aligned} \tag{41}$$

where $Z_c = \{1, c, c^2\}$; see Appendix D 3 for details.

## B.  $S_3$ **gauged-SPT domain wall**

The gauged-SPT domain wall construction reviewed in Section IV A also works for non-Abelian groups. The resulting domain walls are always invertible and give rise to some flux-preserving anyon maps [49]. For $G = S_3$ there is no nontrivial $S_3$ SPT, since the second cohomology group is trivial, i.e.,

$$\mathcal{H}^2(S_3, U(1)) = 1. \tag{42}$$

Thus, the gauged-SPT domain wall construction for the $S_3$ quantum double gives no nontrivial results.

## C.  $S_3 \times S_3$ **SPT-sewn domain wall**

Now we consider the SPT-sewn domain walls with a $S_3 \times S_3$ symmetry. The SPT phases are classified by the second cohomology group, which is

$$\mathcal{H}^2\left(S_3 \times S_3, U(1)\right) = \mathbb{Z}_2. \tag{43}$$

Eq. (43) follows from the Künneth formula [82, 83], which we review in Appendix A 6.

Gauging different SPTs lead to different types of domain walls. Gauging the trivial SPT, we obtain smooth boundaries on both sides. For the nontrivial SPT, upon gauging we obtain a non-invertible domain wall. Only a subset of the anyons can pass through the domain wall. The action of the domain wall on these anyons is

$$A \leftrightarrow A, \quad B \leftrightarrow D, \quad E \leftrightarrow E. \tag{44}$$

The other anyons are condensed on the domain wall.

We now focus on the high-level reasoning behind Eq. (44); we provide an explicit lattice construction of this domain wall in Appendix E 1. Using the Künneth formula, we obtain

$$\mathcal{H}^2\left(S_3 \times S_3, U(1)\right) = \mathcal{H}^1(S_3, \mathcal{H}^1(S_3, U(1))), \tag{45}$$

where $\mathcal{H}^1(S_3, U(1))$ is the group of one-dimensional representations of $S_3$. Eq. (45) implies that $\mathcal{H}^2\left(S_3 \times S_3, U(1)\right)$ is determined from a pairing between elements of $S_3$ and its one-dimensional representations. This is analogous to the case of the toric code in Section IV B, where the 2-cocycle $\eta$ represented a nontrivial pairing between elements of $\mathbb{Z}_2$ and its representations. The corresponding SPT-sewn domain wall gave rise to the $e \leftrightarrow m$ domain wall. Similarly, we can expect the pairing between the group elements and the group representations to correspond to a charge-flux exchange.

Since the only nontrivial one-dimensional representation of $S_3$ is the sign representation in Eq. (37), the elements $tc^i$ for $i = 0, 1, 2$ have nontrivial pairings. Thus, an anyon with the $t$ flux passing through the domain wall becomes an anyon with the $s$ charge, and vice versa. The map is thus given in Eq. (44), and all the other anyons cannot pass through this domain wall.

We remark that the $S_3$ quantum double can be reduced to the toric code model by applying a projection on each edge, leaving a state on a two-dimensional subspace spanned by $|e\rangle$ and $|t\rangle$. In doing so, $A \to 1$, $B \to e$, $D \to m$ and $E \to em$ [84]. Therefore, the above SPT-sewn domain wall becomes the $e \leftrightarrow m$ exchange domain wall in the toric code.

## D.  $S_3 \times \text{Rep}(S_3) \times S_3$ **SPT-sewn domain walls**

We found that the domain walls obtained from gauging a $S_3 \times S_3$ SPT is noninvertible. It turns out the same conclusion holds for $S_3 \times S_3 \times S_3$ symmetry. Thus the approach used for Abelian quantum doubles no longer works for non-Abelian quantum doubles. In what follows, we describe an alternative method which does lead to an invertible domain wall.

The crucial new ingredient we use is the *non-invertible symmetry* [52–54]. Specifically, we consider a non-invertible $S_3 \times \text{Rep}(S_3) \times S_3$ symmetry and apply our SPT-sewing method. The $\text{Rep}(S_3)$ symmetry contains the trivial representation $\mathbf{1}$, a one-dimensional sign representation $s$, and a two-dimensional representation $\pi$. The fusion rules are given by the tensor product of representations [85, 86].

| $\otimes$ | $\mathbf{1}$ | $s$ | $\pi$ |
|---|---|---|---|
| $\mathbf{1}$ | $\mathbf{1}$ | $s$ | $\pi$ |
| $s$ | $s$ | $\mathbf{1}$ | $\pi$ |
| $\pi$ | $\pi$ | $\pi$ | $\mathbf{1} \oplus s \oplus \pi$ |

As before, we consider three layers of 1d lattices, wherein each vertex is associated with a Hilbert space $\mathbb{C}(S_3)$. We use $a_i, b_i$, and $c_i$ to label the vertices on the respective layer. The two $S_3$ symmetries are given by

$$A_g := \prod_i L^g_{+,a_i}, \quad C_g := \prod_i L^g_{+,c_i} \sigma^g_{b_i}, \quad \forall g \in S_3, \tag{46}$$

where the shift operator $L^g_+$ is defined in Eq. (34), and $\sigma^g_{b_i} \equiv \sum_h \left| ghg^{-1} \right\rangle_{b_i} \langle h |$ is a conjugation by element $g$ on qudit at vertex $b_i$. The $\mathrm{Rep}(S_3)$ symmetry is given by a matrix product operator (MPO)

$$B_\rho := \mathrm{Tr}\left( \prod_i Z_{\rho, b_i} \right), \quad \forall \rho \in \mathrm{Rep}(S_3), \tag{47}$$

which deserves a further explanation. First of all, each matrix $Z_\rho$ is defined as

$$Z_\rho = \sum_{g \in S_3} \rho(g) \otimes |g\rangle\langle g|, \tag{48}$$

where $\rho(g)$ acts on the auxiliary (virtual) space. The product of these operators are assumed to follow a particular order, i.e., $\prod_i Z_{\rho, b_i} = \ldots Z_{\rho, b_2} Z_{\rho, b_1}$. The trace in Eq. (47) is taken over the virtual space.

In the literature, SPTs with respect to the $S_3 \times \mathrm{Rep}(S_3) \times S_3$ has not been studied to the best of our knowledge. Nonetheless, there are some related prior works. The $S_3 \times \mathrm{Rep}(S_3) \times S_3$ is a fusion category, which can be seen as an example of the non-invertible symmetry [52–54]. Unlike the classification of bosonic SPT phases with an ordinary symmetry via group cohomology, the SPT phases with a non-invertible symmetry correspond to the fiber functors of a category, and their classifications are still elusive except for some special families of categories [50, 51]. Nonetheless, there are interesting exactly solvable models that respect a related symmetry; see Ref. [55, 87, 88] for studies on SPTs with $G \times \mathrm{Rep}(G)$ symmetry.

We consider two types of SPTs, one being the trivial SPT and the other being the nontrivial one. Upon gauging these, we obtain two invertible domain walls of the $S_3$ quantum double. The first is the trivial domain wall, and the second type of domain wall corresponds to the braided autoequivalence that exchanges $C$ and $F$ anyons [7, 21].

### 1. Trivial domain wall

Let us first start with the trivial domain wall. We consider an $S_3 \times \mathrm{Rep}(S_3) \times S_3$ SPT state given by the wavefunction

$$|SPT_1\rangle = \sum_{\{h_{a_i}, h_{c_i}\}} \left| \cdots, h_{a_i}, h_{b_i}, h_{c_i}, h_{a_{i+1}}, \cdots \right\rangle, \tag{49}$$

where $h_{b_i} \equiv h_{c_i} h_{a_i}^{-1} h_{a_{i-1}} h_{c_{i-1}}^{-1}$ for all vertex $i$. Eq. (49) can be obtained preparing states in $\{a_i\}$ and $\{c_i\}$ in the uniform superposition state and $\{b_i\}$ in $|e\rangle$, followed by a sequence of controlled multiplications [55, 87]. Thus this state can be understood as a generalized cluster state. This state is symmetric under the symmetry operators defined in Eq. (46) and Eq. (47). We can write a Hamiltonian for this state as composed of the

following commuting stabilizers

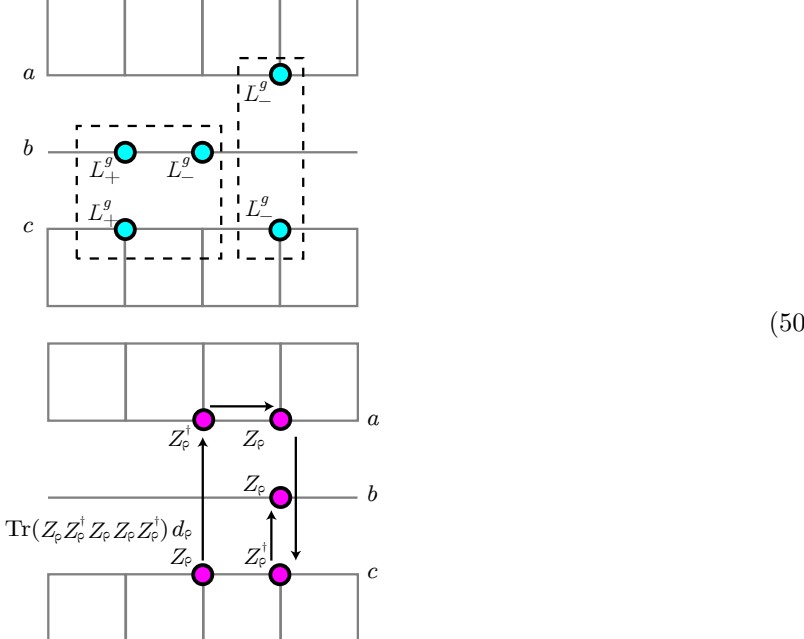

$$(50)$$

where $g \in S_3$, $\rho \in \text{Rep}(S_3)$ and $d_\rho$ is the dimension of $\rho$. The order of the $Z_\rho$ operators are taken as directed in Eq. (50). This Hamiltonian has a unique ground state, and commutes with the symmetry operators, hence is an $S_3 \times \text{Rep}(S_3) \times S_3$ SPT.

To see that this SPT gives rise to a trivial domain wall after gauging, we consider a local projection on each $b_i$ vertex to $|e\rangle$. Since this projection does not commute with some of the SPT stabilizers, the stabilizers for the projected state are given by some combinations or reductions of the original stabilizers. Thus the domain wall stabilizers from gauging the projected state are also given by combinations or reductions of the SPT-sewn domain wall stabilizers. It is then obvious that, if an anyon $a$ can pass through the former domain wall, then $a$ can also pass through the SPT-sewn domain wall. Hence, if the former domain wall is invertible, then the SPT-sewn domain wall is also invertible, and gives rise to same braided autoequivalence (anyon map).

After this projection, the $S_3 \times S_3$ symmetry on the top and bottom layers are given by

$$
\begin{aligned}
U_g^a &:= \prod_i L_{+,a_i}^g, \quad \forall g \in S_3, \\
U_g^c &:= \prod_i L_{+,c_i}^g, \quad \forall g \in S_3.
\end{aligned}
\tag{51}
$$

After gauging, the stabilizers of the projected state becomes the domain wall stabilizers as follows

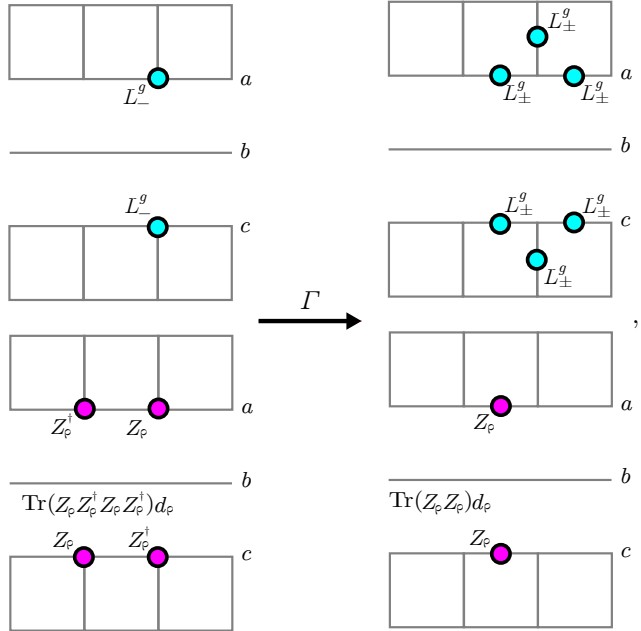

which gives rise to a trivial domain wall. The order of the $Z_\rho$ operators in the lower left are taken in the same order as Eq. (50) without the last $Z_\rho$ on layer $b$. Therefore, the $S_3 \times \mathrm{Rep}(S_3) \times S_3$ SPT-sewn domain wall given from Eq.(49) is an invertible trivial domain wall. We note that this SPT state can be defined for any finite group $G$, and the SPT-sewing construction would always give rise to a trivial domain wall in a $G$ quantum double model.

### 2. $C \leftrightarrow F$ domain wall

Now we consider another $S_3 \times \mathrm{Rep}(S_3) \times S_3$ SPT which becomes the domain wall exchanging $C$ and $F$ anyon after gauging. Given an $S_3$ group element, we can always decompose it as $g = nq$, where $n = c^i$ and $q = t^k$ for some integers $i$ and $k$. The SPT state is given by

$$|SPT_2\rangle = \sum_{\{h_{a_i}, h_{c_i}\}} |\cdots, h_{a_i}, h_{b_i}, h_{c_i}, h_{a_{i+1}}, \cdots\rangle, \tag{52}$$

where $h_{b_i} \equiv h_{c_i} q_{a_i}^{-1} q_{a_{i-1}} h_{c_{i-1}}^{-1}$ for all vertex $i$. The stabilizers of its Hamiltonian are shown in Fig. 5. These stabilizers commute with each other, and are symmetric under the $S_3 \times \mathrm{Rep}(S_3) \times S_3$ symmetry. Thus they give rise to a non-invertible SPT phase.

To show that the SPT-sewn domain wall is the invertible $C \leftrightarrow F$ domain wall, we project each $b_i$ vertex to state $\frac{1}{\sqrt{3}}(|e\rangle + |c\rangle + |c^2\rangle)$, such that $Z = 1$ and $L_-^c = 1$ acting on $b_i$ for all $i$. It turns out that the projected state spontaneously breaks the $S_3 \times S_3$ symmetry into a $(\mathbb{Z}_3 \times \mathbb{Z}_3) \rtimes \mathbb{Z}_2$ symmetry. Furthermore, it corresponds to a nontrivial SPT phase under the unbroken symmetry. In Appendix E 2, we show that the above stabilizers are consistent with the standard fixed-point SPT state construction using 2-cocycles [69].

We now list the stabilizers for the projected state. The first set of stabilizers gives rise to the spontaneous breaking of $\mathbb{Z}_2 \times \mathbb{Z}_2$ generated by $U_t^a$ and $U_t^c$, to the diagonal subgroup generated by operator $U_t^a U_t^c$ defined in Eq. (51). The stabilizer on the left is the localized symmetry operators of the diagonal $\mathbb{Z}_2$ subgroup, and

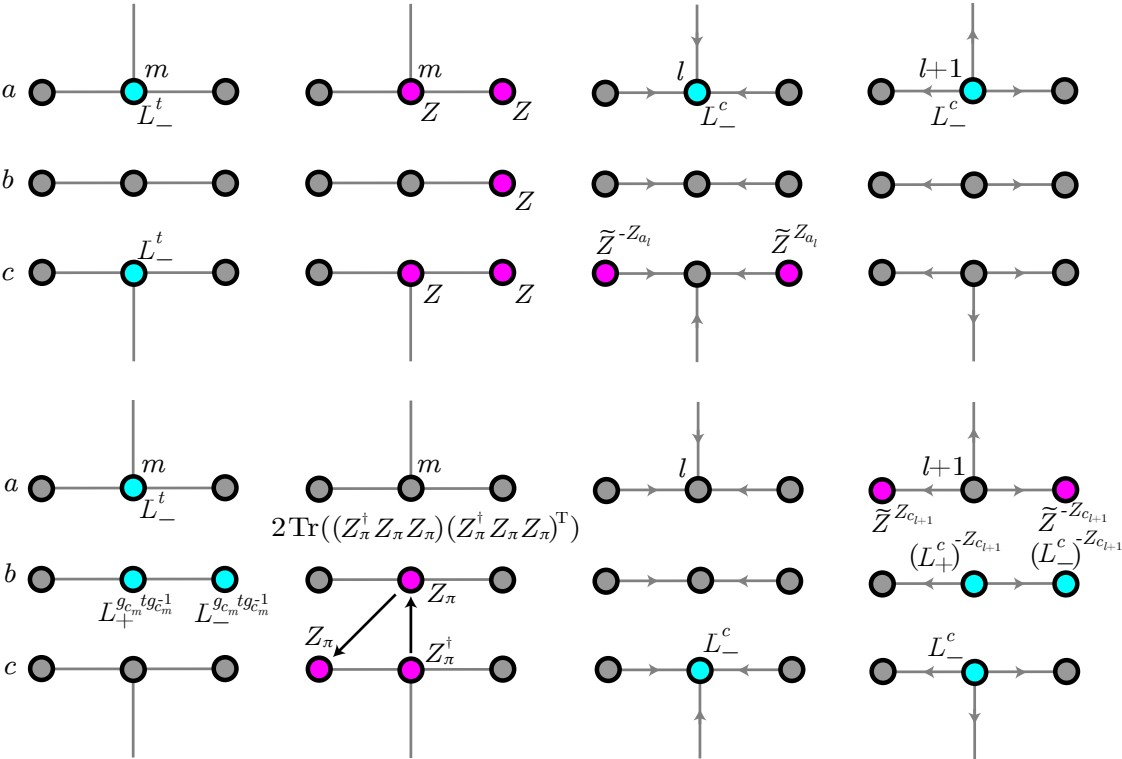

Figure 5. An $S_3 \times \text{Rep}(S_3) \times S_3$ SPT is defined by a Hamiltonian composed of these commuting stabilizers; see Eq (35) for the definition of the operators. In this figure, $m$ and $l$ are site labels for odd and even sites, and $a$, $b$, and $c$ are the labels of each layer. The term $g_{c_m}$ corresponds to the state at site $m$ in layer $c$ and $Z_{c_{l+1}}$ represents the eigenvalue of the Pauli $Z$ operator at site $l+1$ in layer $c$.

the stabilizer on the right is the symmetry breaking order parameter

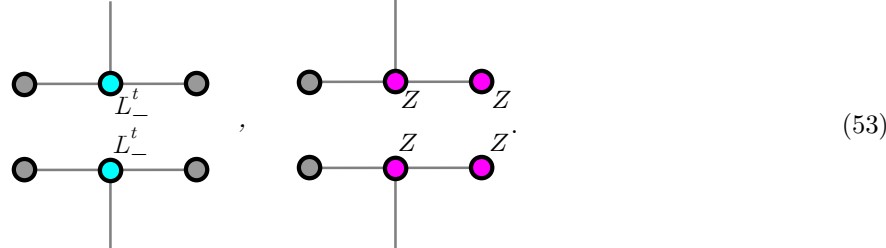

$$\tag{53}$$

The above stabilizers are translational-invariant along the domain wall.

The other set of stabilizers gives rise to the SPT phases under the unbroken $(\mathbb{Z}_3 \times \mathbb{Z}_3) \rtimes \mathbb{Z}_2$ symmetry. This SPT entangles the $\mathbb{Z}_3$ part of the top qudits and the $\mathbb{Z}_3$ part of the bottom qudits, mediated by the $\mathbb{Z}_2$

part of qudits

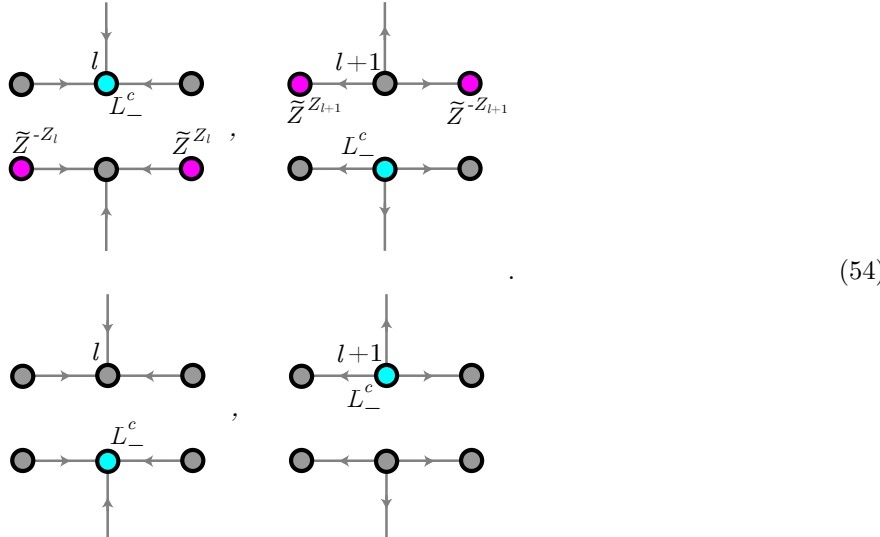

$$\tag{54}$$

Here we assume $l$ is even. $Z_l$ is the eigenvalue of Pauli $Z$ operator at site $l$.

Now we gauge the $S_3$ symmetry of the upper chain together with the product state $|+\rangle$ in the bulk above, and gauge the $S_3$ symmetry of the lower chain together with the product state $|+\rangle$ in the bulk below. The gauging map will take this state to an $S_3$ quantum double ground state, with a gapped domain wall. The domain wall stabilizers are obtained from the above SPT stabilizers via the gauging map defined in Appendix D 1, and the calculations are presented in Appendix E 2.

We summarize all the stabilizers below,

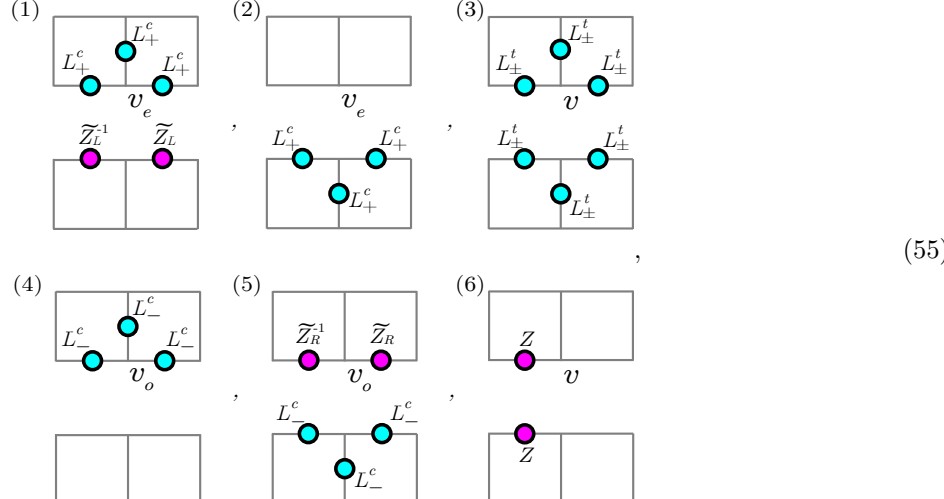

$$\tag{55}$$

in which $\widetilde{Z}_L$, $\widetilde{Z}_R$, and $Z$ are defined in Eq. (35). $v_e$ represents an even site, $v_o$ represents an odd site, and $v$ represents a general site, both even and odd.

If we take a ribbon $\xi$ to be the composition of a ribbon $\xi_1$ above the domain wall, and another ribbon $\xi_2$ below the domain wall as displayed below. In Appendix E 2, we show that the following ribbon operator

$$F_\xi^{CF} = 3F_{\xi_1}^C F_{\xi_2}^F, \tag{56}$$

where $\xi = \xi_1\xi_2$, commutes with all the domain wall stabilizers. Thus, this ribbon operator only creates an $F$ anyon at one end of $\xi_2$, and a $C$ anyon at the other end of $\xi_1$. In other words, anyon $C$ and $F$ are exchanged when passing through this domain wall. Therefore, the SPT-sewn domain wall given by stabilizers in Fig. 5

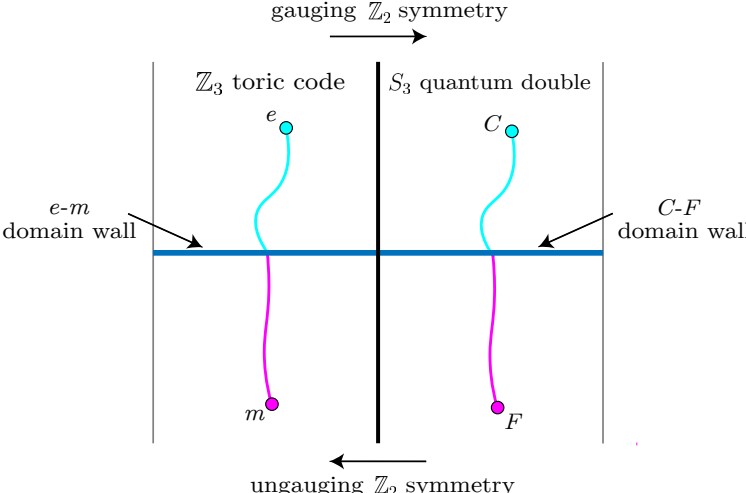

Figure 6. The $C \leftrightarrow F$ domain wall in the $S_3$ quantum double is obtained from the $e \leftrightarrow m$ domain wall in the $\mathbb{Z}_3$ quantum double via gauging the charge conjugation symmetry.

is also an invertible $C \leftrightarrow F$ domain wall.

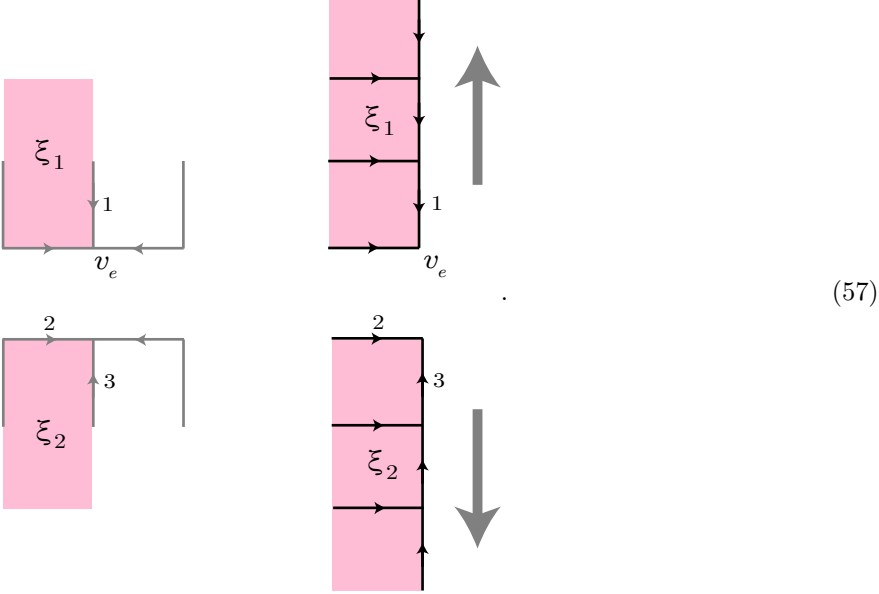

$$(57)$$

We remark that if we only gauge the $\mathbb{Z}_3$ subgroup of $S_3$ on the top and bottom bulks, then the bulk becomes an SET state, i.e., a $\mathbb{Z}_3$ toric code ground state with a $\mathbb{Z}_2$ charge conjugation symmetry. The domain wall becomes an invertible domain wall in this $\mathbb{Z}_3$ quantum double that gives rise to the following anyon map

$$m \to e, \quad e \to m^2. \tag{58}$$

The remaining $\mathbb{Z}_2$ symmetry acts on the qutrits as a charge conjugation. After gauging this charge conjugation symmetry, an $S_3$ quantum double ground state is obtained in the bulk, and the above domain wall in the symmetry enriched topological (SET) order becomes the $C \leftrightarrow F$ domain wall as shown in Fig. 6. This result agrees with Ref. [21, 84].

Based on this construction, we conclude this subsection with a conjecture.

**Conjecture 1.** *In a 2d quantum double (gauge theory) of a finite group $G$, all the invertible domain walls can be constructed from SPT-sewing with $G \times \mathrm{Rep}(G) \times G$ symmetry.*

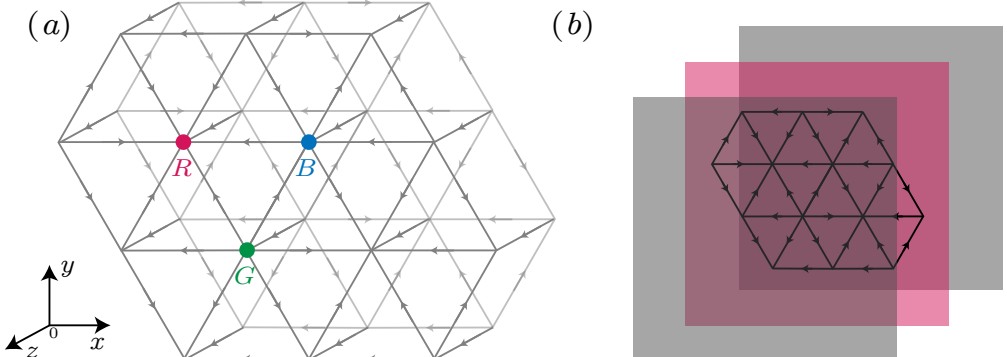

Figure 7. (a) Lattice structure used in the construction of the 3d models. Each triangular lattice is perpendicular to the $z$ direction; on each of them, there are three types of vertices, labeled $R$, $G$, and $B$, respectively. Each lattice perpendicular to the triangular lattices forms a square lattice. (b) Orientation of the domain wall (the pink layer).

When $G$ is Abelian, then the representation category $\mathrm{Rep}(G)$ is isomorphic to group $G$. In this case, all the invertible domain walls can be constructed through SPT-sewing with $G \times G \times G$ symmetry [Theorem 1]. Our result in this section provides a nontrivial evidence for this conjecture for non-Abelian groups. The proof of this conjecture for general $G$ is left for future work.

## VI.  EXOTIC DOMAIN WALLS IN THE 3D TORIC CODE

In three spatial dimensions, the SPT-sewn domain walls can exhibit exotic properties and host anyons which are nontrivially related to the excitations in the bulk. We obtain these domain walls by gauging a two-dimensional SPT embedded in a trivial product state in three dimensions.

This section discusses construction of these domain walls. This entails constructing certain 2d SPTs and gauging them (with respect to the 3d lattice). Note that there is a simple framework for constructing a fixed-point wavefunction of a 2d SPT on a triangular lattice [61, 89]. As such, it will be convenient to work in a slightly non-standard version of the 3d toric code that respects the triangular structure; see Fig. 7 for our convention. For construction of similar models on a cubic lattice, see Section VII.

Upon gauging a trivial product state in three dimensions, we obtain the standard 3d toric code vertex and plaquette terms in the bulk

$$
\begin{aligned}
H &= -\sum_{v \in V} A_v - \sum_{p \in F} B_p \\
&= -\sum_{v \in V} \prod_{v \in e} X_e - \sum_{p \in F} \prod_{e \in p} Z_e.
\end{aligned}
\tag{59}
$$

The ground states are the simultaneous $+1$-eigenstates of all the terms. If a quantum state $|\Psi\rangle$ is the $-1$-eigenstate of a certain vertex term, we say that $|\Psi\rangle$ has a point-like $e$ excitation on that vertex. The $m$ excitation, however, is a loop-like object in the dual lattice, with $B_p |\Psi\rangle = -|\Psi\rangle$ for each plaquette $p$ the loop passes through. A pair of point-like $e$ excitations can be created by applying an open string of Pauli $Z$ operators, whereas a loop-like $m$ excitation is created by a product of Pauli $X$ operators on a membrane bounded by the loop.

When we embed a two-dimensional SPT in a trivial product state, upon gauging, we always obtain the same bulk Hamiltonian away from the domain wall. However, what we obtain at the domain wall depends on the SPT we embed.

We consider three types of SPTs, which we refer to as type-I, type-II, and type-III. They correspond to the nontrivial classes of $\mathcal{H}^3(\mathbb{Z}_2 \times \mathbb{Z}_2 \times \mathbb{Z}_2, U(1))$ [61, 72, 90]. The corresponding cocycles (denoted as $\omega_I, \omega_{II}$,

and $\omega_{III}$) are given as follows

$$\omega_I = (-1)^{pa^{(i)}b^{(i)}c^{(i)}},$$
$$\omega_{II} = (-1)^{pa^{(i)}b^{(j)}c^{(j)}}, \tag{60}$$
$$\omega_{III} = (-1)^{pa^{(i)}b^{(j)}c^{(k)}},$$

where $p \in \{0,1\}$, $A = (a^{(1)}, a^{(2)}, a^{(3)})$, $B = (b^{(1)}, b^{(2)}, b^{(3)})$, $C = (c^{(1)}, c^{(2)}, c^{(3)})$, $a^{(i)}, b^{(j)}, c^{(k)} \in \{0,1\}$, and $(i)$ is the label for different copies.

Because the type-I SPT only involves a single copy of $\mathbb{Z}_2$, it is natural to place the SPT on one triangular lattice, sandwiched by other lattices. Then, the gauging method of Section III readily applies. However, for the type-II, it is more convenient to view it as a gapped boundary of two copies of the toric code, placing two qubits per site. Then, there is a natural $\mathbb{Z}_2 \times \mathbb{Z}_2$ symmetry we can gauge. Similarly, for the type-III case, we can place three qubits at each site and gauge the $\mathbb{Z}_2 \times \mathbb{Z}_2 \times \mathbb{Z}_2$ symmetry.

Before we describe our construction, we remark that the type-II and type-III domain walls are rather exotic. When a charge in the 3d bulk passes through these domain walls, it leaves a semi-loop-like excitation anchored on them. Therefore, we refer to these domain walls as *anchoring domain walls*. The anchoring domain walls can be classified further in terms of the anyon model at the domain wall and its interplay with the excitations in the bulk. The type-II domain wall hosts an Abelian anyon model, whereas the type-III hosts a non-Abelian anyon model.

The rest of this section is organized as follows. We review a method to construct a fixed-point SPT wavefunction in Section VI A, tailored to 2d. We then describe the domain walls of type-I, type-II, and type-III in Section VI B, VI C, and VI D, respectively.

## A. 2d SPT

Without loss of generality, consider one layer of a triangular lattice. Note that the vertices of this lattice can be partitioned into three sets, labeled $R$, $G$, and $B$, so that no neighboring vertices are in the same set; see the $z = 0$ plane in Fig. 7(a) for illustration and the orientation of the edges.

The SPT wavefunction can be obtained by applying the SPT entangler to the symmetric product state [61, 89]. This can be done by assigning a phase factor to each triangle in the following way

$$= \omega(g_3, g_3^{-1}g_2, g_2^{-1}g_1),$$

$$= \omega^{-1}(g_3, g_3^{-1}g_2, g_2^{-1}g_1), \tag{61}$$

where $\omega$ is the cocycle. The resulting wavefunction is

$$|\psi\rangle = \sum_{\{g_v\}} \prod_{\triangle_{123} \in F} \omega^{s(\triangle_{123})}(g_3, g_3^{-1}g_2, g_2^{-1}g_1) \bigotimes_{v \in V} |g_v\rangle, \tag{62}$$

where $g_v$ is the state at the site $v$, $\triangle_{123}$ is a 2-simplex (triangular face), and $s(\triangle_{123}) = \pm 1$ denotes the orientation of $\triangle_{123}$.

The phase factors in Eq. (61) define a constant-depth circuit diagonal in the standard basis. By conjugating the stabilizers of the symmetric product state by this circuit, we obtain the stabilizers of the SPT wavefunction.

## B. Type-I: double semion domain wall

Applying the procedure described in Section VI A to the nontrivial cocycle $\omega_I \in \mathcal{H}^3(\mathbb{Z}_2, U(1))$ in Eq. (60) we obtain

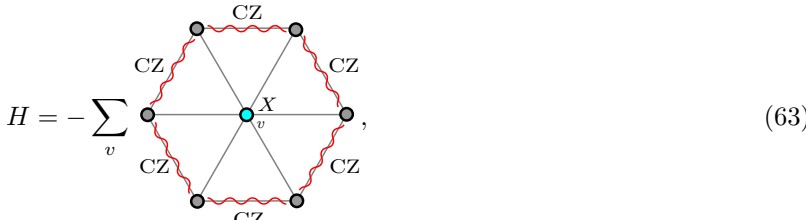

$$H = -\sum_v \tag{63}$$

where the red wavy lines represent CZ gates.

In the current setup, the Hamiltonian in Eq. (63) is defined on a triangular lattice, sandwiched between other layers of the triangular lattice; see Fig. 7. After gauging the global $\mathbb{Z}_2$ symmetry, we obtain the following Hamiltonian

$$H = -\sum_{v \in V} A_v W_v - \sum_{p \in F} B_p, \tag{64}$$

where $A_v$ and $B_p$ are the same set of operators as in Eq. (59). The operator $W_v$ is defined as

$$W_v = \tag{65}$$

where again the red wavy lines represent CZ gates.

If we were to remove the edges perpendicular to the plane in Eq. (64), our procedure would amount to gauging the Levin-Gu SPT [91], which gives rise to the double semion model; see Appendix F 1. In this model, there are four anyons: $1$, $s$, $\bar{s}$ and $s\bar{s}$. Their fusion rules are given by

$$s \times s = 1, \quad \bar{s} \times \bar{s} = 1, \quad s \times \bar{s} = s\bar{s}. \tag{66}$$

Their topological spins are

$$\theta(1) = 1, \ \theta(s) = i, \ \theta(\bar{s}) = -i, \ \theta(s\bar{s}) = 1. \tag{67}$$

The anyons $s$ and $\bar{s}$ are, respectively, a semion and an anti-semion, hence the name double semion.

Due to the edge in the perpendicular direction, there is a nontrivial interplay between the bulk excitations and the excitations at the domain wall. For instance, an $e$ particle entering the domain wall becomes an $s\bar{s}$ boson in the double semion model. This boson can also leave the domain wall and become an $e$ particle on the other side as well. An $m$ string can also pass through this domain wall, with a semion $s$ attached at the intersection of the $m$ string and the domain wall. The behavior of the excitations passing through the domain wall is shown in Fig. 8. For the construction of the operators that create excitations, see Appendix F 1.

The domain wall obtained here is similar to a gapped boundary known as the twisted smooth boundary of the 3d toric code [25, 92, 93]. In that model, an electric charge from the bulk becomes a composite of a semion $s$ and an anti-semion $\bar{s}$. Furthermore a semi-loop of magnetic flux can terminate on it, with the endpoints attached to a pair of semions or anti-semions. However, note that our construction yields a domain wall between two copies of the 3d toric code, not a gapped boundary.

## C. Type-II: anchoring domain wall

Now we consider the cocycle $\omega_{II} \in \mathcal{H}^3(\mathbb{Z}_2 \times \mathbb{Z}_2, U(1))$ in Eq. (60). Because this requires having two $\mathbb{Z}_2$ degrees of freedom, we assume that each site contains two qubits. After applying the corresponding SPT

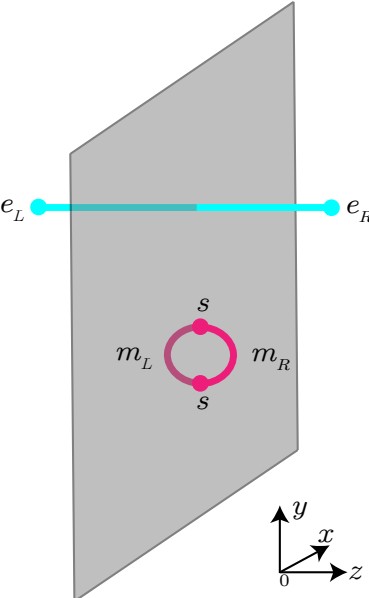

Figure 8. The excitations of the type-I domain wall.

entangler, we obtain the Hamiltonian of the SPT

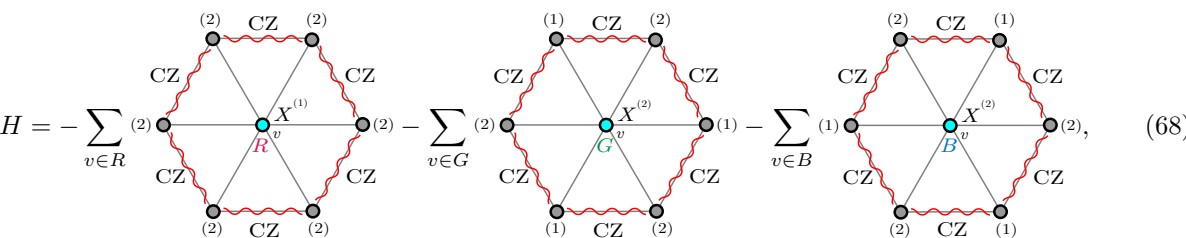

$$H = -\sum_{v \in R} (2) - \sum_{v \in G} (1) - \sum_{v \in B} (2), \qquad (68)$$

where ($i$) labels the qubits at each site and each sum is over the vertices from $R, G,$ and $B$, respectively; see Fig. 7(a). For calculation details, see Appendix F 2.

The Hamiltonian in Eq. (68) is embedded on a boundary of a 3d lattice in Fig. 7. After gauging we obtain a boundary for two copies of the 3d toric code. The resulting Hamiltonian is described in Appendix F 2. This boundary can be viewed alternatively as a domain wall between two 3d toric codes, which is the perspective we now take.

The anyon content of this domain wall can be inferred from a purely 2d model obtained by gauging Eq. (68). This is the twisted $\mathbb{Z}_2 \times \mathbb{Z}_2$ quantum double model, which is in the same phase as the $\mathbb{Z}_4$ quantum double [94]. The anyons for both models are listed in Table II and Table III, and there is a one-to-one correspondence between them. There are 16 types of anyons, generated by $e^{(1,0)}$, $e^{(0,1)}$, $m^{(1,0)}$, and $m^{(0,1)}$, where the superscripts represent layers the anyons belong to.

| 1 | $m^{(1,0)}$ | $e^{(0,1)}$ | $m^{(1,0)}e^{(0,1)}$ |
|---|---|---|---|
| $m^{(0,1)}$ | $m^{(1,1)}$ | $m^{(0,1)}e^{(0,1)}$ | $m^{(1,1)}e^{(0,1)}$ |
| $e^{(1,0)}$ | $m^{(1,0)}e^{(1,0)}$ | $e^{(1,1)}$ | $m^{(1,0)}e^{(1,1)}$ |
| $m^{(0,1)}e^{(1,0)}$ | $m^{(1,1)}e^{(1,0)}$ | $m^{(0,1)}e^{(1,1)}$ | $m^{(1,1)}e^{(1,1)}$ |

Table II. Anyon table of the $\mathbb{Z}_2 \times \mathbb{Z}_2$ twisted quantum double.

To summarize, there are the following anyons.

- Eight bosons: 1, $m^{(1,0)}$, $m^{(0,1)}$, $e^{(1,0)}$, $e^{(0,1)}$, $e^{(1,1)}$, $m^{(1,0)}e^{(0,1)}$, and $m^{(0,1)}e^{(1,0)}$.

- Four fermions: $m^{(0,1)}e^{(0,1)}$, $m^{(1,0)}e^{(1,0)}$, $m^{(1,0)}e^{(1,1)}$, $m^{(0,1)}e^{(1,1)}$.

| 1 | $\widetilde{e}$ | $\widetilde{e}^2$ | $\widetilde{e}^3$ |
|---|---|---|---|
| $\widetilde{m}$ | $\widetilde{m}\widetilde{e}$ | $\widetilde{m}\widetilde{e}^2$ | $\widetilde{m}\widetilde{e}^3$ |
| $\widetilde{m}^2$ | $\widetilde{m}^2\widetilde{e}$ | $\widetilde{m}^2\widetilde{e}^2$ | $\widetilde{m}^2\widetilde{e}^3$ |
| $\widetilde{m}^3$ | $\widetilde{m}^3\widetilde{e}$ | $\widetilde{m}^3\widetilde{e}^2$ | $\widetilde{m}^3\widetilde{e}^3$ |

Table III. Anyon table of the $\mathbb{Z}_4$ quantum double.

- Two semions: $m^{(1,1)}$, $m^{(1,1)}e^{(1,1)}$.

- Two anti-semions: $m^{(1,1)}e^{(0,1)}$, $m^{(1,1)}e^{(1,0)}$.

These anyons have a nontrivial interplay with the bulk excitations, as we explain below; see also Fig. 9. First of all, an $e$ particle from the right bulk, upon entering the domain wall, becomes an $e^{(0,1)}$ anyon on the domain wall. This anyon can be split into a pair of $m^{(1,0)}$ and $m^{(1,0)}e^{(0,1)}$, which are attached to a semi-loop excitation anchored on the opposite side of the domain wall. Thus the $e$ particle from the right bulk gets transformed to a semi-loop anchored on the domain wall on the opposite side. Similarly, an $e$-particle from the left bulk can get transformed to a semi-loop anchored on the domain wall on the opposite side. However, the domain wall anyon attached to the semi-loop is different. Instead of $m^{(1,0)}$, the $m^{(0,1)}$ anyons are attached. For the operators that create the excitations, see Appendix F 2.

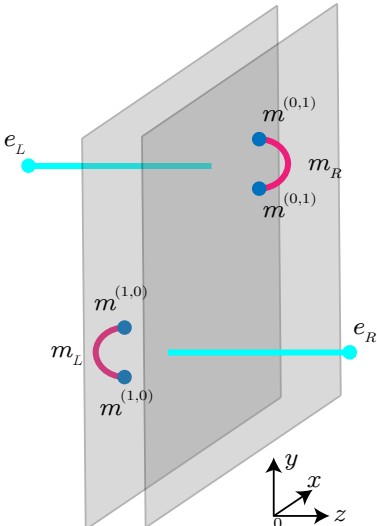

Figure 9. Excitations of the anchoring domain wall (type-II).

Thus, unlike the domain wall in Section VI B, in this domain wall the bulk excitations change their types after passing through. To the best of our knowledge, such a domain wall has not been discussed before. This domain wall generalizes the $e \leftrightarrow m$ domain wall in the 2d toric code and is beyond the classification of the Lagrangian subgroup formalism. Potentially, it could be classified by the *condensable algebra* [17, 21, 24–26].

### D.  Type-III: non-Abelian anchoring domain wall

Now we consider the cocycle $\omega_{III} \in \mathcal{H}^3(\mathbb{Z}_2 \times \mathbb{Z}_2 \times \mathbb{Z}_2, U(1))$ in Eq. (60). In this setup, on each site, we introduce three $\mathbb{Z}_2$ degrees of freedom. After applying the SPT entangler, we obtain

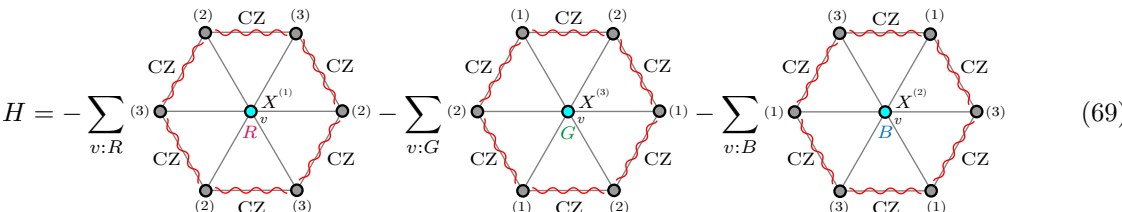

where $(i)$ labels the qubits at each site.

After gauging Eq. (69), we obtain the following

$$H = -\sum_{v \in V} \sum_{g \in G} A_v^g W_v^g - \sum_{p \in F} B_p, \tag{70}$$

where $g = (a, b, c) \in G = \mathbb{Z}_2^3$ and $A_v^g$ is defined as

$$A_v^g = \left(A_v^{(1)}\right)^a \left(A_v^{(2)}\right)^b \left(A_v^{(3)}\right)^c. \tag{71}$$

The $B_p$ term is the same as in Eq. (59), on all three copies. The additional term $W_v^g$ is defined as follows



where $a, b, c \in \{0, 1\}$. For calculation details, see Appendix F 3.

Similar to the other domain walls discussed so far, the anyon content of the domain wall can be inferred by gauging Eq. (69) on a plane. This becomes the $\mathbb{Z}_2^3$ twisted quantum double, which has been widely studied recently in the context of gauged-SPT defects, exotic boundaries, and fault-tolerant quantum computation [49, 95–98]. The $\mathbb{Z}_2^3$ twisted quantum double is equivalent to the $D_4$ quantum double [72, 94], a non-Abelian anyon model. Thus the SPT-sewn domain wall can host non-Abelian anyons. The excitations can be labeled by the magnetic fluxes, $A \in \mathbb{Z}_2^3$, and the associated projective representation corresponding to a 2-cocycle $c_A$, which is given by the slant product of the 3-cocycle

$$c_A(B, C) := i_A \omega(B, C) = \frac{\omega_{III}(A, B, C) \omega_{III}(B, C, A)}{\omega_{III}(B, A, C)}. \tag{73}$$

The anyons are summarized as follows [72].

- $A = (0, 0, 0)$. There are 8 anyons in this class: 1 vacuum and 7 electric charges. We label them as 1, $e^{(100)}$, $e^{(010)}$, $e^{(001)}$, $e^{(110)}$, $e^{(011)}$, $e^{(101)}$ and $e^{(111)}$, where $e^{(abc)} \times e^{(a'b'c')} = e^{(a+a'b+b'c+c')}$.

- $A = (1, 0, 0)$. There are two two-dimensional irreducible representations for $c_A$, one with bosonic and the other with fermionic self-statistics. Therefore, we label them as $m^{(100)}$ and $f^{(100)}$. Similarly, we have $m^{(010)}$ and $f^{(010)}$ when $A = (0, 1, 0)$, and $m^{(001)}$ and $f^{(001)}$ when $A = (0, 0, 1)$.

- $A = (1, 1, 0)$. Similarly to the previous case, there are two two-dimensional irreducible representations for $c_A$, one with bosonic and the other with fermionic self-statistics. We use the same nomenclature to label the anyons in this class as $m^{(110)}$, $m^{(011)}$, $m^{(101)}$, $f^{(110)}$, $f^{(011)}$, and $f^{(101)}$.

- $A = (1, 1, 1)$. In this case, there are two two-dimensional irreducible representations for $c_A$, one with semionic and the other with anti-semionic self-statistics. Therefore, we label them as $s$ and $\bar{s}$.

In summary, there are 22 different types of anyons in $\mathbb{Z}_2^3$ twisted quantum double, including 1 vacuum, 7 electric charges, and 14 dyons.

Similar to the domain wall in Section VI C, this is also an anchoring domain wall. However, an important difference is that this domain wall hosts non-Abelian anyons. For instance, at the intersection point of semi-loop (say, over the first copy) and the domain wall we get the $m^{(100)}$ anyon. These can be fused into electric charges or vacuum, which can then propagate into the bulk as an electric charge. Thus we see that the semi-loop can be transformed into electric charge on the two remaining copies, i.e.,

$$m_R \times m_R = 1 + e_{L1} + e_{L2} + e_{L1}e_{L2}. \tag{74}$$

Here we are considering a setup in which there is one copy of the 3d toric code on the right side of the domain wall and two copies of the 3d toric code on the left side. Then $m_R$ is the endpoint of the semi-loop on the right side and $e_{L1}$ and $e_{L2}$ are the electric charges on the two copies on the left side; see Fig. 10(a).

We can also change the fusion channel by braiding the domain wall magnetic anyons [44, 72]. For instance, we can consider the setup displayed in Fig. 10(b). We first create a magnetic semi-loop for each copy on the left bulk, resulting a pair of $m^{(100)}$ anyons and a pair of $m^{(010)}$ anyons on the domain wall. By braiding one $m^{(100)}$ and one $m^{(010)}$, an $e^{(001)}$ particles are created on the domain wall, which can propagate to the bulk. This happens because braiding two magnetic fluxes of different types generates an electric charge of the third type at the intersection of the path [79]. For the operators that create excitations, see Appendix F 3.

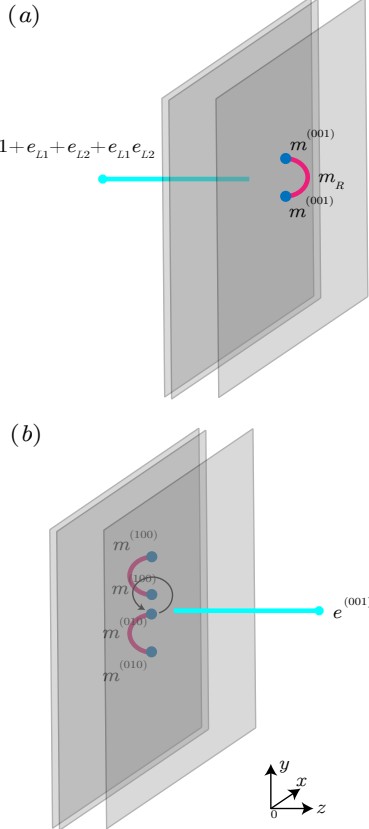

Figure 10. Excitations of the anchoring domain wall (type-III). (a) A semi-loop on one side and a direct sum of electric charges from the other side can condense on this domain wall. (b) Two semi-loops are attached to the domain wall. By braiding one of the $m^{(010)}$ with a $m^{(100)}$, one can obtain a $e^{(001)}$ charge on the other side.

## VII. SIMPLE MODELS OF THE EXOTIC DOMAIN WALLS

In this section, we describe simplified models that produce the same physics as the ones described in Sections VI B and VI C. There are two advantages of these simplified models. First, they are defined on a cubic lattice, similar to the well-known 3d toric code model. Second, they are (generalized) Pauli models [99], which make them easy to study.

We consider a cubic lattice throughout this section. In the bulk, each edge hosts one qubit, and the 3d toric code model is given as in Eq. (59),

$$H = -\sum_{v \in V} A_v - \sum_{p \in F} B_p, \tag{75}$$

where $A_v$ and $B_p$ are given by

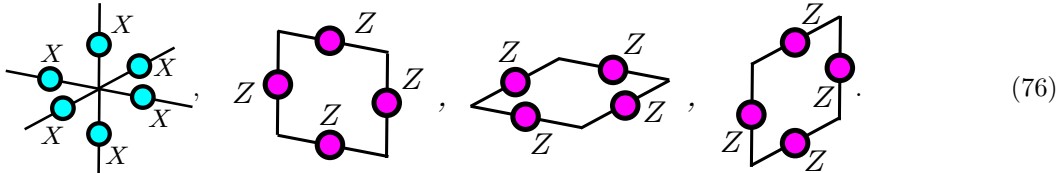

$$\tag{76}$$

### A. Type-I domain wall

We now describe a simple model of the type-I domain wall. Away from the domain wall, each edge hosts a single qubit. For the edges on the domain wall, we introduce two qubits per site. These two qubits can be viewed as a four-dimensional qudit. The set of operators acting on this qudit is generated by the generalized Pauli $\widetilde{X}$ and $\widetilde{Z}$ operators

$$\widetilde{X} = \sum_{h=0}^{3} |h+1 \mod 4\rangle\langle h|, \ \ \widetilde{Z} = \sum_{h=0}^{3} i^h |h\rangle\langle h|. \tag{77}$$

Because the stabilizers away from the domain wall are the same as that of the 3d toric code, we focus on the stabilizers at the domain wall. The domain wall stabilizers are given as follows

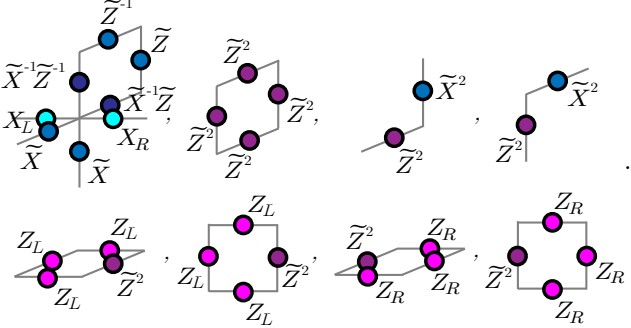

The domain wall contains the plaquette depicted as the second term and its translations in the $x$ and $y$ directions. Here $L$ and $R$ represent the qubits lying to the left and the right of the domain wall.

On the domain wall there are four anyons: $1$, $s$, $\bar{s}$ and $s\bar{s}$. The fusion rules are given by

$$s \times s = 1, \quad \bar{s} \times \bar{s} = 1, \quad s \times \bar{s} = s\bar{s}. \tag{78}$$

Their topological spins are

$$\theta(1) = 1, \ \theta(s) = i, \ \theta(\bar{s}) = -i, \ \theta(s\bar{s}) = 1. \tag{79}$$

The anyon $s$ is a semion, while $\bar{s}$ is an anti-semion.

On the domain wall, pairs of $s$, $\bar{s}$, and $s\bar{s}$ are created by the following operators[2]

$$W^s = \begin{array}{c} \widetilde{Z} \\ \widetilde{X} \end{array} , W^{\bar{s}} = \begin{array}{c} \widetilde{Z} \\ \widetilde{X}^{-1} \end{array} , W^{s\bar{s}} = \begin{array}{c} \\ \widetilde{Z}^2 \end{array}. \tag{80}$$

Note that when a pair of $s$ or $\bar{s}$ anyons is created, two semi-loops of magnetic flux are also generated in the bulk on both sides of the domain wall. A pair of $s\bar{s}$ anyons can be created solely on the domain wall. However, by applying a string of Pauli $Z$ operators, one can move an $s\bar{s}$ anyon into the bulk, where it becomes an electric charge.

In terms of the bulk excitations, the above description can be rephrased as follows. An $e$ particle entering the domain wall becomes an $s\bar{s}$ boson in the double semion model. This boson can further leave the domain wall and become an $e$ particle on the other side. Therefore, $e$ can pass through this domain wall unchanged. Similarly, an $m$ string can also pass through this domain wall, with a semion $s$ anchored to the intersection of the $m$ string and the domain wall.

### B.  SPT-sewn domain wall: $\mathbb{Z}_4$ quantum double

Now we introduce a model of the type-II domain wall. The stabilizers of this model are shown below:

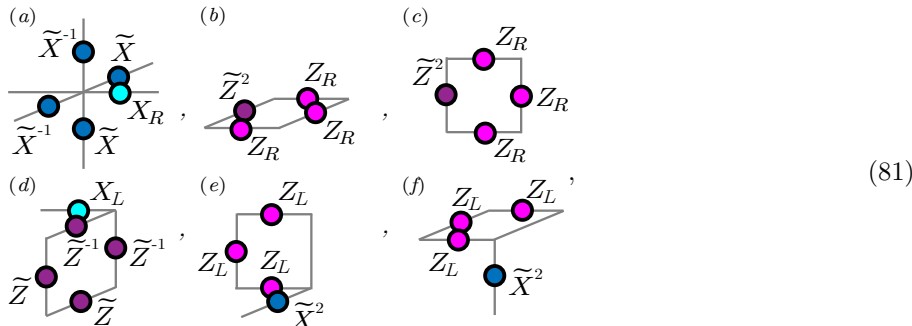

$$\tag{81}$$

following the convention used in Section VII A.

On the domain wall, the excitations are the anyons of the $\mathbb{Z}_4$ quantum double. The set of anyons is generated by $\{\tilde{e}, \tilde{m}\}$. Both of them have bosonic statistics. They have bosonic self-statistics

$$\theta(\tilde{e}) = \theta(\tilde{m}) = 1. \tag{82}$$

The combinations of $\tilde{e}$ and $\tilde{m}$ can have semionic or fermionic self-statistics, namely

$$\theta(\tilde{m}\tilde{e}) = i, \quad \theta(\tilde{m}^2\tilde{e}) = -1. \tag{83}$$

By applying the following operator on the domain wall

$$W^{\tilde{e}^2} = \begin{array}{c} \widetilde{Z}^2 \\ \bullet \end{array}, \tag{84}$$

the neighboring stabilizers in Eq. (81)(a) are violated, creating a pair of $\tilde{e}^2$ anyons on the corresponding vertices. By further applying Pauli $Z$ operators on the connected edges in the right bulk, this pair of $\tilde{e}^2$ anyons moves into the bulk and becomes a pair of $e$ charges. We conclude that an $e$-particle from the right bulk corresponds to a $\tilde{e}^2$ anyon on the domain wall.

---

[2] Here we only consider creating anyon pairs in a specific direction. For the full discussion we refer the reader to Ref. [99].

Similarly, by applying the following operator on the domain wall

$$W^{\tilde{m}^2} = \begin{array}{c}\bullet\end{array} \widetilde{X}^2, \tag{85}$$

the neighboring stabilizers in Eq. (81)(d) are violated, creating a pair of $\tilde{m}^2$ anyons on the corresponding faces. By further applying Pauli $Z$ operators on the connected edges in the left bulk, this pair of $\tilde{m}^2$ anyons moves into the bulk and becomes a pair of $e$ charges. We conclude that an $e$ particle from the left bulk corresponds to an $\tilde{m}^2$ anyon on the domain wall.

Now, let us consider the following scenario. By applying the following operator on the domain wall

$$W^{\tilde{e}} = \begin{array}{c}\widetilde{Z}\\\bullet\end{array}, \tag{86}$$

a pair of $\tilde{e}$ and $\tilde{e}^3$ anyons is created on the domain wall. As we discussed before, an $\tilde{e}^2$ anyon on the domain wall can become an $e$ charge in the right bulk. Therefore, we can apply Pauli $Z$ strings to move an $\tilde{e}^2$ anyon into the bulk, leaving a pair of $\tilde{e}$ anyons on the domain wall. Furthermore, the operator in Eq. (86) violates the stabilizer in Eq. (81)(e). By further applying a Pauli $X$ operator in the left bulk, a semi-loop of magnetic flux is generated. Therefore, the resulting configuration is a semi-loop of magnetic flux anchored on the domain wall, with the endpoints attached to a pair of $\tilde{e}$ anyons, and an electric charge emerging on the other side.

Similar scenario happens when we apply the operator $W^{\tilde{m}}$ on the domain wall

$$W^{\tilde{m}} = \begin{array}{c}\bullet\end{array} \widetilde{X}. \tag{87}$$

After applying this operator, a pair of $\tilde{m}$ and $\tilde{m}^3$ anyons are created on the domain wall. As we discussed before, an $\tilde{m}^2$ anyon on the domain wall can become an $e$ charge in the left bulk. Therefore, we can apply Pauli $Z$ strings to move an $\tilde{m}^2$ anyon into the bulk, leaving a pair of $\tilde{m}$ anyons on the domain wall. Furthermore, the operator in Eq. (87) violates the stabilizer in Eq. (81)(c). A semi-loop of magnetic flux is thus generated in the right bulk. The nontrivial properties of this domain wall are depicted in Fig. 11.

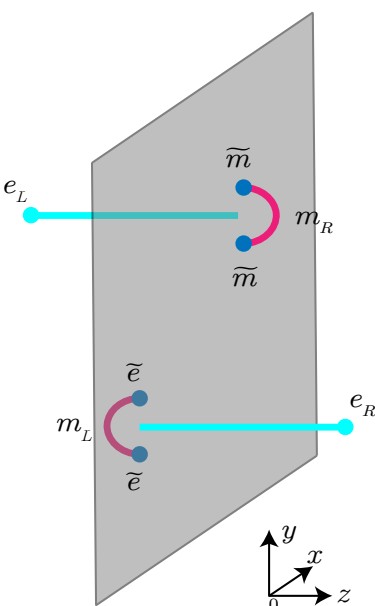

Figure 11. Excitations of the anchoring domain wall in terms of the anyons of the $\mathbb{Z}_4$ quantum double.

## VIII. DISCUSSION

In this article, we have introduced *SPT-sewing*, which is a systematic method of constructing gapped domain walls of topologically ordered systems. Compared to the prior works [48, 49], where gauging was used to create lower-dimensional defects from defects in a simpler topologically ordered (or trivial) phase, SPT-sewing uses a lower-dimensional SPT to entangle two disconnected topological orders.

Using SPT-sewing, we proved that all the invertible domain walls in the 2d Abelian $G$ quantum double model can be constructed from $G \times G \times G$ SPT-sewing, and illustrated the discussion with several examples. We conjecture that all the invertible domain walls in a generic 2d quantum double can be constructed from SPT-sewing with a non-invertible $G \times \text{Rep}(G) \times G$ symmetry. Our investigation of the $S_3$ quantum double model provides a nontrivial evidence of this conjecture.

We also used our method of SPT-sewing to construct *anchoring domain walls*, which are exotic domain walls in the 3d toric code. We provided the domain wall lattice models constructed by sewing the canonical 2d SPT states on a triangular lattice given by the 3-cocycles, as well as the equivalent but simpler domain wall lattice models on a cubic lattice. These domain walls transform point-like excitations into semi-loop-like excitations anchored on them. As such, anchoring domain walls appear to generalize the $e \leftrightarrow m$ exchange domain wall [6] to higher dimensions.

There are several future directions to pursue. First, it is natural to expect that SPT-sewing will lead to efficient methods of creating gapped domain walls in topologically ordered lattice models. Since the gauging map for solvable groups can be implemented via constant-depth adaptive circuits [28–31], we anticipate the domain walls for Abelian models, the $C \leftrightarrow F$ domain wall and the exotic domain walls in 3d that we described in our work to be also implementable via constant-depth adaptive circuits. Second, it is natural to ask whether there is a field-theoretical description of the SPT-sewing construction and the models we discovered. To the best of our knowledge, the anchoring domain walls in the 3d toric code have not been studied before and they appear not to fit into the existing categorical classification of lower-dimensional defects in Ref. [24].[3] We leave their classifications and extensions to higher dimensions for future work. Third, it would be interesting to understand if the anchoring domain walls can be used to design a self-correcting quantum memory [100]. The anchoring domain walls transform point-like excitations to semi-loop-like excitations, thereby inhibiting their propagation. However, it remains unclear how to construct a model, such as one based on arranging these domain walls into a defect network [101–103], that employs the anchoring domain wall to produce a macroscopic energy barrier for self-correction.

## ACKNOWLEDGMENTS

ZS thanks helpful discussions with Yuanjie Ren and Rohith Sajith. We thank Liang Kong for helpful discussions. This work was supported by the U.S. National Science Foundation under Grant No. NSF DMR-2316598 (YL). ZS and IK were supported by funds from the UC Multicampus Research Programs and Initiatives of the University of California, Grant Number M23PL5936.

[1] A. Y. Kitaev, Fault-tolerant quantum computation by anyons, Annals of physics **303**, 2 (2003).

[2] M. A. Levin and X.-G. Wen, String-net condensation: A physical mechanism for topological phases, Phys. Rev. B **71**, 045110 (2005).

[3] K. Walker and Z. Wang, (3+ 1)-tqfts and topological insulators, Frontiers of Physics **7**, 150 (2012).

[4] A. Kitaev, Anyons in an exactly solved model and beyond, Annals of Physics **321**, 2 (2006).

[5] I. H. Kim and D. Ranard, Classifying 2d topological phases: mapping ground states to string-nets (2024), arXiv:2405.17379 [quant-ph].

[6] H. Bombin, Topological order with a twist: Ising anyons from an abelian model, Phys. Rev. Lett. **105**, 030403 (2010).

[7] S. Beigi, P. W. Shor, and D. Whalen, The quantum double model with boundary: condensations and symmetries, Communications in mathematical physics **306**, 663 (2011).

---

[3] We thank Liang Kong for the related discussion.

[8] A. Kapustin and N. Saulina, Topological boundary conditions in abelian chern–simons theory, Nuclear Physics B **845**, 393 (2011).

[9] A. Kitaev and L. Kong, Models for gapped boundaries and domain walls, Communications in Mathematical Physics **313**, 351 (2012).

[10] M. Barkeshli, C.-M. Jian, and X.-L. Qi, Classification of topological defects in abelian topological states, Phys. Rev. B **88**, 241103 (2013).

[11] M. Barkeshli, C.-M. Jian, and X.-L. Qi, Theory of defects in abelian topological states, Phys. Rev. B **88**, 235103 (2013).

[12] S. B. Bravyi and A. Y. Kitaev, Quantum codes on a lattice with boundary, arXiv preprint quant-ph/9811052 (1998).

[13] V. Ostrik, Module categories over the drinfeld double of a finite group, arXiv preprint math/0202130 (2002).

[14] A. Davydov, Modular invariants for group-theoretical modular data. i, Journal of Algebra **323**, 1321 (2010).

[15] A. Davydov, D. Nikshych, and V. Ostrik, On the structure of the witt group of braided fusion categories, Selecta Mathematica **19**, 237 (2013).

[16] M. Levin, Protected edge modes without symmetry, Physical Review X **3**, 021009 (2013).

[17] L. Kong, Anyon condensation and tensor categories, Nuclear Physics B **886**, 436 (2014).

[18] T. Lan, J. C. Wang, and X.-G. Wen, Gapped domain walls, gapped boundaries, and topological degeneracy, Physical review letters **114**, 076402 (2015).

[19] J. Fuchs, C. Schweigert, and A. Valentino, A geometric approach to boundaries and surface defects in dijkgraaf–witten theories, Communications in Mathematical Physics **332**, 981 (2014).

[20] P. Huston, F. Burnell, C. Jones, and D. Penneys, Composing topological domain walls and anyon mobility, SciPost Physics **15**, 076 (2023).

[21] R. Xu and H. Yang, 2-morita equivalent condensable algebras in topological orders, arXiv preprint arXiv:2403.19779 (2024).

[22] I. Cong, M. Cheng, and Z. Wang, Hamiltonian and algebraic theories of gapped boundaries in topological phases of matter, Communications in Mathematical Physics **355**, 645 (2017).

[23] L. Kong and Z.-H. Zhang, An invitation to topological orders and category theory, arXiv preprint arXiv:2205.05565 (2022).

[24] L. Kong, Z.-H. Zhang, J. Zhao, and H. Zheng, Higher condensation theory, arXiv preprint arXiv:2403.07813 (2024).

[25] J. Zhao, J.-Q. Lou, Z.-H. Zhang, L.-Y. Hung, L. Kong, and Y. Tian, String condensations in 3+ 1d and lagrangian algebras, arXiv preprint arXiv:2208.07865 (2022).

[26] L. Kong, Y. Tian, and Z.-H. Zhang, Defects in the 3-dimensional toric code model form a braided fusion 2-category, Journal of High Energy Physics **2020**, 1 (2020).

[27] B. Shi and I. H. Kim, Entanglement bootstrap approach for gapped domain walls, Phys. Rev. B **103**, 115150 (2021).

[28] N. Tantivasadakarn, R. Thorngren, A. Vishwanath, and R. Verresen, Long-range entanglement from measuring symmetry-protected topological phases, Phys. Rev. X **14**, 021040 (2024).

[29] R. Verresen, N. Tantivasadakarn, and A. Vishwanath, Efficiently preparing schrödinger's cat, fractons and non-abelian topological order in quantum devices (2022), arXiv:2112.03061 [quant-ph].

[30] S. Bravyi, I. Kim, A. Kliesch, and R. Koenig, Adaptive constant-depth circuits for manipulating non-abelian anyons (2022), arXiv:2205.01933 [quant-ph].

[31] T.-C. Lu, L. A. Lessa, I. H. Kim, and T. H. Hsieh, Measurement as a shortcut to long-range entangled quantum matter, PRX Quantum **3**, 040337 (2022).

[32] N. Tantivasadakarn, R. Verresen, and A. Vishwanath, Shortest route to non-abelian topological order on a quantum processor, Phys. Rev. Lett. **131**, 060405 (2023).

[33] N. Tantivasadakarn, A. Vishwanath, and R. Verresen, Hierarchy of topological order from finite-depth unitaries, measurement, and feedforward, PRX Quantum **4**, 020339 (2023).

[34] Y. Li, H. Sukeno, A. P. Mana, H. P. Nautrup, and T.-C. Wei, Symmetry-enriched topological order from partially gauging symmetry-protected topologically ordered states assisted by measurements, Physical Review B **108**, 115144 (2023).

[35] Y. Li, M. Litvinov, and T.-C. Wei, Measuring topological field theories: Lattice models and field-theoretic description, arXiv preprint arXiv:2310.17740 (2023).

[36] H. Sukeno and T.-C. Wei, Quantum simulation of lattice gauge theories via deterministic duality transformations assisted by measurements, Phys. Rev. A **109**, 042611 (2024).

[37] A. Lyons, C. F. B. Lo, N. Tantivasadakarn, A. Vishwanath, and R. Verresen, Protocols for creating anyons and defects via gauging (2024), arXiv:2411.04181 [quant-ph].

[38] B. Yoshida, Topological phases with generalized global symmetries, Physical Review B **93**, 155131 (2016).

[39] S. Vijay, J. Haah, and L. Fu, Fracton topological order, generalized lattice gauge theory, and duality, Physical Review B **94**, 235157 (2016).

[40] D. J. Williamson, Fractal symmetries: Ungauging the cubic code, Physical Review B **94**, 155128 (2016).

[41] A. Kubica and B. Yoshida, Ungauging quantum error-correcting codes (2018), arXiv:1805.01836 [quant-ph].

[42] W. Shirley, K. Slagle, and X. Chen, Foliated fracton order from gauging subsystem symmetries, SciPost Physics **6**, 041 (2019).

[43] T. Rakovszky and V. Khemani, The physics of (good) ldpc codes i. gauging and dualities (2023), arXiv:2310.16032 [quant-ph].

[44] M. Iqbal, N. Tantivasadakarn, R. Verresen, S. L. Campbell, J. M. Dreiling, C. Figgatt, J. P. Gaebler, J. Johansen, M. Mills, S. A. Moses, *et al.*, Non-abelian topological order and anyons on a trapped-ion processor, Nature **626**, 505 (2024).

[45] M. Foss-Feig, A. Tikku, T.-C. Lu, K. Mayer, M. Iqbal, T. M. Gatterman, J. A. Gerber, K. Gilmore, D. Gresh, A. Hankin, *et al.*, Experimental demonstration of the advantage of adaptive quantum circuits, arXiv preprint arXiv:2302.03029 (2023).

[46] Y. Song, L. Beltrán, I. Besedin, M. Kerschbaum, M. Pechal, F. Swiadek, C. Hellings, D. C. Zanuz, A. Flasby, J.-C. Besse, and A. Wallraff, Realization of constant-depth fan-out with real-time feedforward on a superconducting quantum processor (2024), arXiv:2409.06989 [quant-ph].

[47] M. Iqbal, A. Lyons, C. F. B. Lo, N. Tantivasadakarn, J. Dreiling, C. Foltz, T. M. Gatterman, D. Gresh, N. Hewitt, C. A. Holliman, J. Johansen, B. Neyenhuis, Y. Matsuoka, M. Mills, S. A. Moses, P. Siegfried, A. Vishwanath, R. Verresen, and H. Dreyer, Qutrit toric code and parafermions in trapped ions (2024), arXiv:2411.04185 [quant-ph].

[48] Y. Ren, N. Tantivasadakarn, and D. J. Williamson, Efficient preparation of solvable anyons with adaptive quantum circuits (2024), arXiv:2411.04985 [quant-ph].

[49] M. Barkeshli, Y.-A. Chen, S.-J. Huang, R. Kobayashi, N. Tantivasadakarn, and G. Zhu, Codimension-2 defects and higher symmetries in (3+1)D topological phases, SciPost Phys. **14**, 065 (2023).

[50] R. Thorngren and Y. Wang, Fusion category symmetry i: anomaly in-flow and gapped phases, arXiv preprint arXiv:1912.02817 (2019).

[51] K. Inamura, Topological field theories and symmetry protected topological phases with fusion category symmetries, Journal of High Energy Physics **2021**, 1 (2021).

[52] J. McGreevy, Generalized symmetries in condensed matter, Annual Review of Condensed Matter Physics **14**, 57 (2023).

[53] S. Schäfer-Nameki, Ictp lectures on (non-) invertible generalized symmetries, Physics Reports **1063**, 1 (2024).

[54] S.-H. Shao, What's done cannot be undone: Tasi lectures on non-invertible symmetry, arXiv preprint arXiv:2308.00747 (2023).

[55] C. Fechisin, N. Tantivasadakarn, and V. V. Albert, Non-invertible symmetry-protected topological order in a group-based cluster state, arXiv preprint arXiv:2312.09272 (2023).

[56] S. Seifnashri and S.-H. Shao, Cluster state as a noninvertible symmetry-protected topological phase, Physical Review Letters **133**, 116601 (2024).

[57] Z. Jia, Generalized cluster states from Hopf algebras: non-invertible symmetry and Hopf tensor network representation, JHEP **09**, 147, arXiv:2405.09277 [quant-ph].

[58] L. Bhardwaj, L. E. Bottini, S. Schafer-Nameki, and A. Tiwari, Lattice models for phases and transitions with non-invertible symmetries, arXiv preprint arXiv:2405.05964 (2024).

[59] Y. Li and M. Litvinov, Non-invertible spt, gauging and symmetry fractionalization, arXiv preprint arXiv:2405.15951 (2024).

[60] K. Kawagoe, C. Jones, S. Sanford, D. Green, and D. Penneys, Levin-wen is a gauge theory: entanglement from topology (2024), arXiv:2401.13838 [cond-mat.str-el].

[61] B. Yoshida, Gapped boundaries, group cohomology and fault-tolerant logical gates, Annals of Physics **377**, 387 (2017).

[62] H. Bombin and M. Martin-Delgado, Family of non-abelian kitaev models on a lattice: Topological condensation and confinement, Physical Review B—Condensed Matter and Materials Physics **78**, 115421 (2008).

[63] K. Roumpedakis, S. Seifnashri, and S.-H. Shao, Higher gauging and non-invertible condensation defects, Communications in Mathematical Physics **401**, 3043 (2023).

[64] N. Tantivasadakarn and X. Chen, String operators for cheshire strings in topological phases, arXiv preprint arXiv:2307.03180 (2023).

[65] R. Raussendorf, D. E. Browne, and H. J. Briegel, Measurement-based quantum computation on cluster states, Phys. Rev. A **68**, 022312 (2003).

[66] J. C. Wang, Z.-C. Gu, and X.-G. Wen, Field-theory representation of gauge-gravity symmetry-protected topological invariants, group cohomology, and beyond, Physical review letters **114**, 031601 (2015).

[67] L. Tsui and X.-G. Wen, Lattice models that realize $\mathbb{Z}_{n-1}$ symmetry-protected topological states for even $n$, Physical Review B **101**, 035101 (2020).

[68] Y.-A. Chen and S. Tata, Higher cup products on hypercubic lattices: application to lattice models of topological phases, Journal of Mathematical Physics **64** (2023).

[69] X. Chen, Y.-M. Lu, and A. Vishwanath, Symmetry-protected topological phases from decorated domain walls, Nature communications **5**, 3507 (2014).

[70] M. Barkeshli, Y.-A. Chen, P.-S. Hsin, and R. Kobayashi, Higher-group symmetry in finite gauge theory and stabilizer codes, SciPost Physics **16**, 089 (2024).

[71] M. Barkeshli, P.-S. Hsin, and R. Kobayashi, Higher-group symmetry of (3+1)d fermionic $\mathbb{Z}_2$ gauge theory: Logical ccz, cs, and t gates from higher symmetry, SciPost Phys. **16**, 122 (2024).

[72] M. d. W. Propitius, Topological interactions in broken gauge theories, arXiv preprint hep-th/9511195 (1995).

[73] M. Barkeshli, P. Bonderson, M. Cheng, and Z. Wang, Symmetry fractionalization, defects, and gauging of topological phases, Physical Review B **100**, 115147 (2019).

[74] S. Bravyi and D. Maslov, Hadamard-free circuits expose the structure of the clifford group, IEEE Transactions on Information Theory **67**, 4546 (2021).

[75] H. Bombin and M. A. Martin-Delgado, Topological quantum distillation, Physical review letters **97**, 180501 (2006).

[76] H. Bombin and M. Martin-Delgado, Exact topological quantum order in d= 3 and beyond: Branyons and brane-net condensates, Physical Review B—Condensed Matter and Materials Physics **75**, 075103 (2007).

[77] A. Kubica, B. Yoshida, and F. Pastawski, Unfolding the color code, New Journal of Physics **17**, 083026 (2015).

[78] M. S. Kesselring, J. C. Magdalena de la Fuente, F. Thomsen, J. Eisert, S. D. Bartlett, and B. J. Brown, Anyon condensation and the color code, PRX Quantum **5**, 010342 (2024).

[79] M. Davydova, N. Tantivasadakarn, S. Balasubramanian, and D. Aasen, Quantum computation from dynamic automorphism codes, Quantum **8**, 1448 (2024).

[80] M. S. Kesselring, F. Pastawski, J. Eisert, and B. J. Brown, The boundaries and twist defects of the color code and their applications to topological quantum computation, Quantum **2**, 101 (2018).

[81] J. C. Bridgeman, S. D. Bartlett, and A. C. Doherty, Tensor networks with a twist: Anyon-permuting domain walls and defects in projected entangled pair states, Physical Review B **96**, 10.1103/physrevb.96.245122 (2017).

[82] E. H. Spanier and E. H. Spanier, *Algebraic topology* (Springer Science & Business Media, 1989).

[83] X.-G. Wen, Construction of bosonic symmetry-protected-trivial states and their topological invariants via $g \times so(\infty)$ nonlinear $\sigma$ models, Physical Review B **91**, 205101 (2015).

[84] Y. Ren and P. Shor, Topological quantum computation assisted by phase transitions, arXiv preprint arXiv:2311.00103 (2023).

[85] P. Etingof, D. Nikshych, and V. Ostrik, On fusion categories, Annals of mathematics , 581 (2005).

[86] P. Etingof, S. Gelaki, D. Nikshych, and V. Ostrik, *Tensor categories*, Vol. 205 (American Mathematical Soc., 2015).

[87] C. G. Brell, Generalized cluster states based on finite groups, New Journal of Physics **17**, 023029 (2015).

[88] K. Inamura and S. Ohyama, 1+ 1d spt phases with fusion category symmetry: interface modes and non-abelian thouless pump, arXiv preprint arXiv:2408.15960 (2024).

[89] X. Chen, Z.-C. Gu, Z.-X. Liu, and X.-G. Wen, Symmetry protected topological orders and the group cohomology of their symmetry group, Physical Review B—Condensed Matter and Materials Physics **87**, 155114 (2013).

[90] B. Yoshida, Topological color code and symmetry-protected topological phases, Phys. Rev. B **91**, 245131 (2015).

[91] M. Levin and Z.-C. Gu, Braiding statistics approach to symmetry-protected topological phases, Physical Review B—Condensed Matter and Materials Physics **86**, 115109 (2012).

[92] W. Ji, N. Tantivasadakarn, and C. Xu, Boundary states of three dimensional topological order and the deconfined quantum critical point, SciPost Physics **15**, 231 (2023).

[93] Z.-X. Luo, Gapped boundaries of (3+ 1)-dimensional topological order, Physical Review B **107**, 125425 (2023).

[94] Y. Hu and Y. Wan, Electric-magnetic duality in twisted quantum double model of topological orders, Journal of High Energy Physics **2020**, 1 (2020).

[95] A. Dua, T. Jochym-O'Connor, and G. Zhu, Quantum error correction with fractal topological codes, Quantum **7**, 1122 (2023).

[96] G. Zhu, T. Jochym-O'Connor, and A. Dua, Topological order, quantum codes, and quantum computation on fractal geometries, PRX Quantum **3**, 030338 (2022).

[97] Z. Song and G. Zhu, Magic boundaries of 3d color codes, arXiv preprint arXiv:2404.05033 (2024).

[98] R. Wen, String condensation and topological holography for 2+ 1d gapless spt, arXiv preprint arXiv:2408.05801 (2024).

[99] T. D. Ellison, Y.-A. Chen, A. Dua, W. Shirley, N. Tantivasadakarn, and D. J. Williamson, Pauli stabilizer models of twisted quantum doubles, PRX Quantum **3**, 010353 (2022).

[100] B. J. Brown, D. Loss, J. K. Pachos, C. N. Self, and J. R. Wootton, Quantum memories at finite temperature, Rev. Mod. Phys. **88**, 045005 (2016).

[101] D. Aasen, D. Bulmash, A. Prem, K. Slagle, and D. J. Williamson, Topological defect networks for fractons of all types, Physical Review Research **2**, 043165 (2020).

[102] Z. Song, A. Dua, W. Shirley, and D. J. Williamson, Topological defect network representations of fracton stabilizer codes, PRX Quantum **4**, 010304 (2023).

[103] D. J. Williamson and N. Baspin, Layer codes, Nature Communications **15**, 9528 (2024).

[104] X. Chen, Z.-C. Gu, and X.-G. Wen, Local unitary transformation, long-range quantum entanglement, wave function renormalization, and topological order, Physical Review B—Condensed Matter and Materials Physics

**82**, 155138 (2010).

[105] A. Bullivant, Y. Hu, and Y. Wan, Twisted quantum double model of topological order with boundaries, Phys. Rev. B **96**, 165138 (2017).

[106] Y. Hu, Y. Wan, and Y.-S. Wu, Twisted quantum double model of topological phases in two dimensions, Physical Review B—Condensed Matter and Materials Physics **87**, 125114 (2013).

[107] A. Mesaros and Y. Ran, Classification of symmetry enriched topological phases with exactly solvable models, Phys. Rev. B **87**, 155115 (2013).

**Appendix A: Introduction to group cohomology, topological action, and SPT states**

In this section, we provide a brief introduction to group cohomology, SPT topological actions, and the corresponding SPT fixed-point states.

## 1. Group cohomology

An $n$-cochain is a map from group elements in $G^n = G \times ... \times G$ to a $U(1)$ value[4]. We denote it as $c_n(g_1, g_2, ..., g_n)$. The set of $n$-cochain forms a group $\mathcal{C}^n(G, U(1))$[5], and the group elements have the following multiplication rule,

$$(c \cdot c')_n(g_1, g_2, ..., g_n) = c_n(g_1, g_2, ..., g_n) \cdot c'_n(g_1, g_2, ..., g_n). \tag{A1}$$

The coboundary operator $\delta$ is a map from $\mathcal{C}^n(G, U(1))$ to $\mathcal{C}^{n+1}(G, U(1))$, $\delta : \mathcal{C}^n \to \mathcal{C}^{n+1}$, and the element $c_n \in \mathcal{C}^n(G, U(1))$ to $\delta c_{n+1} \in \mathcal{C}^{n+1}(G, U(1))$. The map is given by the following.

$$\delta c_{n+1}(g_1, g_2, ..., g_{n+1}) := c_n(g_2, g_3, ..., g_{n+1}) c_n(g_1, g_2, ..., g_n)^{(-1)^{n+1}} \prod_{i=1}^{n} c_n(g_1, ..., g_i g_{i+1}, ..., g_{n+1})^{(-1)^i}. \tag{A2}$$

One can check the coboundary operator $\delta$ is nilpotent, which satisfied $\delta^2 = 1$. We can further define two subgroups, $\mathcal{Z}^n(G, U(1)) \subset \mathcal{C}^n(G, U(1))$ and $\mathcal{B}^n(G, U(1)) \subset \mathcal{C}^n(G, U(1))$, such that

$$\mathcal{Z}^n(G, U(1)) := \{\omega_n \mid \delta\omega_n = 1\}, \tag{A3}$$

which is the $n$-cocycle, and

$$\mathcal{B}^n(G, U(1)) := \{b_n \mid b_n = \delta c_n, \quad \forall c_{n-1} \in \mathcal{C}^{n-1}\}, \tag{A4}$$

which is the $n$-coboundary. By the nilpotency property, one can check that $\mathcal{B}^n(G, U(1)) \subseteq \mathcal{Z}^n(G, U(1))$. The cohomology group can be defined as,

$$\mathcal{H}^n(G, U(1)) := \mathcal{Z}^n(G, U(1))/\mathcal{B}^n(G, U(1)). \tag{A5}$$

Let us discuss some simple examples. When $n = 1$, we have

$$\mathcal{Z}^1(G, U(1)) = \{\omega_1 \mid \omega_1(g_1)\omega_1(g_2) = \omega_1(g_1 g_2)\} \tag{A6}$$

and,

$$\mathcal{B}^1(G, U(1)) = \{1\}. \tag{A7}$$

Therefore, the first cohomology group $\mathcal{H}^1(G, U(1)) = \mathcal{Z}^1(G, U(1))/\mathcal{B}^1(G, U(1))$ is the set of all the inequivalent 1d representations of $G$.

When $n = 2$, we have

$$\mathcal{Z}^2(G, U(1)) = \{\omega_2 \mid \omega_2(g_1, g_2 g_3)\omega_2(g_2, g_3) = \omega_2(g_1 g_2, g_3)\omega_2(g_1, g_2)\}, \tag{A8}$$

and

$$\mathcal{B}^2(G, U(1)) = \{\omega_2 \mid \omega_2(g_1, g_2) = \omega_1(g_1)\omega_1(g_2)/\omega_1(g_1 g_2)\}. \tag{A9}$$

The cohomology group $\mathcal{H}^2(G, U(1))$ classifies the inequivalent projective representations.

––––––––

[4] In general, $U(1)$ can be replaced by any $G$-module $M$. Throughout this paper, we only study the case when $M = U(1)$.
[5] In the literature, it is also denoted by $\mathcal{C}^n(X, M)$, representing the group of $M$-valued $n$-cochains on the topological space $X$. This group consists of all maps from the $n$-simplices of $X$ to $M$.

When $n = 3$, we have

$$\mathcal{Z}^3(G, U(1)) = \{\omega_3 \mid \frac{\omega_3(g_2, g_3, g_4)\omega_3(g_1, g_2 g_3, g_4)\omega_3(g_1, g_2, g_3)}{\omega_3(g_1 g_2, g_3, g_4)\omega_3(g_1, g_2, g_3 g_4)} = 1\}, \tag{A10}$$

and

$$\mathcal{B}^3(G, U(1)) = \{\omega_3 \mid \omega_3(g_1, g_2, g_3) = \frac{\omega_2(g_1, g_2 g_3)\omega_2(g_2, g_3)}{\omega_2(g_1 g_2, g_3)\omega_2(g_1, g_2)}\}, \tag{A11}$$

The cohomology group $\mathcal{H}^3(G, U(1))$ classifies inequivalent classes of $F$-symbols of the fusion category $\mathrm{Vec}_G$.

Besides the coboundary operator $\delta$, one could also define the slant product $i_g : \mathcal{C}^n \to \mathcal{C}^{n-1}$, such that

$$i_g c_{n-1}(g_1, g_2, ..., g_{n-1}) := c_n(g, g_1, g_2, ..., g_{n-1})^{(-1)^{n-1}} \prod_{i=1}^{n-1} c_n(g_1, ..., g_i, g, g_{i+1}, ..., g_{n-1})^{(-1)^{n-1+i}}. \tag{A12}$$

The slant product and the coboundary operator have the following relation,

$$\delta(i_g c)_n = i_g(\delta c)_n, \quad \forall c_n \in \mathcal{C}^n. \tag{A13}$$

For the above equation, we can also see that if $c_n$ is an $n$-cocycle, then $i_g c_{n-1}$ is an $(n-1)$-cocycle. Therefore, the slant product gives rise to a homomorphism $i_g : \mathcal{H}^n(G, U(1)) \to \mathcal{H}^{n-1}(G, U(1))$, $\forall g \in G$.

Physically, the slant product of the 3-cocycle $\omega^3 \in \mathcal{H}^3(G, U(1))$ corresponds to the definition of the dyon charges of the $G$ twisted quantum double. We refer the readers to Ref. [72] for more details.

## 2. Cup product

Instead of the algebraic interpretation of group cohomology we introduced above, there is also a geometric interpretation. In this interpretation, $\mathcal{C}^n(G, M)$ is the group of $n$-cochains satisfying the following condition,

$$\mathcal{C}^n(G, M) := \{\nu_n | g\nu_n(g_0, g_1, ..., g_n) = \nu_n(gg_0, gg_1, ..., gg_n)\}, \tag{A14}$$

in which $\nu_n$ is a map such that $\nu_n : G^{n+1} \to M$. The relation between $\nu_n$ and $c_n$ is given as follows:

$$\begin{aligned} c_n(g_1, g_2, ..., g_n) &= \nu_n(1, g_1, g_1 g_2, ..., g_1 g_2 \cdots g_n) \\ &= \nu_n(1, \tilde{g}_1, \tilde{g}_2, ..., \tilde{g}_n) \end{aligned} \tag{A15}$$

There is a map between a $G$-valued $n$-simplex and the $n$-cochain $\nu_n$. When $n = 1$, we have

$$\overset{g_2}{\bullet} \longleftarrow \overset{g_1}{\bullet} \longrightarrow \nu_1(g_1, g_2). \tag{A16}$$

When $n = 2$, we have the following

$$\longrightarrow \nu_2(g_1, g_2, g_3). \tag{A17}$$

More examples can be found in Ref. [89]. Because of this geometric interpretation, the expression for coboundary operation becomes more compact. We have

$$\delta\nu_{n+1}(g_0, ..., g_{n+1}) = \prod_{i=0}^{n+1} \nu_n^{(-1)^i}(g_0, ..., g_{i-1}, g_{i+1}, ..., g_{n+1}) \tag{A18}$$

The coboundaries and cocycles are defined similarly as in Eq. (A4) and Eq. (A3).

Consider we have two cochains $\nu_{n_1} \in \mathcal{C}^{n_1}(G, U(1))$ and $\nu_{n_2} \in \mathcal{C}^{n_2}(G, U(1))$. We can construct a map such that $\mathcal{C}^{n_1}(G, U(1)) \times \mathcal{C}^{n_2}(G, U(1)) \to \mathcal{C}^{n_1+n_2}(G, U(1))$[6]:

$$\nu_{n_1}(g_0, g_1, ..., g_{n_1}) \cup \nu_{n_2}(g_{n_1}, g_{n_1+1}, ..., g_{n_1+n_2}) = \nu_{n_1+n_2}(g_0, g_1, ..., g_{n_1+n_2}). \tag{A19}$$

The product $\cup$ is called the *cup product*. The cup product satisfies the Leibniz Rule.

$$\delta(\nu_{n_1} \cup \nu_{n_2}) = \delta(\nu_{n_1}) \cup \nu_{n_2} + (-1)^{n_1} \nu_{n_1} \cup \delta(\nu_{n_2}). \tag{A20}$$

From the Leibniz rule we can see that the product of two cocycles gives rise to another cocycle.

### 3. Topological action and SPT state

In this subsection, we provide a brief introduction to the topological action, and its corresponding fixed-point SPT wavefunctions.

For a $(d+1)$-dimensional $G$-SPT state, the topological action is given by

$$S^{\text{top}}[M, A] = 2\pi i \int_M \mathcal{L}[A], \tag{A21}$$

in which $A$ is a background gauge field that is a $G$-valued 1-cocycle, and $M$ is the spacetime manifold. The Lagrangian $\mathcal{L}[A]$ is a topological invariant and satisfies,

$$\delta \mathcal{L} = 0, \tag{A22}$$

where the coboundary operator $\delta$ is defined in Eq. (A4). Two Lagrangians differed by a coboundary $\delta\Theta$ are equivalent,

$$\mathcal{L}' = \mathcal{L} + \delta\Theta, \tag{A23}$$

where $\Theta$ is a $d$-cochain. Distinct classes of inequivalent Lagrangians give rise to different bosonic $G$-SPT phases, which are classified by the cohomology group $\mathcal{H}^{d+1}(G, U(1))$.

To obtain the corresponding SPT state, one could write the physical action of the theory in terms of the physical fields (matter fields) from the topological action. When the symmetry group is $\mathbb{Z}_n$, for example, we can replace the the $\mathbb{Z}_n$-valued gauge field $A$ by $[\delta\phi]_n$, where $[x]_n := x \mod n$. $\phi$ is the physical field of the SPT which is a $\mathbb{Z}_n$-valued 0-cochain. The global symmetry is given by the following transformation,

$$\phi \to \phi + c, \tag{A24}$$

where $c$ is a $\mathbb{Z}_n$-valued 0-cocycle.

Upon this replacement, the physical Lagrangian is obtained as a coboundary $\mathcal{L}[\delta\phi] = \delta\omega[\phi]$. On an manifold $M$ with boundary $\partial M$, the path integral of the theory gives rise to a quantum state. Therefore, the SPT state defined on such manifold is independent of the bulk, and can be fully characterized by its boundary $\partial M$,

$$|\Psi\rangle_{SPT} \propto \sum_{\phi \in \mathcal{C}^0(\partial M, G)} e^{2\pi i \int_{\partial M} \omega[\phi]} |\phi\rangle, \tag{A25}$$

in which $\phi$ is a $G$-valued 0-cochain on $\partial M$, and $\omega[\phi]$ is the Lagrangian on $\partial M$.

In the remainder of this subsection, we provide an example to illustrate how to write down the SPT state given a topological action on a lattice. Let us consider the topological action given by Eq. (6),

$$\mathcal{L} = \frac{1}{2} A_1 \cup A_2, \tag{A26}$$

---

[6] In general, the cup product gives rise to the following map, $\mathcal{C}^{n_1}(G, M_1) \times \mathcal{C}^{n_2}(G, M_2) \to \mathcal{C}^{n_1+n_2}(G, M_1 \otimes_{\mathbb{Z}} M_2)$, where $M_1$ and $M_2$ are $G$-modules, and $\otimes_{\mathbb{Z}}$ maps two modules $M_1$ over $\mathbb{Z}$ and $M_2$ over $\mathbb{Z}$ to another module $M_3$ over $\mathbb{Z}$ such that $M_3 = M_1 \otimes_{\mathbb{Z}} M_2$. When $M_1 = M_2 = U(1)$, $M_3 = U(1)$. We only consider this case throughout this paper.

where $A_1$ and $A_2$ are $\mathbb{Z}_2$-valued 1-cocycles. A discussion of the cup product is provided in Appendix A 2. After we substitute the gauge fields by the corresponding physical fields, $\delta\phi_1$ and $\delta\phi_2$, we have,

$$S = 2\pi i \int_M \frac{1}{2}\delta\phi_1 \cup \delta\phi_2 = 2\pi i \int_M \delta\left(\frac{1}{2}\phi_1 \cup \delta\phi_2\right), \tag{A27}$$

where $\phi_1$ and $\phi_2$ are $\mathbb{Z}_2$-valued 0-cocycles. One can check this action has a $\mathbb{Z}_2 \times \mathbb{Z}_2$ symmetry given by the following transformation,

$$\phi_1 \to \phi_1 + c_1, \quad \phi_2 \to \phi_2 + c_2, \tag{A28}$$

where $c_1$ and $c_2$ are 0-cocycles. Using the Stokes' theorem, the SPT state on the spatial dimension is thus given by the path integral,

$$|\Psi\rangle_{SPT} = \sum_{\phi_1,\phi_2} e^{2\pi i \int_{\partial M} \frac{1}{2}\phi_1 \cup \delta\phi_2}|\phi_1, \phi_2\rangle. \tag{A29}$$

On a 1d spin chain with $n$ sites, where each site hosts two qubits, $\phi_1, \phi_2 \in \{0,1\}^n$ and $|\phi_1, \phi_2\rangle$ corresponds to a configuration of the qubits in the computational basis. We denote the values of the fields at site $i$ as $\phi_1(i)$ and $\phi_2(i)$. Let us consider the following branching structure of the chain,

$$\begin{array}{c} i-1 \quad i \quad i+1 \\ \phi_1 \\ \phi_2 \end{array} \tag{A30}$$

On each site, there are two qubits corresponding to $\phi_1$ and $\phi_2$ respectively. We first combine the terms on edge $\langle i-1, i\rangle$ and edge $\langle i, i+1\rangle$ for even $i$,

$$\begin{aligned} &(\phi_1 \cup \delta\phi_2)_{\langle i-1,i\rangle} + (\phi_1 \cup \delta\phi_2)_{\langle i,i+1\rangle} \\ &= \phi_1(i)(\phi_2(i) - \phi_2(i-1)) + \phi_1(i)(\phi_2(i+1) - \phi_2(i)) \\ &= \phi_1(i)(\phi_2(i-1) + \phi_2(i+1)). \end{aligned} \tag{A31}$$

We see that $\phi_1(odd)$ and $\phi_2(even)$ are decoupled from the rest. Therefore, we can ignore the decoupled qubits, such that effectively there is only one qubit at each site, as depicted below,

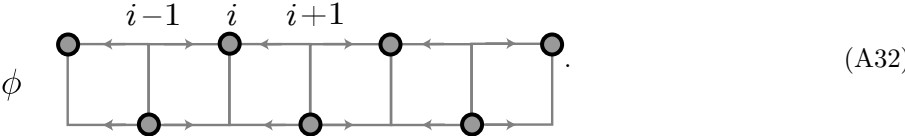

$$\begin{array}{c} i-1 \quad i \quad i+1 \\ \phi \end{array} \tag{A32}$$

The path integral in Eq. (A29) can be written in the following form

$$\begin{aligned} |\Psi\rangle_{SPT} &= \sum_{\{\phi(i)\}} e^{\pi i \sum_{i \text{ even}} \phi_1(i)(\phi_2(i-1)+\phi_2(i+1))}|\{\phi(i)\}\rangle \\ &= (-1)^{\sum_{i \text{ even}} \frac{1-Z_i}{2}\frac{1-Z_{i-1}}{2} + \frac{1-Z_i}{2}\frac{1-Z_{i+1}}{2}}|+\rangle^{\otimes n} \\ &= \prod_{i \text{ even}} CZ_{i,i-1}CZ_{i,i+1}|+\rangle^{\otimes n} \end{aligned} \tag{A33}$$

This is the ground state wavefunction of 1d $\mathbb{Z}_2 \times \mathbb{Z}_2$ cluster state and the Hamiltonian can be written as follows,

$$H_{\text{cluster}} = -\sum_i Z_{i-1}X_iZ_{i+1}. \tag{A34}$$

### 4. $G$-SPT phases in $2+1$D

In this subsection, we construct the $2+1$D $G$-SPT Hamiltonian and ground state wavefuncion. Consider a lattice $\mathcal{L}$ with vertices $v \in \mathcal{V}$. We assign one qudit per vertex. The qudit state is given by $|g\rangle$, $g \in G$. The trivial $G$-symmetric ground state can be written as,

$$|\Psi\rangle = \left( \frac{1}{|G|} \sum_{g \in G} |g\rangle \right)^{\otimes N}. \tag{A35}$$

The global symmetry operator is given by the tensor product of $L_+^g$ operators on every qudit, $\prod_v L_+^g$.

The local Hamiltonian is given by the right group multiplication. We have

$$H = -\sum_v L_{-,v}^g. \tag{A36}$$

In 2+1D bosonic qudit systems with finite global symmetry group $G$, it has been shown that these systems can be classified according to the third group cohomology $H^3(G, U(1))$ [89]. Different phases are labeled by cohomology classes of 3-cocycles $[\omega] \in H^3(G, U(1))$, and cannot be transformed into one another by finite-depth, symmetry-preserving local unitary circuits [104]. The trivial class in the cohomology group corresponds to the ground state given in Eq. (A35) and the Hamiltonian in Eq. (A36). It is also referred to as the trivial symmetry-protected topological (SPT) phases. The nontrivial classes in the cohomology group give rise to the nontrivial SPT phases.

When there exists nontrivial 3-cocycles, $\omega_3(g_1, g_2, g_3)$, the corresponding SPT ground state can be obtained by applying the SPT entangler $U_{SPT}$ on the product state in Eq. (A35). The SPT entangler is a phase gate. The phases correspond to different states are given as follows.

$$= \omega_3(g_3, g_3^{-1}g_2, g_2^{-1}g_1), \qquad = \omega_3^{-1}(g_3, g_3^{-1}g_2, g_2^{-1}g_1). \tag{A37}$$

The fixed-point SPT wavefunction can be written as [61, 89],

$$|\Psi\rangle_{\text{SPT}} = U_{\text{SPT}} \left( \frac{1}{|G|} \sum_{g \in G} |g\rangle \right)^{\otimes N} = \sum_{\{g_i\}} \prod_{\triangle_{123}} \omega_3^{s(\triangle_{123})}(g_3, g_3^{-1}g_2, g_2^{-1}g_1)|\{g_i\}\rangle, \tag{A38}$$

in which $\prod_{\triangle_{123}} \omega_3$ is the product of the phase factors $\omega_3$ of a certain spin configuration, the sum is over all the spin configurations, and $s(\triangle_{123}) = \pm 1$ depends on the orientation of the triangle, as discussed in Eq. (A37).

The SPT Hamiltonian is given by

$$H_{\text{SPT}} = U_{\text{SPT}} H U_{\text{SPT}}^\dagger, \tag{A39}$$

where $H$ is given in Eq. (A36).

The SPT ground state Eq. (A38) is symmetric under global symmetry transformation, $\prod_v L_{+,v}^g$. We have

$$
\begin{aligned}
\prod_v L_{+,v}^g |\Psi\rangle_{\text{SPT}} &= \sum_{\{g_i\}} \prod_{\triangle} \omega_3^{s(\triangle)}(g_3, g_3^{-1}g_2, g_2^{-1}g_1) \prod_v L_{+,v}^g |\{g_i\}\rangle \\
&= \sum_{\{g_i\}} \prod_{\triangle} \omega_3^{s(\triangle)}(g_3, g_3^{-1}g_2, g_2^{-1}g_1)|\{gg_i\}\rangle \\
&= \sum_{\{g_i\}} \prod_{\triangle} \omega_3^{s(\triangle)}(g^{-1}g_3, g_3^{-1}g_2, g_2^{-1}g_1)|\{g_i\}\rangle,
\end{aligned}
\tag{A40}
$$

By applying the 3-cocycle condition in Eq. (A10), we obtain the following relation,

$$\omega_3(g^{-1}g_3, g_3^{-1}g_2, g_2^{-1}g_1) = \frac{\omega_3(g^{-1}, g_2, g_2^{-1}g_1)\omega_3(g^{-1}, g_3, g_3^{-1}g_2)}{\omega_3(g^{-1}, g_3, g_3^{-1}g_1)}\omega_3(g_3, g_3^{-1}g_2, g_2^{-1}g_1) \tag{A41}$$

One can show that, on a closed manifold, we have,

$$\prod_{\triangle}\omega_3^{s(\triangle)}(g_3, g_3^{-1}g_2, g_2^{-1}g_1) = \prod_{\triangle}\omega_3^{s(\triangle)}(g^{-1}g_3, g_3^{-1}g_2, g_2^{-1}g_1). \tag{A42}$$

Therefore, we have

$$\prod_v L_{+,v}^g|\Psi\rangle_{\text{SPT}} = |\Psi\rangle_{\text{SPT}}, \tag{A43}$$

when the manifold is closed, and $\prod_v L_{+,v}^g$ is the global symmetry of the SPT phases.

### 5.  $1+1$D $(\mathbb{Z}_3 \times \mathbb{Z}_3) \rtimes \mathbb{Z}_2$ SPT phases

As an illustrative example, and also for the benefit of later discussion, let us consider constructing the lattice model for the $1+1$D $(\mathbb{Z}_3 \times \mathbb{Z}_3) \rtimes \mathbb{Z}_2$ SPT. Consider we have a $1+1D$ $S_3 \times S_3$ state, and the $\mathbb{Z}_2 \times \mathbb{Z}_2$ subgroup breaks in the the diagonal group $\mathbb{Z}_2$. The corresponding nontrivial SPTs are classified by the nontrivial classes $[\omega] \in H^2((\mathbb{Z}_3 \times \mathbb{Z}_3) \rtimes \mathbb{Z}_2, U(1))$. In this subsection, we show in details on how to construct the ground state wavefunction and the corresponding stabilizers. This SPT will later have an important role. After gauging, it becomes the $C \leftrightarrow F$ exchange domain wall of $S_3$ quantum double; see Appendix E 2.

The group elements of $(\mathbb{Z}_3 \times \mathbb{Z}_3) \rtimes \mathbb{Z}_2$ are generated by $c_1, c_2$, and $t$. The group presentation is

$$c_1^3 = c_2^3 = 1, \quad tc_1t = c_1^2, \quad tc_2t = c_2^2, \quad c_1c_2 = c_2c_1. \tag{A44}$$

The group cohomology is given by

$$\mathcal{H}^2((\mathbb{Z}_3 \times \mathbb{Z}_3) \rtimes \mathbb{Z}_2, U(1)) = \mathbb{Z}_3. \tag{A45}$$

The 2-cocycles can be written as follows,

$$\omega_2\left(c_1^{i_1}c_2^{j_1}t^{k_1}, c_1^{i_2}c_2^{j_2}t^{k_2}\right) = \exp\left(\frac{2\pi i}{3}ni_1j_2(-1)^{k_1}\right). \tag{A46}$$

Similar to the $2+1$D cases, one can define the SPT entangler for the $1+1$D SPT. The action of the entangler on 1D is given by

$$\underset{g_1}{\bullet} \longrightarrow \underset{g_2}{\bullet} = \omega_2(g_2, g_2^{-1}g_1), \quad \underset{g_2}{\bullet} \longleftarrow \underset{g_1}{\bullet} = \omega_2^{-1}(g_2, g_2^{-1}g_1). \tag{A47}$$

The SPT-entangler can thus be written as,

$$\omega_2\left(c_1^{i_2}c_2^{j_2}t^{k_2}, t^{-k_2}c_2^{-j_2}c_1^{-i_2}c_1^{i_1}c_2^{j_1}t^{k_1}\right) = \omega_2\left(c_1^{i_2}c_2^{j_2}t^{k_2}, c_1^{(i_1-j_2)(-1)^{k_2}}c_2^{(j_1-j_2)(-1)^{k_2}}t^{k_1-k_2}\right)$$
$$= \exp\left(\frac{2\pi i}{3}ni_2(j_1 - j_2)\right). \tag{A48}$$

Let us consider the following lattice with the SPT entangler,

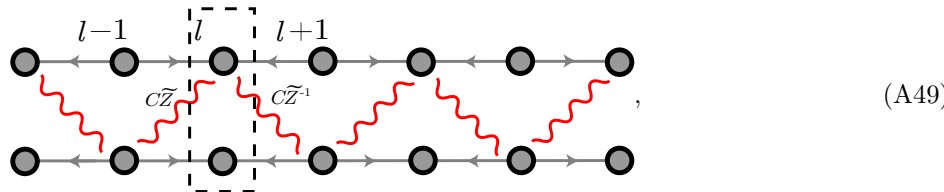

$$\tag{A49}$$

in which $l$ is the label of site. Without loss of generality, we assume $l$ is even throughout the paper. At each site $l$, there is a $(\mathbb{Z}_3 \times \mathbb{Z}_3) \rtimes \mathbb{Z}_2$ qudit. The qudit state can be written as $|c_1^{i_l} c_2^{j_l} t^{k_l}\rangle$. $|c_1^{i_l}\rangle$ represents the state of the upper qutrit, $|c_2^{j_l}\rangle$ represents the state of the lower qutrit, and $|t^{k_l}\rangle$ represents the state of the qubit.

After applying the SPT entangler, the original stabilizers are no longer stabilizers. To find the new stabilizers, we do the following calculation.[7]

$$
\begin{aligned}
&L_{-,l}^{c_1} \sum_{i,j} \exp\left(\frac{2\pi i}{3} \sum_m i_m(j_{m-1} - j_{m+1})\right) |..., c_1^{i_m} c_2^{j_m} t^{k_m}, ...\rangle \\
&= \sum_{i,j} \exp\left(\frac{2\pi i}{3} \sum_m i_m(j_{m-1} - j_{m+1})\right) |..., c_1^{i_l - Z_l} c_2^{j_l} t^{k_l}, ...\rangle \\
&= \sum_{i,j} \exp\left(\frac{2\pi i}{3}\left((i_l + Z_l)(j_{l-1} - j_{l+1}) + \sum_{m \neq l} i_m(j_{m-1} - j_{m+1})\right)\right) |..., c_1^{i_l} c_2^{j_l} t^{k_l}, ...\rangle \\
&= \sum_{i,j} \exp\left(\frac{2\pi i}{3}\left(Z_l(j_{l-1} - j_{l+1}) + \sum_m i_m(j_{m-1} - j_{m+1})\right)\right) |..., c_1^{i_l} c_2^{j_l} t^{k_l}, ...\rangle .
\end{aligned}
\tag{A50}
$$

Reorganizing the above equation, we get

$$
\begin{aligned}
&L_{-,l}^{c_1} \left(\widetilde{Z}_{l+1,2} \widetilde{Z}_{l-1,2}^{-1}\right)^{Z_l} \sum_{i,j} \exp\left(\frac{2\pi i}{3} \sum_m i_m(j_{m-1} - j_{m+1})\right) |..., c_1^{i_m} c_2^{j_m} t^{k_m}, ...\rangle \\
&= \sum_{i,j} \exp\left(\frac{2\pi i}{3} \sum_m i_m(j_{m-1} - j_{m+1})\right) |..., c_1^{i_m} c_2^{j_m} t^{k_m}, ...\rangle .
\end{aligned}
\tag{A51}
$$

in which $\widetilde{Z}_{l+1,2} = \exp(2\pi i j_{l+1}/3)$, $\widetilde{Z}_{l-1,2}^{-1} = \exp(2\pi i j_{l-1}/3)$ correspond to the eigenvalues of the generalized Pauli $Z$ operator on the qutrit states $|j_{l+1}\rangle$ and $|j_{l-1}\rangle$, and $Z_l = (-1)^{k_l}$ corresponds to the eigenvalue of Pauli $Z$ on the state $|k_l\rangle$ Therefore, $L_{-,l}^{c_1} \left(\widetilde{Z}_{l+1,2} \widetilde{Z}_{l-1,2}^{-1}\right)^{Z_l}$ is a stabilizer of the SPT. Similarly, one can obtain all the other stabilizers, which are given as follows,

$$\tag{A52}$$

The last $L_-^t$ operator is applied on the qubit at each site.

---

[7] Here we assume $n = 1$. The $n = 2$ result can be obtained similarly.

## 6. Künneth formula and the classification of 1d domain walls

When $M$ is Abelian, finitely-generated, and a trivial $G \times G'$ module, we have the following Künneth formula [83].

$$\mathcal{H}^d \left( G \times G', M \right) = \bigoplus_{k=0}^{d} \mathcal{H}^k \left( G, \mathcal{H}^{d-k} \left( G', M \right) \right). \tag{A53}$$

In 1+1d, the Künneth formula can help us understand the domain wall classifications. Let us look at one example. When $d = 2$, $M = U(1)$, and $G = G'$ are finite Abelian groups, we have

$$\mathcal{H}^2(G \times G, U(1)) = \mathcal{H}^0(G, \mathcal{H}^2(G, U(1))) \oplus \mathcal{H}^1(G, \mathcal{H}^1(G, U(1))) \oplus \mathcal{H}^2(G, \mathcal{H}^0(G, U(1)))$$
$$= \mathcal{H}^2(G, U(1)) \oplus \mathcal{H}^1(G, \hat{G}) \oplus \mathcal{H}^2(G, U(1)), \tag{A54}$$

in which $\hat{G}$ is the representation of $G$. The $\mathcal{H}^2(G, U(1))$ classifies gauged $G$-SPT defects, while the $\mathcal{H}^1(G, \hat{G})$ classifies the maps between electric and magnetic sectors of $G$ quantum double. As we mentioned in the previous subsection Appendix A 1, the first cohomology group $\mathcal{H}^1(G, U(1))$ classifies inequivalent 1d representations, while $\mathcal{H}^1(G, \hat{G})$ classifies group homomorphism between $G$ and $\hat{G}$. In quantum double models, magnetic fluxes are labeled by different conjugacy classes while the electric charges are labeled by different irreducible representations. When $G$ is Abelian, each elements corresponds to a conjugacy class and the representation of $G$ is just itself. Therefore, the group homomorphism between $G$ and $\hat{G}$ just tells us the map of anyons between two quantum doubles with symmetry $G$. When $\hat{G}$ is trivial, this corresponds to the map that maps all magnetic fluxes to identity (vacuum), which gives rise to the smooth boundary. When it is nontrivial, it corresponds to a map between magnetic fluxes and electric charges between two $G$ quantum doubles.

When we have a $G^{(1)} \times G^{(2)} \times G^{(3)}$ gauged SPT as a domain wall, the Künneth formula is as follows,

$$\mathcal{H}^2(G^{(1)} \times G^{(2)} \times G^{(3)}, U(1)) = \mathcal{H}^2(G^{(1)}, U(1)) \oplus \mathcal{H}^2(G^{(2)}, U(1)) \oplus \mathcal{H}^2(G^{(3)}, U(1))$$
$$\oplus \mathcal{H}^1(G^{(1)}, \hat{G}^{(2)}) \oplus \mathcal{H}^1(G^{(3)}, \hat{G}^{(2)}) \oplus \mathcal{H}^1(G^{(1)}, \hat{G}^{(3)}). \tag{A55}$$

For a gauged $G^{(1)} \times G^{(2)} \times G^{(3)}$ SPT domain wall, it is classified by the gauged $G^{(i)}$-SPT defect, as well as the anyon maps between every two $G$ quantum doubles. Similar decomposition holds for more $G$ symmetries added.

When $G$ is non-Abelian, there exists higher-dimensional representations. Therefore, the domain wall given by $\mathcal{H}^1(G, \mathrm{Rep}^{1\mathrm{d}}(G'))$ is not invertible since the fluxes which correspond to higher-dimensional representations maps to 1 (vacuum). For example, when $G = G' = S_3$, and $M = U(1)$, we have

$$\mathcal{H}^2 \left( S_3 \times S_3, U(1) \right) = \mathcal{H}^0 \left( S_3, \mathcal{H}^2 \left( S_3, U(1) \right) \right) \oplus \mathcal{H}^1 \left( S_3, \mathcal{H}^1 \left( S_3, U(1) \right) \right) \oplus \mathcal{H}^2 \left( S_3, \mathcal{H}^0 \left( S_3, U(1) \right) \right)$$
$$= \mathcal{H}^2(S_3, U(1)) \oplus \mathcal{H}^1(S_3, \mathrm{Rep}^{1\mathrm{d}}(S_3)) \oplus \mathcal{H}^2(S_3, U(1)), \tag{A56}$$

The nontrivial class in $\mathcal{H}^1(S_3, \mathrm{Rep}^{1\mathrm{d}}(S_3))$ gives rise to the anyon maps $[e] \to 1$, $[c] \to 1$, and $[t] \to -1$. Therefore, it realizes the invertible domain wall that swaps $D$ and $B$ anyons, while is non-inveritible for $C$, $F$, $G$, and $H$. An in-depth discussion of the domain walls in $S_3$ quantum double is provided in Appendix E.

## Appendix B: Proof of Lemma 2 and Theorem 1

### 1. Proof of Lemma 2

According to the theorem 1 in Ref. [74], with some adjustments, any action on the Pauli operators can given by the conjugation of the following canonical unitary operator $U = \Omega \Pi \Omega'$. The three unitary operators

are

$$\Omega = \prod_{i,j=1}^{n} CZ_{i,j}^{\Gamma_{i,j}}, \quad \Omega' = \prod_{i,j=1}^{n} CZ_{i,j}^{\Gamma'_{i,j}},$$

$$\Pi = (\prod_{i \neq j} CX_{i,j}^{\Delta_{i,j}}) S (\prod_{i=1}^{n} H_i^{h_i}) (\prod_{i \neq j} CX_{i,j}^{\Delta'_{i,j}}),$$

(B1)

where $h_i, \Gamma_{i,j}, \Gamma'_{i,j} = 0, 1$, and $S$ is a permutation between n qubits. The matrices $\Delta, \Delta'$ are upper-triangular and unit-diagonal, i.e., $\Delta_{i,j} = \Delta'_{i,j} = 0$ for $i > j$ and $\Delta_{i,i} = \Delta'_{i,i} = 1$ for all $i$. The product of $CX$ gates is ordered such that the control qubit index increases from the left to the right. For example when $n = 4$, $\prod_{i \neq j} CX_{i,j}^{\Delta_{i,j}}$ is

$$CX_{1,2}^{\Delta_{1,2}} CX_{1,3}^{\Delta_{1,3}} CX_{1,4}^{\Delta_{1,4}} CX_{2,3}^{\Delta_{2,3}} CX_{2,4}^{\Delta_{2,4}} CX_{3,4}^{\Delta_{3,4}}.$$

(B2)

As we argued, the conjugation of the Clifford operators on the Pauli $Z_i$ and $X_j$ operators has a correspondence to the braided autoequivalences of anyon $e_i$ and $m_j$. We claim that the following SPT-sewing with $G \times G \times G$ symmetry gives rise to the corresponding anyon map

$$S_{\text{top}} = \int \sum_{i,j} \frac{\Gamma_{i,j}}{2} C_i \cup C_j + \sum_{i,j} \frac{\Gamma'_{i,j}}{2} A_i \cup A_j$$

$$+ \sum_{\substack{i,j, \\ h_j=0}} \frac{\Delta'_{i,j}}{2} A_i \cup B_{s(j)} + \sum_{\substack{j,k, \\ h_j=0}} \frac{\overline{\Delta}_{k,s(j)}}{2} B_{s(j)} \cup C_k + \sum_{\substack{i,j,k, \\ h_j=1}} \frac{\Delta'_{i,j} \overline{\Delta}_{k,s(j)}}{2} A_i \cup C_k,$$

(B3)

where $A_i$, $B_i$, and $C_i$ are the background gauge fields for the $i$-th $\mathbb{Z}_2$ subgroup in the first, the second, and the third $G$ symmetry respectively. To show the anyon map from this SPT-sewing, we first notice that the unitaries $\Omega$ and $\Omega'$ correspond to the flux-preserving maps of gauged-SPT domain walls given by the first two terms in the topological action.

For the conjugation of the unitary $\Pi$, we first notice that the anyon map from a unitary $(\prod_{i \neq j} CX_{i,j}^{\Delta_{i,j}})$ is given by

$$m_i \to \prod_{j=1}^{n} m_j^{\Delta_{i,j}}, \quad e_i \to \prod_{j=1}^{n} e_j^{\overline{\Delta}_{j,i}},$$

(B4)

where $\overline{\Delta}$ is the inverse matrix. As two special cases of the unitary $\Pi$, when $h_i \equiv 0$, the corresponding anyon map is given by

$$m_i \to \prod_{j,k=1}^{n} m_k^{\Delta'_{i,j} \Delta_{s(j),k}}, e_i \to \prod_{j,k=1}^{n} e_k^{\overline{\Delta}'_{j,i} \overline{\Delta}_{k,s(j)}}.$$

(B5)

We can use the following SPT-sewing to achieve this map

$$S_{\text{top}} = \int \sum_{i,j} \frac{\Delta'_{i,j}}{2} A_i \cup B_{s(j)} + \sum_{j,k} \frac{\overline{\Delta}_{k,s(j)}}{2} B_{s(j)} \cup C_k.$$

(B6)

When $h_i \equiv 1$, the corresponding anyon map is given by

$$m_i \to \prod_{j,k=1}^{n} e_k^{\Delta'_{i,j} \overline{\Delta}_{k,s(j)}}, e_i \to \prod_{j,k=1}^{n} m_k^{\overline{\Delta}'_{j,i} \Delta_{s(j),k}}.$$

(B7)

We can use the following SPT-sewing to achieve this map

$$S_{\text{top}} = \int \sum_{i,j,k} \frac{\Delta'_{i,j} \overline{\Delta}_{k,s(j)}}{2} A_i \cup C_k.$$

(B8)

In the general case of $h_i$, the conjugation of $\Pi$ gives rise to the following anyon map

$$m_i \to \prod_{k,h_j=0} m_k^{\Delta'_{i,j}\Delta_{s(j),k}} \prod_{k,h_j=1} e_k^{\Delta'_{i,j}\overline{\Delta}_{k,s(j)}}, \quad e_i \to \prod_{k,h_j=0} e_k^{\overline{\Delta}'_{j,i}\overline{\Delta}_{k,s(j)}} \prod_{k,h_j=1} m_k^{\overline{\Delta}'_{j,i}\Delta_{s(j),k}}. \tag{B9}$$

To see how the last three terms of the topological action give rise to the above anyon map, let us fold the topological order along the domain wall, such that the bulk becomes a quantum double of group $G \times G$, and the domain wall becomes a gapped boundary. To study the anyon map give by the SPT-sewn domain wall, we can look at the anyons that are condensed on this boundary. After unfolding, the anyon map is essentially from the component of a condensed anyon that is in the first copy, to the component that is in the second copy of the $G$ quantum double.

If we stack on top of the boundary another $G$ symmetry breaking state, we can think of this folded boundary as a SPT-sewn domain wall with $(G^{(A)} \times G^{(C)}) \times (G^{(B)} \times G^{(D)})$ symmetry, fusing with a smooth boundary. The folded SPT topological action is given by

$$S_{\text{top}} = \int \sum_{\substack{i,j,k,\\h_j=1}} \frac{\Delta'_{i,j}\overline{\Delta}_{k,s(j)}}{2} A_i \cup C_k + \sum_{\substack{i,j,\\h_j=0}} \frac{\Delta'_{i,j}}{2} A_i \cup B_{s(j)} + \sum_{\substack{i,j,\\h_j=0}} \frac{\overline{\Delta}_{i,s(j)}}{2} C_i \cup B_{s(j)}. \tag{B10}$$

The first term gives rise to a gauged-SPT domain wall, and the last two terms give rise to an SPT-sewn domain wall. We can thus write the 2-cocycle of this SPT as $\nu = \omega_1\eta$, the anyon map is given in Section IV B. We denote the anyons in copy $A$, $B$ and copy $C$, $D$ of the $G$ quantum double as $c$ and $\tilde{c}$ respectively. The anyon map from $\omega_1$ is flux-preserving,

$$m_i \xrightarrow{\omega_1} m_i \prod_{k,h_j=1} \tilde{e}_k^{\Delta'_{i,j}\overline{\Delta}_{k,s(j)}}, \quad \tilde{m}_i \xrightarrow{\omega_1} \tilde{m}_i \prod_{k,h_j=1} e_k^{\Delta'_{k,j}\overline{\Delta}_{i,s(j)}}, \quad e_i, \tilde{e}_i \xrightarrow{\omega_1} e_i, \tilde{e}_i. \tag{B11}$$

The anyon map from $\eta$ is a flux-charge exchange,

$$m_i \xrightarrow{\eta} \prod_{h_j=0} e_{s(j)}^{\Delta'_{i,j}}, \quad e_i \xrightarrow{\eta} \prod_{h_j=0} m_{s(j)}^{\overline{\Delta}'_{j,i}}, \quad \tilde{m}_i \xrightarrow{\eta} \prod_{h_j=0} e_{s(j)}^{\overline{\Delta}_{i,s(j)}}, \quad \tilde{e}_i \xrightarrow{\eta} \prod_{h_j=0} m_{s(j)}^{\Delta_{s(j),i}}. \tag{B12}$$

We notice that since $\eta$ is a degenerate 2-cocycle, which means the sewing of two smooth boundaries are not dense enough. As a result, the anyon map given above is not invertible, in that some fluxes might be condensed on, and some charges cannot pass through this SPT-sewing defect. For example, the product of charges on the RHS of the first two terms in Eq. (B11) are two anyons that cannot pass through

$$\prod_{k,h_j=1} \tilde{e}_k^{\Delta'_{i,j}\overline{\Delta}_{k,s(j)}} \xrightarrow{\eta} \prod_{k,h_j=1,h_r=0} m_{s(r)}^{\Delta_{s(r),k}\Delta'_{i,j}\overline{\Delta}_{k,s(j)}} = 1,$$
$$\prod_{k,h_j=1} e_k^{\Delta'_{k,j}\overline{\Delta}_{i,s(j)}} \xrightarrow{\eta} \prod_{k,h_j=1,h_r=0} m_{s(r)}^{\overline{\Delta}'_{r,k}\Delta'_{k,j}\overline{\Delta}_{i,s(j)}} = 1. \tag{B13}$$

Since our aim is to find all the anyons that are condensed on this boundary, we can just take the following bulk anyons,

$$m_i \prod_{k,h_j=1} \tilde{e}_k^{\Delta'_{i,j}\overline{\Delta}_{k,s(j)}}, \quad \tilde{m}_i \prod_{k,h_j=1} e_k^{\Delta'_{k,j}\overline{\Delta}_{i,s(j)}}, \tag{B14}$$

which will become $m_i$ and $\tilde{m}_i$ after the flux-preserving map given by $\omega_1$. Therefore either they can pass through this $\eta$ SPT-sewn domain wall, or they are condensed on this domain wall already. Combining the two anyon maps, we find that a combination of these two anyons is condensed on the domain wall,

$$m_i \prod_{k,h_j=1} \tilde{e}_k^{\Delta'_{i,j}\overline{\Delta}_{k,s(j)}} \xrightarrow{\omega_1\eta} \prod_{h_j=0} e_{s(j)}^{\Delta'_{i,j}}, \quad \tilde{m}_i \prod_{k,h_j=1} e_k^{\Delta'_{k,j}\overline{\Delta}_{i,s(j)}} \xrightarrow{\omega_1\eta} \prod_{h_j=0} e_{s(j)}^{\overline{\Delta}_{i,s(j)}}. \tag{B15}$$

Thus, taking the following combination

$$
m_i \prod_{k,h_j=1} \tilde{e}_k^{\Delta'_{i,j}\overline{\Delta}_{k,s(j)}} \prod_{t,h_r=0} \tilde{m}_t^{\Delta'_{i,r}\Delta_{s(r),t}}
$$

$$
= \Big(m_i \prod_{k,h_j=1} \tilde{e}_k^{\Delta'_{i,j}\overline{\Delta}_{k,s(j)}}\Big) \prod_{t,h_r=0} \Big(\tilde{m}_t \prod_{k,h_j=1} e_k^{\Delta'_{k,j}\overline{\Delta}_{t,s(j)}}\Big)^{\Delta'_{i,r}\Delta_{s(r),t}} \tag{B16}
$$

$$
\xrightarrow{\omega_1\eta} \Big(\prod_{h_j=0} e_{s(j)}^{\Delta'_{i,j}}\Big) \prod_{t,h_r=0} \Big(\prod_{h_j=0} e_{s(j)}^{\overline{\Delta}_{t,s(j)}}\Big)^{\Delta'_{i,r}\Delta_{s(r),t}} = 1.
$$

After unfolding, the above condensed anyon gives the first term in Eq. (B9). To find the other set of condensed anyon, we take use of the fact that all the flux anyons on the smooth boundary are condensed, and they can be pushed through the $\eta$ SPT-sewn domain wall into the bulk,

$$
1 \xleftarrow{\text{smooth}} \prod_{h_j=0} m_{s(j)}^{\overline{\Delta}'_{j,i}} \xleftarrow{\omega_1\eta} \prod_{h_j=0} \Big(\prod_t e_t^{\Delta'_{t,j}} \prod_k \tilde{e}_k^{\overline{\Delta}_{k,s(j)}}\Big)^{\overline{\Delta}'_{j,i}},
$$

$$
1 \xleftarrow{\eta} \prod_{k,h_j=1} \tilde{m}_k^{\overline{\Delta}'_{j,i}\Delta_{s(j),k}} \xleftarrow{\omega_1} \prod_{h_j=1} \Big(\prod_k \tilde{m}_k^{\overline{\Delta}_{s(j),k}} \prod_t e_t^{\Delta'_{t,j}}\Big)^{\overline{\Delta}'_{j,i}}. \tag{B17}
$$

Taking a product of the above, we obtain the following condensed anyon,

$$
e_i \prod_{k,h_j=0} \tilde{e}_k^{\overline{\Delta}'_{j,i}\overline{\Delta}_{k,s(j)}} \prod_{k,h_j=1} \tilde{m}_k^{\overline{\Delta}'_{j,i}\Delta_{s(j),k}} \tag{B18}
$$

After unfolding, the above condensed anyon gives the second term in Eq. (B9).

## 2.  Proof of Theorem 1

In this section, we prove that $G \times G \times G$ SPT-sewing gives rise to all the invertible domain walls in a 2d $G$ quantum double for any Abelian group $G$. An Abelian group is always isomorphic to a product of cyclic group, $G = \mathbb{Z}_{n_1} \times \mathbb{Z}_{n_2} \times \cdots \mathbb{Z}_{n_q}$. Thus the most general form of topological action that characterizes 1d $G \times G \times G$ SPT is given by

$$
S_{\text{top}} = \sum_{i,j=1}^q \int \frac{\Gamma_{i,j}^a}{gcd(n_i,n_j)} A_i \cup A_j + \frac{\Gamma_{i,j}^b}{gcd(n_i,n_j)} B_i \cup B_j + \frac{\Gamma_{i,j}^c}{gcd(n_i,n_j)} C_i \cup C_j
$$

$$
+ \frac{\Delta_{i,j}^{ab}}{gcd(n_i,n_j)} A_i \cup B_j + \frac{\Delta_{i,j}^{ac}}{gcd(n_i,n_j)} A_i \cup C_j + \frac{\Delta_{i,j}^{bc}}{gcd(n_i,n_j)} B_i \cup C_j, \tag{B19}
$$

where $\Gamma_{i,j}, \Delta_{i,j} = 0, 1, \cdots, gcd(n_i,n_j) - 1$, and $\Gamma_{i,i} = 0$. It turns out that like the case when $G = \mathbb{Z}_2^n$, to construct invertible domain walls, we do not need all those SPT phases. In the following, we will focus on the $\Gamma_{i,j}^b = 0$ cases.

Let us project the SPT state given by the topological action into the subspace of $X_i^{b,v} \equiv 1$, where $X_i^{b,v}$ is the generalized Pauli X operator for the $\mathbb{Z}_{n_i}$ subgroup of the second $G$ symmetry on vertex $v$. (While the projection is not necessary, it simplifies the analysis.) After the projection, the resultant state has symmetry $G^a \times G^c$, and is possibly in the spontaneously symmetry broken phase. The partition function of this phase is given by [35]

$$
Z(A,C) = \prod_{j=1}^q \delta \left( \sum_{i,k=1}^q \frac{\Delta_{i,j}^{ab}}{gcd(n_i,n_j)} A_i + \frac{\Delta_{j,k}^{bc}}{gcd(n_j,n_k)} C_k \mod 1 \right)
$$

$$
\exp\left(2\pi i \sum_{i,j=1}^q \int \frac{\Gamma_{i,j}^a}{gcd(n_i,n_j)} A_i \cup A_j + \frac{\Gamma_{i,j}^c}{gcd(n_i,n_j)} C_i \cup C_j + \frac{\Delta_{i,j}^{ac}}{gcd(n_i,n_j)} A_i \cup C_j \right). \tag{B20}
$$

The delta functions in the partition function is a symmetry breaking term. For example, if we take $A_1$ as a $\mathbb{Z}_2$-valued background gauge field, and $A_2$ as a $\mathbb{Z}_4$-valued background gauge field. A delta function $\delta(\frac{1}{2}A_1 \mod 1)$ in the partition function indicates the spontaneously breaking of symmetry from $\mathbb{Z}_2 \times \mathbb{Z}_4$ to the $\mathbb{Z}_4$ subgroup. A delta function $\delta(\frac{1}{2}A_1 + \frac{1}{4}A_4 \mod 1)$ in the partition function indicates the symmetry breaking from $\mathbb{Z}_2 \times \mathbb{Z}_4$ to a $\mathbb{Z}_2 = \langle t_1 t_2^2 \rangle$ subgroup, where $t_1$ is the generator of $\mathbb{Z}_2$ and $t_2$ is the generator of $\mathbb{Z}_4$. In comparison, a delta function $\delta(\frac{1}{2}A_1 + \frac{1}{2}A_4 \mod 1)$ in the partition function indicates the symmetry breaking from $\mathbb{Z}_2 \times \mathbb{Z}_4$ to $\mathbb{Z}_2^2 = \langle t_1 t_2, t_2^2 \rangle$.

After we apply the gauging map, this (possibly symmetry breaking) 1d $G^a \times G^b$ phase becomes a gapped domain wall in the $G$ quantum double. Comparing this domain wall with the domain wall obtained from the SPT state before projection, we notice that with the projection the stabilizers need to be updated appropriately. Nonetheless, if an anyon $a$ can pass through the former domain wall, it can also pass through the SPT-sewn domain wall. Hence if the former domain wall is invertible, the SPT-sewn domain wall is also invertible, giving rise to the same braided autoequivalence (anyon map).

To study the SPT-sewing construction of invertible domain walls, we now focus on the domain walls from gauging the projected states, which has a $G \times G$ symmetry (that can include symmetry-breaking cases).

**Theorem 2.** *In a 2d quantum double (gauge theory) of a finite group $G$, all the domain walls can be constructed from gauging 1d phases with $G \times G$ symmetry.*

*Proof.* The 1d phases under $G \times G$ symmetry are classified by a pair $(K, \nu)$, where $K \subseteq G \times G$ is the unbroken subgroup, and the 2-cocycle $[\nu] \in H^2(K, U(1))$ characterizes the SPT order with the unbroken $K$ symmetry. In Ref. [7], the domain wall Hamiltonian in a $G$ quantum double on square lattice are given, which corresponds to gauging the fixed-point $K$ SPT states broken from the $G \times G$ symmetry. In Ref. [61, 105], several examples and extensions of this results are considered. We note that, if two states are in the same phase under $G \times G$ symmetry, then there exists a shallow depth symmetric circuit to map one to the other. As a result, the two domain walls obtained after gauging are also related by a shallow depth circuit, hence of the same type. $\square$

From Theorem 2, Theorem 1 immediately follows.

Let us make a few remarks on the properties of a domain wall obtained from gauging given a pair $(K, \nu)$, focusing on an Abelian group $G$. If $K = G \times G$, the domain walls are indeed the $G \times G$ SPT-sewn domain wall, discussed in Sec. IV B. When $K = 1$, we have a state breaking all the symmetries before gauging. The corresponding domain wall is a rough boundary for both left and right bulk quantum double, as we showed for toric code model in Table I. Furthermore, when $K = G^a$, we obtain after gauging a rough boundary of the right bulk, and a boundary of the left bulk that is a $\nu$ gauged-SPT domain wall followed by a smooth boundary.

Suppose we write $G = \mathbb{Z}_{n_1} \times \mathbb{Z}_{n_2} \times \cdots \mathbb{Z}_{n_q}$, where each $\mathbb{Z}_{n_i}$ is a $p$-group, i.e., $n_i$ is a prime number to some power. We can similarly decompose the Abelian subgroup $K$ into a product of $p$-groups. The partition function of the $(K, 0)$ symmetry breaking phase is in general given by

$$\prod_s \delta(\sum_i \frac{k_{s,i}}{n_i} A_i + \frac{k'_{s,i}}{n_i} C_i \mod 1). \tag{B21}$$

It is straightforward to show that each delta function gives rise to an anyon map. Namely, an anyon $\prod_i e_i^{k_{s,i}}$ from the left bulk passing through this domain wall becomes $\prod_i e_i^{k'_{s,i}}$ in the right bulk. We denote the generator of $\mathbb{Z}_{n_i}$ as $t_i$. Elements $\{g_s \equiv \prod_i t_i^{k_{s,i}}\}$ and $\{g'_s \equiv \prod_i t_i^{k'_{s,i}}\}$ respectively generate subgroups of $G$, dubbed $H$ and $H'$ below. For an invertible $(K, 0)$ domain wall, the two groups should be isomorphic. We can write both groups as product of $p$-cyclic groups. Furthermore, we can always decompose the group $G$ as a product of $p$-cyclic groups, such that each $p$-cyclic subgroup of $H$ ($H'$) is a subgroup of a certain $p$-cyclic subgroup of $G$. Different decompositions of group $G$ might be different from our original decomposition by some isomorphisms. The two decompositions given from $H$ and $H'$ thus correspond to two linear maps of the background gauge fields $\tilde{A}_r = \sum_i k_{i,r} A_i$, and $\tilde{C}_r = \sum_k k'_{r,k} C_k$. The coefficients are given by $k_{i,r} = \Delta^a_{i,r} N_r / gcd(n_i, N_r)$ and $k'_{r,k} = \Delta^c_{r,k} N_r / gcd(n_k, N_r)$, where $\Delta$'s are integers, and $N_r$ is the order of the $r$-th cyclic subgroup of $H \subset G$. The delta function can thus be rewritten as

$$\prod_r \delta\Big(\frac{\tilde{k}_r}{n_r} \tilde{A}_r + \frac{\tilde{k}_r}{n_r} \tilde{C}_r \mod 1\Big) = \prod_r \delta\Big(\sum_{i,k} \frac{\Delta^a_{i,r}}{gcd(n_i, N_r)} A_i + \frac{\Delta^c_{r,k}}{gcd(n_k, N_r)} C_k \mod 1\Big). \tag{B22}$$

This is the general partition functions of $(K, 0)$ phases, if they give rise to invertible domain walls after gauging. We can embed the subgroup $H$ into $G^b$, such that the above partition function can always be written as in Eq. (B20) by choosing the corresponding values for parameters $\Delta_{i,r}^a$ and $\Delta_{r,k}^c$.

Now we consider a generic pair $(K, \nu)$. Given the subgroup $K$, we can construct two isomorphisms on $G$ as described above, such that the background gauge fields for $G^a$ and $G^c$ becomes $\tilde{A}_i$ and $\tilde{C}_k$. As we show above, the delta functions in the partition function become "diagonal" as on the left hand side of Eq. (B22). The gauge fields for the unbroken subgroup $K$ are essentially given by $\sum_i l_i \tilde{A}_i + l_i' \tilde{C}_i$, such that the vector formed by components $l_i$ is orthogonal to vector $(\cdots, 0, \tilde{k}_r, 0, \cdots, 0, \tilde{k}_r, 0, \cdots)$ for all $r$. Given a 2-cocycle $\nu$ for group $K$, we can write down its corresponding topological action using the $K$ gauge fields. In the end, we can always write the topological action given from $\nu$ as a bilinear form of $A_i$ and $C_j$, which is written as in Eq. (B20) by choosing the corresponding parameters.

To summarize, the key point is that all the domain walls in a $G$ quantum double can be obtained by gauging a 1d phase characterized by a pair $(K, \nu)$, where $K \subseteq G \times G$ [Theorem 2]. In the Abelian case, one can see this more explicitly; as long as $(K, \nu)$ gives rise to an invertible domain wall, we can write its partition function as in Eq. (B22). Thus, all the invertible domain walls can be obtained from gauging 1d phases with such partition functions. Recall that phases with such partition functions can be obtained by applying projections on some $G \times G \times G$ SPT states. Therefore, from SPT-sewing $G \times G \times G$ SPT states, we can construct all the invertible domain walls in an Abelian $G$ quantum double.

## Appendix C: Domain walls in $\mathbb{Z}_3$ toric code from $\mathbb{Z}_3 \times \mathbb{Z}_3$ phases

In this section, we provide an example of the domain walls in $\mathbb{Z}_3$ quantum double. The relations between the subgroups $K \in \mathbb{Z}_3 \times \mathbb{Z}_3$, the corresponding 2-cocycles, and the anyon automorphisms are displayed in Table IV. Instead of discussing the symmetry-breaking picture, we focus on constructing the invertible domain walls from SPT-sewing.

First, let us discuss the domain walls from $\mathbb{Z}_3 \times \mathbb{Z}_3$ SPT-sewing. The calculation of 2-cocycles shows that there are two nontrivial classes of $\mathbb{Z}_3 \times \mathbb{Z}_3$-SPT.

$$\mathcal{H}^2(\mathbb{Z}_3 \times \mathbb{Z}_3, U(1)) = \mathbb{Z}_3. \tag{C1}$$

They correspond to the $e \leftrightarrow m$ and $e \leftrightarrow m^{-1}$ domain walls $S_{\psi_1}$ and $S_{\psi_2}$. Here we use the former one as an example. The Hamiltonians and the gauging map are given by,

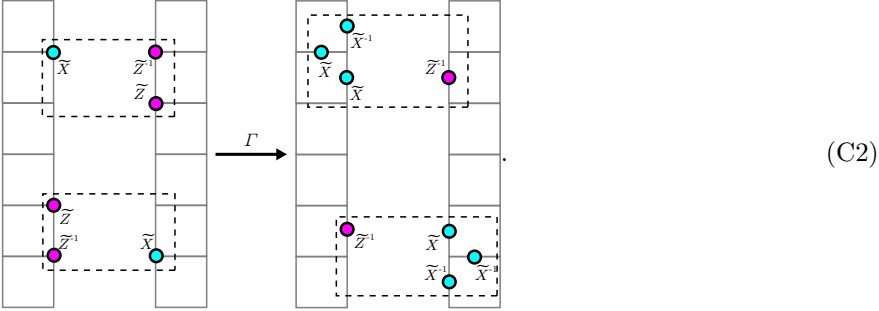

$$\tag{C2}$$

The second invertible domain wall is the trivial domain wall $S_1$, which maps every anyon to itself. The Hamiltonian of the SPT-sewing defect, the domain wall, and the gauging map between them are given as

follows,

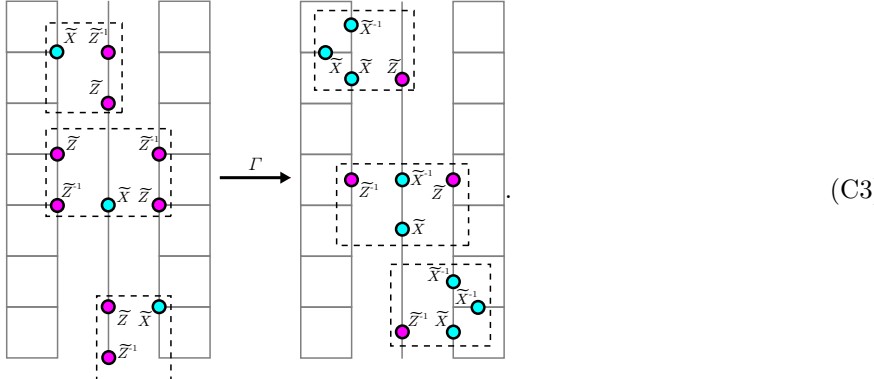

$$\text{(C3)}$$

Here the $\widetilde{X}$ and $\widetilde{Z}$ are generalized Pauli operators and they have the following actions on states.

$$\widetilde{X}|j\rangle = |j + 1 \bmod 3\rangle, \quad \widetilde{Z}|j\rangle = e^{2\pi ij/3}|j\rangle, \qquad \text{(C4)}$$

in which $j \in \{0, 1, 2\}$, $\widetilde{X}^2 = \widetilde{X}^{-1}$, and $\widetilde{Z}^2 = \widetilde{Z}^{-1}$.

One can see that after projecting (measuring) all the qutrits in the middle layer to the $\widetilde{X} = +1$ eigenstate, the symmetry breaks into the diagonal subgroup and the domain wall corresponding to $K = \mathbb{Z}_3^{diag_1}$ and $\omega_2 = 1$.

The last example is the the $e \leftrightarrow e^{-1}$ domain wall $S_{cc}$, which is also called the charge conjugation domain wall. The construction of this domain wall is given as follows,

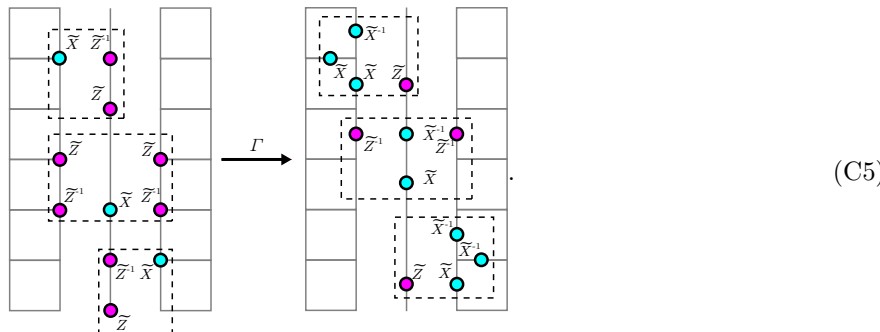

$$\text{(C5)}$$

Similarly, projecting all the qutrit states in the middle layer to the $\widetilde{X} = +1$ eigenstate gives rise to the symmetry breaking domain wall corresponding to $K = \mathbb{Z}_3^{diag_2}$ and $\omega_2 = 1$.

As we mentioned earlier, the SPT state corresponds to a topological action. For the $\mathbb{Z}_3$ quantum double, the topological actions of the corresponding SPTs are given by

$$S_1 = \int \frac{1}{3} A_1 \cup \widetilde{A}_3 + \frac{1}{3} A_2 \cup A_2, \qquad \text{(C6)}$$

$$S_2 = \int \frac{1}{3} A_1 \cup A_2 + \frac{1}{3} \widetilde{A}_2 \cup A_3, \qquad \text{(C7)}$$

$$S_3 = \int \frac{1}{3} A_1 \cup A_2 + \frac{1}{3} A_2 \cup A_3, \qquad \text{(C8)}$$

in which $\widetilde{A}_i = -A_i$. $S_1$ corresponds to the $e \leftrightarrow m$ domain wall, $S_2$ corresponds to the trivial domain wall, and $S_3$ corresponds to the charge-conjugation domain wall.

| Subgroup $K$ | 2-cocycle | Domain wall |
|---|---|---|
| $\mathbb{Z}_3^{diag_1}$ | trivial | $S_1$ |
| $\mathbb{Z}_3^{diag_2}$ | trivial | $S_{cc}$ |
| $\mathbb{Z}_3^{(1)} \times \mathbb{Z}_3^{(2)}$ | nontrivial | $S_{\psi_1}$ |
| $\mathbb{Z}_3^{(1)} \times \mathbb{Z}_3^{(2)}$ | nontrivial | $S_{\psi_2}$ |
| $\mathbb{Z}_3^{(1)} \times \mathbb{Z}_3^{(2)}$ | trivial | $S_m$ |
| $\mathbb{Z}_3^{(1)}$ | trivial | $S_{me}$ |
| $\mathbb{Z}_3^{(2)}$ | trivial | $S_{em}$ |
| $\mathbb{Z}_1$ | trivial | $S_e$ |

Table IV. Correspondence between the gapped phases under $\mathbb{Z}_3^{(1)} \times \mathbb{Z}_3^{(2)}$, and the gapped domain walls in $\mathbb{Z}_3$ quantum double. $K$ is the unbroken subgroup, and different 2-cocycles $[\omega] \in \mathcal{H}^2(K, U(1))$ correspond to different SPT phases under the unbroken $K$ symmetry.

## Appendix D: Gauging global symmetries

### 1. Gauging map

In this subsection, we provide a brief discussion of the gauging map for ground state wavefunctions and operators in systems with a finite symmetry group $G$.

Gauging is a bijective, isometric duality map from wavefunctions with global symmetries to wavefunctions with gauge (local) symmetries [61]. Consider a lattice $\mathcal{L}$ with vertices $v \in \mathcal{V}$ and edges $e \in \mathcal{E}$. We assign a $G$-qudit to each vertex and edge, where each qudit state is represented by $|g\rangle$ with $g \in G$. The gauging map $\Gamma$ between states on the vertices and states on the edges is given as follows:

$$ (D1) $$

Therefore, for a state on the square lattice, the gauging map is,

$$ (D2) $$

Consider the symmetric ground state

$$|\Psi\rangle = \left( \frac{1}{|G|} \sum_{g \in G} |g\rangle \right)^{\otimes N}, \tag{D3}$$

which is equivalent to the superposition of all the possible spin configurations. The global symmetry is given by the tensor product of left group multiplication, $\prod_v L_{+,v}^g$. One can check that after gauging, this global symmetry operator maps to identity.

$$\prod_v L_{+,v}^g \xrightarrow{\Gamma} I \tag{D4}$$

The local Hamiltonian is given by the right group multiplication. We have

$$H = -\sum_v L_{-,v}^g. \tag{D5}$$

After gauging, the $L^g_{-,v}$ operator becomes the star term of the quantum double model. We have

$$L^g_{-,v} \xrightarrow{\Gamma} A^g_v. \tag{D7}$$

The zero-flux condition is automatically guaranteed by the gauging map. For example, for the state in Eq. (D2), we have,

$$(g_3^{-1}g_4)^{-1}(g_2^{-1}g_3)^{-1}(g_2^{-1}g_1)(g_1^{-1}g_4) = 1. \tag{D8}$$

Therefore, after gauging the state in Eq. (D3), we can get the ground state wavefunction of $G$ quantum double with the following Hamiltonian [1],

$$H = -\sum_v A_v - \sum_p B_p, \tag{D9}$$

in which

$$A_v = \frac{1}{|G|}\sum_g A^g_v, \quad B_p = \sum_{g_1 g_2 \ldots g_k = 1}\prod_{i=1}^{k} T^{g_i}_{\pm, e \in \partial p}. \tag{D10}$$

As demonstrated in this example, gauging the trivial SPT phase gives rise to the quantum double $D(G)$. More generally, gauging the nontrivial SPT phases give rise to the twisted quantum doubles $D^\omega(G)$ [91, 106].

## 2. Gauging $S_3$ global symmetry

In this subsection, we take the $S_3$ symmetry as an example to show how to sequentially gauge the $\mathbb{Z}_3$ and $\mathbb{Z}_2$ subgroups, and finally obtain the ground state and Hamiltonian of the $S_3$ quantum double.

The $S_3$ group is the semidirect product of $\mathbb{Z}_3$ and $\mathbb{Z}_2$, $S_3 = \mathbb{Z}_3 \rtimes \mathbb{Z}_2$. It is a non-Abelian group consisting of six elements, $\{e, c, c^2, t, ct, c^2t\}$, where $e$ is the identity element, $c^3 = t^2 = e$, $tct = c^2$ and $tc^2t = c$. Since $S_3$ has a semidirect product structure, we can choose a basis consisting of product states over $\mathbb{C}^3 \otimes \mathbb{C}^2$:

$$|g\rangle \to |c^n t^q\rangle = |c^n\rangle \otimes |t^q\rangle, \tag{D11}$$

where $n \in \{0, 1, 2\}$, $q \in \{0, 1\}$, $g \in S_3$, $c^n \in \mathbb{Z}_3$ and $t^q \in \mathbb{Z}_2$. The left and right multiplications are

$$\begin{aligned}
L^c_+|c^n t^q\rangle &= |c^{n+1}t^q\rangle, \quad L^c_-|c^n t^q\rangle = |c^{n-(-1)^q}t^q\rangle \\
L^t_+|c^n t^q\rangle &= |c^{-n}t^{q+1}\rangle, \quad L^c_-|c^n t^q\rangle = |c^n t^{q-1}\rangle
\end{aligned} \tag{D12}$$

We can also define the generalized Pauli $Z$ operators as linear combinations of $T^g_+$ operators,

$$\begin{aligned}
\widetilde{Z}_L &= \sum_{j,k} e^{2\pi ij/3} T^{c^j t^k}_+, \\
\widetilde{Z}_R &= \sum_{j,k} e^{2\pi ij/3} T^{t^k c^j}_+, \\
Z &= \sum_{j,k} (-1)^k T^{c^j t^k}_+.
\end{aligned} \tag{D13}$$

Consider $S_3$ qudits are assigned to the vertices of a square lattice, and each qudit is in the $+1$-eigenstate of $L_-^g$ operator. A spin configuration can be depicted as follows,

$$
\begin{array}{c}
n_1 q_1 \\
n_2 q_2 \quad n_0 q_0 \quad n_4 q_4 \\
n_3 q_3
\end{array}
\tag{D14}
$$

For the sake of simplicity, we denote the $S_3$ state in terms of $n_i q_i$ instead of $c^{n_i} t^{q_i}$.

Let us first gauge the $\mathbb{Z}_3$ symmetry. After gauging, the ground state becomes the following,

$$
\begin{array}{c}
q_1 \\
n_0 - n_1 \\
q_2 \quad n_2 - n_0 \quad n_0 - n_4 \quad q_4 \\
q_0 \\
n_3 - n_0 \\
q_3
\end{array}
\tag{D15}
$$

where the $\mathbb{Z}_2$ qubits are on the vertices, and the $\mathbb{Z}_3$ qutrits are on the edges. By applying the same method as Eq. (D6), one can find the local stabilizers become,

$$
L_{-,v}^c \xrightarrow{\Gamma_{\mathbb{Z}_3}}
\left(
\begin{array}{c}
L_-^c \\
L_+^c \; q_0 \; L_-^c \\
v \\
L_+^c
\end{array}
\right)^{Z_0}
, \quad
L_{-,v}^t \xrightarrow{\Gamma_{\mathbb{Z}_3}} L_{-,v}^t,
\tag{D16}
$$

in which $Z_0 = (-1)^{q_0}$ corresponds to the eigenvalue of the Pauli $Z$ operator on the state $|q_0\rangle$. These are the stabilizers of the symmetry-enriched topological (SET) phases. Then we can further gauge the remaining $\mathbb{Z}_2$ symmetry. We get

$$
\left(
\begin{array}{c}
L_-^c \\
L_+^c \; q_0 \; L_-^c \\
v \\
L_+^c
\end{array}
\right)^{Z_0}
\xrightarrow{\Gamma_{\mathbb{Z}_2}}
\begin{array}{c}
L_-^c \\
L_+^c \quad L_-^c \\
v \\
L_+^c
\end{array}
, \quad
L_{-,v}^t \xrightarrow{\Gamma_{\mathbb{Z}_2}}
\begin{array}{c}
L_-^t \\
L_+^t \quad L_-^t \\
v \\
L_+^t
\end{array}
,
\tag{D17}
$$

which are the star terms of $S_3$ quantum double.

In the presence of the smooth boundary, after sequentially gauge the $\mathbb{Z}_3$ and $\mathbb{Z}_2$ symmetry, the generalized Ising stabilizers maps to the following:

$$
\begin{array}{c}
L_-^c \\
v
\end{array}
\xrightarrow{\Gamma_{S_3}}
\begin{array}{c}
L_-^c \\
L_+^c \quad L_-^c \\
v
\end{array}
, \quad
\begin{array}{c}
L_-^t \\
v
\end{array}
\xrightarrow{\Gamma_{S_3}}
\begin{array}{c}
L_-^t \\
L_+^t \quad L_-^t \\
v
\end{array}
.
\tag{D18}
$$

### 3. Ribbon operators of $S_3$ quantum double

Given a ribbon $\xi$, one can define a ribbon operator $F_\xi^{(h,g)}$ such that it commutes with all the stabilizers except for the stabilizers at the endpoints [1]. The ribbon operator of quantum doubles is defined in Fig. 4

and the multiplication and comultiplication rules are provided in Eq. (39) and Eq. (40).

In general, a ribbon operator that create anyonic excitation labeled by $(C, R)$ is given by [1, 30, 62]

$$F_\xi^{(C,R);(\mathbf{u},\mathbf{v})} := \frac{d_\pi}{|E(C)|} \sum_{k \in E(C)} \left(\Gamma_\pi^{-1}(k)\right)_{j,j'} F_\xi^{(c_i^{-1}, p_i k p_{i'}^{-1})}, \tag{D19}$$

in which $E(C)$ is the centralizer group of an element $r_C \in C$, $d_\pi$ is the dimension of the irreducible representation of the centralizer group. Here $\mathbf{u} = (i, j)$ and $\mathbf{v} = (i', j')$, where $i, i' \in \{1, \ldots, |C|\}$ and $j, j' \in \{1, \ldots, d_\pi\}$ are the indices for the elements of the conjugacy class and the matrix row/column indices respectively. $\Gamma_\pi(k)$ is the $d_\pi$ dimensional irreducible representation of $k \in E(C)$. Lastly, we choose $\{p_i\}_{i=1}^{|C|} \in G$ such that $c_i = p_i r_C p_i^{-1} \in C$.

We can write down all the eight ribbon operators for anyons $A$ to $H$ of $S_3$ quantum double. (We denote the third root of unity as $\omega$ below.) The ribbon operators for the three pure electric charges are given by:

$$F_\xi^{([e],1)} = \frac{1}{6} \left( F_\xi^{e,e} + F_\xi^{e,c} + F_\xi^{e,c^2} + F_\xi^{e,t} + F_\xi^{e,tc} + F_\xi^{e,tc^2} \right), \tag{D20}$$

$$F_\xi^{([e],-1)} = \frac{1}{6} \left( F_\xi^{e,e} + F_\xi^{e,c} + F_\xi^{e,c^2} - F_\xi^{e,t} - F_\xi^{e,tc} - F_\xi^{e,tc^2} \right), \tag{D21}$$

$$F_\xi^{([e],\pi)} = \frac{1}{3} \left[ \begin{pmatrix} 1 & 0 \\ 0 & 1 \end{pmatrix} F_\xi^{e,e} + \begin{pmatrix} \bar\omega & 0 \\ 0 & \omega \end{pmatrix} F_\xi^{e,c} + \begin{pmatrix} \omega & 0 \\ 0 & \bar\omega \end{pmatrix} F_\xi^{e,c^2} + \begin{pmatrix} 0 & 1 \\ 1 & 0 \end{pmatrix} F_\xi^{e,t} + \begin{pmatrix} 0 & \omega \\ \bar\omega & 0 \end{pmatrix} F_\xi^{e,tc} + \begin{pmatrix} 0 & \bar\omega \\ \omega & 0 \end{pmatrix} F_\xi^{e,tc^2} \right], \tag{D22}$$

corresponding to the ribbon operators for the $A, B$, and $C$ anyons.

The ribbon operators for anyons with $[t]$ flux are given by:

$$F_\xi^{([t],1)} = \frac{1}{2} \left[ \begin{pmatrix} F_\xi^{t,e} & F_\xi^{t,c^2} & F_\xi^{t,c} \\ F_\xi^{tc,c} & F_\xi^{tc,e} & F_\xi^{tc,c^2} \\ F_\xi^{tc^2,c^2} & F_\xi^{tc^2,c} & F_\xi^{tc^2,e} \end{pmatrix} + \begin{pmatrix} F_\xi^{t,t} & F_\xi^{t,tc^2} & F_\xi^{t,tc} \\ F_\xi^{tc,tc^2} & F_\xi^{tc,tc} & F_\xi^{tc,t} \\ F_\xi^{tc^2,tc} & F_\xi^{tc^2,t} & F_\xi^{tc^2,tc^2} \end{pmatrix} \right], \tag{D23}$$

$$F_\xi^{([t],-1)} = \frac{1}{2} \left[ \begin{pmatrix} F_\xi^{t,e} & F_\xi^{t,c^2} & F_\xi^{t,c} \\ F_\xi^{tc,c} & F_\xi^{tc,e} & F_\xi^{tc,c^2} \\ F_\xi^{tc^2,c^2} & F_\xi^{tc^2,c} & F_\xi^{tc^2,e} \end{pmatrix} - \begin{pmatrix} F_\xi^{t,t} & F_\xi^{t,tc^2} & F_\xi^{t,tc} \\ F_\xi^{tc,tc^2} & F_\xi^{tc,tc} & F_\xi^{tc,t} \\ F_\xi^{tc^2,tc} & F_\xi^{tc^2,t} & F_\xi^{tc^2,tc^2} \end{pmatrix} \right], \tag{D24}$$

for the $D$ and $E$ anyons.

Finally, we have the ribbon operators for anyons with $[c]$ flux:

$$F_\xi^{([c],1)} = \frac{1}{3} \left[ \begin{pmatrix} F_\xi^{c^2,e} & F_\xi^{c^2,t} \\ F_\xi^{c,t} & F_\xi^{c,e} \end{pmatrix} + \begin{pmatrix} F_\xi^{c^2,c} & F_\xi^{c^2,tc^2} \\ F_\xi^{c,tc} & F_\xi^{c,c^2} \end{pmatrix} + \begin{pmatrix} F_\xi^{c^2,c^2} & F_\xi^{c^2,tc} \\ F_\xi^{c,tc^2} & F_\xi^{c,c} \end{pmatrix} \right], \tag{D25}$$

$$F_\xi^{([c],\omega)} = \frac{1}{3} \left[ \begin{pmatrix} F_\xi^{c^2,e} & F_\xi^{c^2,t} \\ F_\xi^{c,t} & F_\xi^{c,e} \end{pmatrix} + \bar\omega \begin{pmatrix} F_\xi^{c^2,c} & F_\xi^{c^2,tc^2} \\ F_\xi^{c,tc} & F_\xi^{c,c^2} \end{pmatrix} + \omega \begin{pmatrix} F_\xi^{c^2,c^2} & F_\xi^{c^2,tc} \\ F_\xi^{c,tc^2} & F_\xi^{c,c} \end{pmatrix} \right], \tag{D26}$$

$$F_\xi^{([c],\bar\omega)} = \frac{1}{3} \left[ \begin{pmatrix} F_\xi^{c^2,e} & F_\xi^{c^2,t} \\ F_\xi^{c,t} & F_\xi^{c,e} \end{pmatrix} + \omega \begin{pmatrix} F_\xi^{c^2,c} & F_\xi^{c^2,tc^2} \\ F_\xi^{c,tc} & F_\xi^{c,c^2} \end{pmatrix} + \bar\omega \begin{pmatrix} F_\xi^{c^2,c^2} & F_\xi^{c^2,tc} \\ F_\xi^{c,tc^2} & F_\xi^{c,c} \end{pmatrix} \right] \tag{D27}$$

for the $F, G$, and $H$ anyon.

The comultiplication rule of anyonic ribbon operators can be obtained by matrix product of the internal states up to a constant factor for normalization.

$$F_\xi^{(C,R)} = \frac{|E(C)|}{d_\pi} F_{\xi_1}^{(C,R)} F_{\xi_2}^{(C,R)}, \tag{D28}$$

in which $\xi = \xi_1 \xi_2$.

Here we show the comutipilication rules of $C$ and $F$ anyons as examples. The $C$ anyon ribbon operator is given by,

$$F_\xi^{([e],\pi)} = \frac{1}{3} \begin{pmatrix} F_\xi^{e,e} + \bar\omega F_\xi^{e,c} + \omega F_\xi^{e,c^2} & F_\xi^{e,t} + \omega F_\xi^{e,tc} + \bar\omega F_\xi^{e,tc^2} \\ F_\xi^{e,t} + \bar\omega F_\xi^{e,tc} + \omega F_\xi^{e,tc^2} & F_\xi^{e,e} + \omega F_\xi^{e,c} + \bar\omega F_\xi^{e,c^2} \end{pmatrix} \tag{D29}$$

By applying the comultiplication rule Eq. (40), the $\left(F_\xi^{([e],\pi)}\right)_{11}$ term can be equivalently written as

$$
\begin{aligned}
\frac{1}{3}\left(F_\xi^{e,e} + \bar{\omega}F_\xi^{e,c} + \omega F_\xi^{e,c^2}\right) &= \frac{1}{3}\sum_{k\in S_3}\left(F_{\xi_1}^{e,k}\otimes F_{\xi_2}^{e,k^{-1}} + \bar{\omega}F_{\xi_1}^{e,k}\otimes F_{\xi_2}^{e,k^{-1}c} + \omega F_{\xi_1}^{e,k}\otimes F_{\xi_2}^{e,k^{-1}c^2}\right)\\
&= \frac{1}{3}\left(F_{\xi_1}^{e,e} + \bar{\omega}F_{\xi_1}^{e,c} + \omega F_{\xi_1}^{e,c^2}\right)\otimes\left(F_{\xi_2}^{e,e} + \bar{\omega}F_{\xi_2}^{e,c} + \omega F_{\xi_2}^{e,c^2}\right)\\
&+ \frac{1}{3}\left(F_{\xi_1}^{e,t} + \omega F_{\xi_!}^{e,tc} + \bar{\omega}F_{\xi_1}^{e,tc^2}\right)\otimes\left(F_{\xi_2}^{e,t} + \bar{\omega}F_{\xi_2}^{e,tc} + \omega F_{\xi_2}^{e,tc^2}\right)\\
&= 3\left(F_{\xi_1}^{([e],\pi)}\right)_{11}\left(F_{\xi_2}^{([e],\pi)}\right)_{11} + 3\left(F_{\xi_1}^{([e],\pi)}\right)_{12}\left(F_{\xi_2}^{([e],\pi)}\right)_{21}.
\end{aligned}
\tag{D30}
$$

The $\left(F_\xi^{([e],\pi)}\right)_{12}$ term can be written as

$$
\begin{aligned}
\frac{1}{3}\left(F_\xi^{e,t} + \omega F_\xi^{e,tc} + \bar{\omega}F_\xi^{e,tc^2}\right) &= \frac{1}{3}\sum_{k\in S_3}\left(F_{\xi_1}^{e,k}\otimes F_{\xi_2}^{e,k^{-1}t} + \omega F_{\xi_1}^{e,k}\otimes F_{\xi_2}^{e,k^{-1}tc} + \bar{\omega}F_{\xi_1}^{e,k}\otimes F_{\xi_2}^{e,k^{-1}tc^2}\right)\\
&= \frac{1}{3}\left(F_{\xi_1}^{e,e} + \bar{\omega}F_{\xi_1}^{e,c} + \omega F_{\xi_1}^{e,c^2}\right)\otimes\left(F_{\xi_2}^{e,t} + \omega F_{\xi_2}^{e,tc} + \bar{\omega}F_{\xi_2}^{e,tc^2}\right)\\
&+ \frac{1}{3}\left(F_{\xi_1}^{e,t} + \omega F_{\xi_1}^{e,tc} + \bar{\omega}F_{\xi_1}^{e,tc^2}\right)\otimes\left(F_{\xi_2}^{e,e} + \omega F_{\xi_2}^{e,c} + \bar{\omega}F_{\xi_2}^{e,c^2}\right)\\
&= 3\left(F_{\xi_1}^{([e],\pi)}\right)_{11}\left(F_{\xi_2}^{([e],\pi)}\right)_{12} + 3\left(F_{\xi_1}^{([e],\pi)}\right)_{12}\left(F_{\xi_2}^{([e],\pi)}\right)_{22}.
\end{aligned}
\tag{D31}
$$

The decomposition of the other terms can be obtained similarly. Finally, we have

$$
F_\xi^{([e],\pi)} = 3F_{\xi_1}^{([e],\pi)}F_{\xi_2}^{([e],\pi)}
\tag{D32}
$$

Next, let us check the $F$ ribbon operator. We have,

$$
F_\xi^{([c],1)} = \frac{1}{3}\begin{pmatrix} F_\xi^{c^2,e} + F_\xi^{c^2,c} + F_\xi^{c^2,c^2} & F_\xi^{c^2,t} + F_\xi^{c^2,tc^2} + F_\xi^{c^2,tc}\\ F_\xi^{c,t} + F_\xi^{c,tc} + F_\xi^{c,tc^2} & F_\xi^{c,e} + F_\xi^{c,c^2} + F_\xi^{c,c} \end{pmatrix}
\tag{D33}
$$

The first term $\left(F_\xi^{([c],1)}\right)_{11}$ can be written as

$$
\begin{aligned}
\frac{1}{3}\left(F_\xi^{c^2,e} + F_\xi^{c^2,c} + F_\xi^{c^2,c^2}\right) &= \frac{1}{3}\sum_{k\in S_3}\left(F_{\xi_1}^{c^2,k}\otimes F_{\xi_2}^{k^{-1}c^2k,k^{-1}} + F_{\xi_1}^{c^2,k}\otimes F_{\xi_2}^{k^{-1}c^2k,k^{-1}c} + F_{\xi_1}^{c^2,k}\otimes F_{\xi_2}^{k^{-1}c^2k,k^{-1}c^2}\right)\\
&= \frac{1}{3}\left(F_{\xi_1}^{c^2,e} + F_{\xi_1}^{c^2,c} + F_{\xi_1}^{c^2,c^2}\right)\otimes\left(F_{\xi_2}^{c^2,e} + F_{\xi_2}^{c^2,c} + F_{\xi_2}^{c^2,c^2}\right)\\
&+ \frac{1}{3}\left(F_{\xi_1}^{c^2,t} + F_{\xi_1}^{c^2,tc^2} + F_{\xi_1}^{c^2,tc}\right)\otimes\left(F_{\xi_2}^{c,t} + F_{\xi_2}^{c,tc} + F_{\xi_2}^{c,tc^2}\right)
\end{aligned}
\tag{D34}
$$

The decompostion of the other terms can be obtained similarly. We conclude that

$$
F_\xi^{([c],1)} = 3F_{\xi_1}^{([c],1)}F_{\xi_2}^{([c],1)}
\tag{D35}
$$

## Appendix E: Domain walls in $S_3$ quantum double

In this section, we provide more details about the construction of the domain walls introduced in Sec. V. The structure of this section is as follows: In Appendix E 1, we will construct the $S_3 \times S_3$ SPT-sewn domain wall, which corresponds to the $B \leftrightarrow D$ domain wall discussed in Sec. V C. In Appendix E 2, we will construct the $C \leftrightarrow F$ domain wall. The corresponding $(\mathbb{Z}_3 \times \mathbb{Z}_3) \rtimes \mathbb{Z}_2$ SPT can be obtained from measuring out the middle layer of the $S_3 \times \text{Rep}(S_3) \times S_3$ SPT-sewn domain wall as discussed in Sec. V D.

## 1. $S_3 \times S_3$ **SPT-sewn domain wall**

Consider we have two $S_3$ Ising models with smooth boundaries in the symmetric phase. We can sew the two smooth boundaries together by applying the SPT entangler, which corresponds to the nontrivial 2-cocycles $\omega_2 \in H^2(S_3 \times S_3, U(1))$. We have

$$H^2(S_3 \times S_3, U(1)) = \mathbb{Z}_2, \tag{E1}$$

The nontrivial 2-cocycle is given by

$$\omega_2\left((t^{i_1}c^{j_1}, t^{i_2}c^{j_2}), (t^{i'_1}c^{j'_1}, t^{i'_2}c^{i'_2})\right) = (-1)^{i_1 j'_2}. \tag{E2}$$

By scrutinizing the exact form of the 2-cocycle, we find the nontrivial 2-cocycle of $S_3 \times S_3$ in fact comes form the $\mathbb{Z}_2 \times \mathbb{Z}_2$ subgroup. The 2-cocile of the nontrivial $\mathbb{Z}_2 \times \mathbb{Z}_2$ SPT is given by

$$\nu_2\left((t^{i_1}, t^{i_2}), (t^{i'_1}, t^{i'_2})\right) = (-1)^{i_1 i'_2}, \tag{E3}$$

which is exactly the same as Eq. (E2).

Therefore, the $B \leftrightarrow D$ domain wall can be obtained by sewing the $\mathbb{Z}_2 \times \mathbb{Z}_2$ subgroup of the smooth boundaries with $S_3 \times S_3$ symmetry and gauging the global $S_3 \times S_3$ symmetry. The SPT Hamiltonian and the gauging map are shown as follows.

For the $\mathbb{Z}_2$ layer, we have

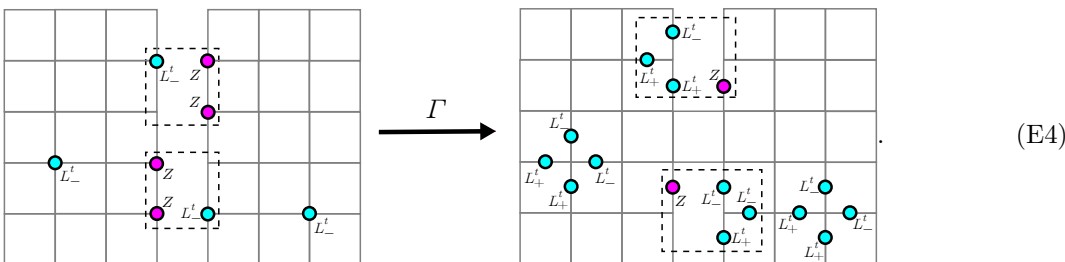

$$\tag{E4}$$

For the $\mathbb{Z}_3$ layer, we have

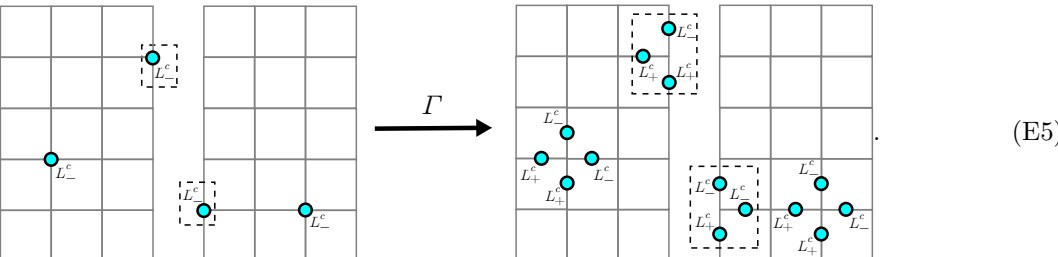

$$\tag{E5}$$

To show this domain wall condenses $B$ and $D$ anyons, we can show that $F_\xi^{([e],-1)} \otimes F_\xi^{([t],1)}$ and $F_\xi^{([t],1)} \otimes F_\xi^{([e],-1)}$ commute with the stabilizers on the domain wall. The nontrivial stabilizers on the domain wall are the following:

$$\tag{E6}$$

For the simplicity of discussion, we consider a ribbon $\xi$ passing across the domain wall. The left part of the ribbon is labeled as $\xi_L$, while the right part of the ribbon is labeled as $\xi_R$. We denote $\xi = \xi_L \xi_R$.

By definition, $A$ anyon (the vacuum) can go through the domain wall. Pairs of $B$ anyons in the left bulk are created by the ribbon operator $F_{\xi_L}^{([\mathrm{e}],-1)}$, which is a string of Pauli $Z$ operators on qubits. Since the Pauli $Z$ string attaches to the domain wall from the left bulk, the first term in Eq. (E6) is violated. One can further apply the ribbon operator of the $D$ anyon on the other side, namely, $F_{\xi_R}^{([\mathrm{t}],1)}$. We find that each entry of $F_{\xi_R}^{([\mathrm{t}],1)}$ contains a $\mathbb{Z}_2$ shift operator at the endpoint attached to the domain wall, which anti-commutes with the Pauli $Z$ operator in the first term in Eq. (E6) as well. Therefore, the product $F_{\xi_L}^{([\mathrm{e}],-1)}F_{\xi_R}^{([\mathrm{t}],1)}$ commutes with the stabilizers on the domain wall. One can also verify $F_{\xi_L}^{([\mathrm{t}],1)}F_{\xi_R}^{([\mathrm{e}],-1)}$ commutes with the domain wall stabilizers as well.

Now consider the scenario where two $E$ anyon ribbon operators are attached to the same site on the domain wall, $F_{\xi_L}^{([\mathrm{t}],-1)}F_{\xi_R}^{([\mathrm{t}],-1)}$. We have the following fusion rule [7]:

$$B \times D = E \tag{E7}$$

For the ribbon operator, according to the multiplication rule in Eq. (39), we have

$$F_\xi^{([\mathrm{e}],-1)}F_\xi^{([\mathrm{t}],1)} = F_\xi^{([\mathrm{t}],-1)}. \tag{E8}$$

One can check it by substituting Eq. (D21) and Eq. (D23) into the above equation.

Therefore, we can consider aligning the ribbon operators for the $B$ and $D$ anyons together on each side of the domain wall. Specifically, we have the ribbon operators $F_{\xi_L}^{([\mathrm{e}],-1)}F_{\xi_R}^{([\mathrm{t}],1)}$, and $F_{\xi_L}^{([\mathrm{t}],1)}F_{\xi_R}^{([\mathrm{e}],-1)}$. By multiplying them together, we obtain $F_{\xi_L}^{([\mathrm{t}],-1)}F_{\xi_R}^{([\mathrm{t}],-1)}$. As we have shown earlier this subsection, both $F_{\xi_L}^{([\mathrm{e}],-1)}F_{\xi_R}^{([\mathrm{t}],1)}$ and $F_{\xi_L}^{([\mathrm{t}],1)}F_{\xi_R}^{([\mathrm{e}],-1)}$ commute independently with the domain wall stabilizers. Therefore, their multiplication also commutes with the stabilizers.

We conclude that the $S_3 \times S_3$ SPT-sewn domain wall is invertible for $A$, $B$, $D$, and $E$ anyons. For all the other anyons, this domain wall is not traversable.

### 2.   $C \leftrightarrow F$ domain wall

In this subsection, we construct the $C \leftrightarrow F$ domain wall by gauging the $(\mathbb{Z}_3 \times \mathbb{Z}_3) \rtimes \mathbb{Z}_2$ SPT on a codimension-1 submanifold of the 2d $S_3$ Ising model. The $(\mathbb{Z}_3 \times \mathbb{Z}_3) \rtimes \mathbb{Z}_2$ SPT is calculated in Appendix A 5. Unlike in Appendix A 5, where we assume there is one qubit per site, here we assume there are two qubits per site. However, since the $\mathbb{Z}_2 \times \mathbb{Z}_2$ group breaks into the diagonal subgroup, the two setups are equivalent up to isometry. Therefore, the qubit state at each site can be written as $|t_1^{k_l}t_2^{k_l}\rangle$, where $|t_1^{k_l}\rangle$ is the qubit state above, while $|t_2^{k_l}\rangle$ is the qubit state below.

After gauging the $\mathbb{Z}_3$ symmetry, the $\mathbb{Z}_3$ stabilizers on the domain wall becomes the following,

$$. \tag{E9}$$

And the $\mathbb{Z}_2$ operators are unchanged. The ground state of the above Hamiltonian corresponds to the domain wall Hamiltonian of a symmetry-enriched topological (SET) phase. Specifically, it corresponds to the $\mathbb{Z}_3$ quantum double enriched by the $\mathbb{Z}_2$ charge conjugation symmetry. The ground state wavefunction can be written as follows:

$$|\Psi\rangle_{\text{SET}} \propto \sum_{\{n_i\},\{q_j\}} |c^{s(\triangle)(n_{i'}-n_i)}\rangle \otimes |t^{q_j}\rangle, \tag{E10}$$

where $i$ and $i'$ are neighboring vertices, $s(\triangle) = \pm 1$ depending on the orientation of the edge, $n_i \in \{0,1,2\}$, and $q_j \in \{0,1\}$. Along the domain wall, we require an additional constraint that the top and bottom qubit states are the same in the group basis.

Then we further gauge the remaining $\mathbb{Z}_2$ symmetry. We use the first stabilizer in Eq. (E9) as an example. We have

In which $j_{l+1,l} = j_{l+1} - j_l$, $i_l, j_l \in \{0,1,2\}$, $k_l \in \{0,1\}$, and $|\Psi\rangle_{\text{QD}}$ is the ground state wavefunction for the $S_3$ quantum double with the above domain wall. Similarly, one can get the other stabilizers. We summarize

all the stabilizers below.

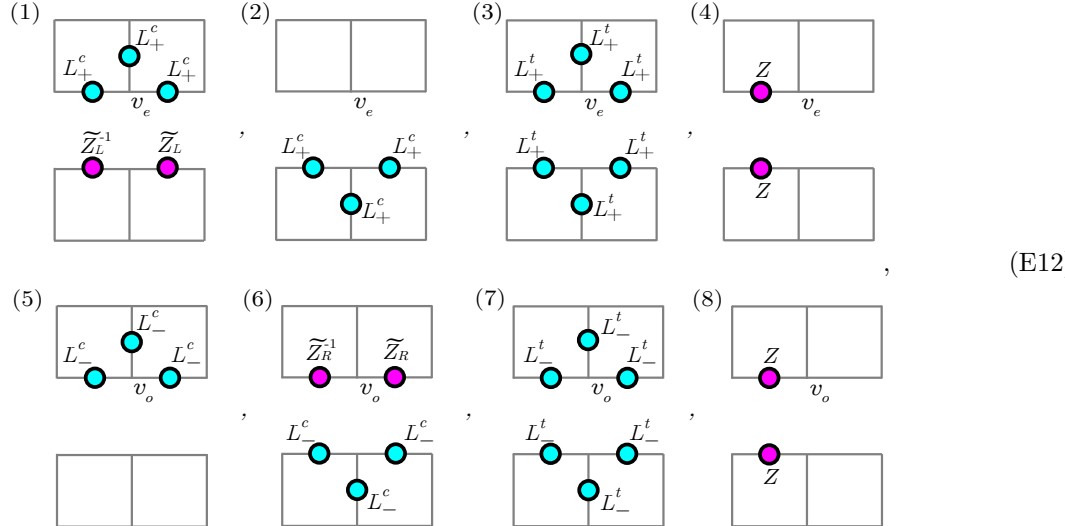

$$F_\xi^{CF} = 3F_{\xi_1}^C F_{\xi_2}^F, \tag{E13}$$

in which $\widetilde{Z}_L$, $\widetilde{Z}_R$, and $Z$ are defined in Eq. (35). Although Eq. (E11)(1) and (3) do not commute with each other, conjugating (1) by (3) yields, the inverse of (1). Thus they are frustration-free. The same conclusion can be applied between (5) and (7). If needed, these can be turned into a commuting Hamiltonian by summing the terms over $c$ and $t$.

Following the discussion in Appendix D 3, The $CF$ ribbon operator can be written as follows,

$$F_\xi^{CF} = 3F_{\xi_1}^C F_{\xi_2}^F, \tag{E13}$$

in which $\xi = \xi_1\xi_2$ and the intersection of $\xi_1$ and $\xi_2$ is at the domain wall; see Eq. (E14) for an illustration.

There are two types of stabilizers on the domain wall that connect the two bulks: Eq. (E12)(1) and Eq. (E12)(6), and Eq. (E12)(3) and Eq. (E12)(7). One can check that all the other stabilizers commute trivially with the $CF$ ribbon operator. We use Eq. (E12)(1) and Eq. (E12)(3) as nontrivial examples to illustrate that the $CF$ ribbon operator commutes with the stabilizers on the domain wall. The commutation for the remaining cases can be shown similarly.

The branching structure and labels of ribbons and qudits are displayed below, as well as the ribbon we choose.

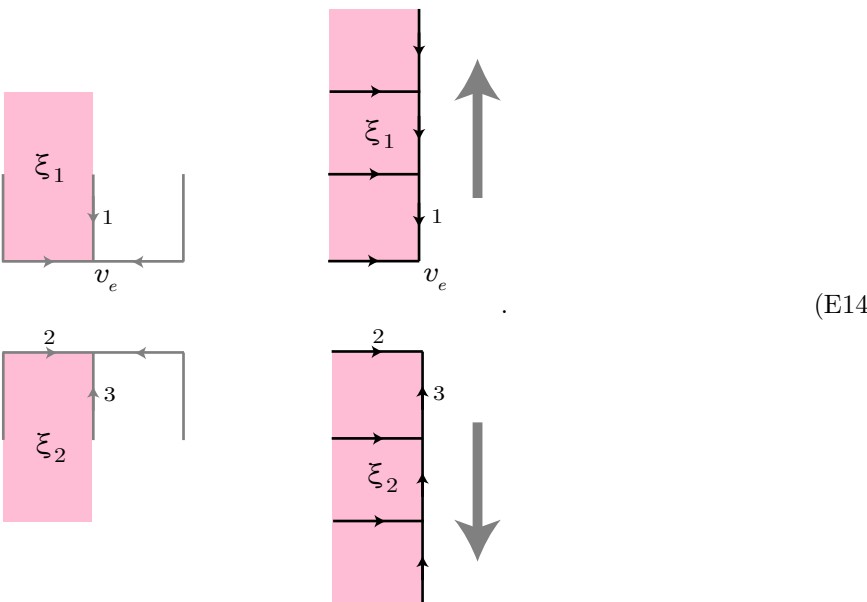

$$\tag{E14}$$

The arrow in the above equation shows the direction of the anyons.

We place the $C$ ribbon operator on $\xi_1$ and $F$ ribbon operator on $\xi_2$. Note the ribbon operator has a nontrivial overlap with domain wall stabilizers only on qudit $1, 2$, and $3$. Thus it suffices to check the commutation relation with respect to a "short" ribbon operator supported on these sites; the commutation relations for the longer ribbon operators then follow from the comultiplication rule [Appendix D 3].

The nontrivial part of the calculation can be summarized as follows. The relevant matrix entries of Eq. (E13) are

$$
\begin{aligned}
3\left(F_\xi^{CF}\right)_{11} &= \left(F_{\xi_1}^{e,e} + \bar\omega F_{\xi_1}^{e,c} + \omega F_{\xi_1}^{e,c^2}\right) \otimes \left(F_{\xi_2}^{c^2,e} + F_{\xi_2}^{c^2,c} + F_{\xi_2}^{c^2,c^2}\right) \\
&\quad + \left(F_{\xi_1}^{e,t} + \omega F_{\xi_1}^{e,tc} + \bar\omega F_{\xi,1}^{e,tc^2}\right) \otimes \left(F_{\xi_2}^{c,t} + F_{\xi_2}^{c,tc} + F_{\xi_2}^{c,tc^2}\right), \\
3\left(F_\xi^{CF}\right)_{12} &= \left(F_{\xi_1}^{e,e} + \bar\omega F_{\xi_1}^{e,c} + \omega F_{\xi_1}^{e,c^2}\right) \otimes \left(F_{\xi_2}^{c^2,t} + F_{\xi_2}^{c^2,tc} + F_{\xi_2}^{c^2,tc^2}\right) \\
&\quad + \left(F_{\xi_1}^{e,t} + \bar\omega F_{\xi_1}^{e,tc^2} + \omega F_{\xi_1}^{e,tc}\right) \otimes \left(F_{\xi_2}^{c,e} + F_{\xi_2}^{c,c} + F_{\xi_2}^{c,c^2}\right).
\end{aligned}
\tag{E15}
$$

Before we proceed, we point out a subtlety on the orientation of the ribbon operator. If we rotate Eq. (E14) counterclockwise so that it becomes horizontal, and compare it with the ribbon operator defined in Fig. 4, we find the orientation of the horizontal line reverses in the middle. If the reversed orientation is applied everywhere, the delta function defined in Fig. 4 would become $\delta_{g^{-1}, x_1 x_2 x_3}$.

First, let us check the commutation relation between Eq. (E12)(1) and the first term in Eq. (E15). Because $F_{\xi_1}^{(e,g)}$ projects the state on the qudit 1 to $g^{-1}$, we get

$$
L_{+,1}^c F_{\xi_1}^{(e,g)} = F_{\xi_1}^{(e,gc^{-1})} L_{+,1}^c,
\tag{E16}
$$

Furthermore, we get

$$
\widetilde{Z}_{L,2}^{-1} F_{\xi_2}^{(c^2,g)} = \omega F_{\xi_2}^{(c^2,g)} \widetilde{Z}_{L,2}^{-1}, \quad \widetilde{Z}_{L,2}^{-1} F_{\xi_2}^{(c,g)} = \bar\omega F_{\xi_2}^{(c,g)} \widetilde{Z}_{L,2}^{-1},
\tag{E17}
$$

From these relations, it follows that $\left(F_\xi^{CF}\right)_{11}$ and $\left(F_\xi^{CF}\right)_{12}$ are invariant under the conjugation by Eq. (E12)(1). Similarly, one can show that the other matrix elements commute with Eq. (E12)(1) as well.

The next nontrivial commutation relation is between Eq. (E12)(3) and $F_\xi^{CF}$. To that end, we note

$$
L_{+,1}^t F_{\xi_1}^{e,g} = F_{\xi_1}^{e,gt} L_{+,1}^t,
\tag{E18}
$$

and

$$
\left(L_{+,2}^t \otimes L_{+,3}^t\right) F_{\xi_2}^{c^2,g} = F_{\xi_2}^{c,gt} \left(L_{+,2}^t \otimes L_{+,3}^t\right), \quad \left(L_{+,2}^t \otimes L_{+,3}^t\right) F_{\xi_2}^{c,g} = F_{\xi_2}^{c^2,gt} \left(L_{+,2}^t \otimes L_{+,3}^t\right).
\tag{E19}
$$

Thus conjugation by $\left(L_{+,1}^t \otimes L_{+,2}^t \otimes L_{+,3}^t\right)$ effectively exchanges the first and the second term of $\left(F_\xi^{CF}\right)_{11}$ [Eq. (E15)]; the same conclusion applies to $\left(F_\xi^{CF}\right)_{12}$ as well. It is also straightforward to check that $\left(F_\xi^{CF}\right)_{21}$ and $\left(F_\xi^{CF}\right)_{22}$ also commute with $\left(L_{+,1}^t \otimes L_{+,2}^t \otimes L_{+,3}^t\right)$. We conclude that $F_\xi^{CF}$ commutes with Eq. (E12)(3).

The other stabilizers on the domain wall—specifically (2), (4), (5), (6), (7), and (8) in Eq. (E12)—commute trivially with $F_\xi^{CF}$. Therefore, the $CF$ ribbon operator $F_\xi^{CF}$ [Eq. (E14)] commutes with the domain wall stabilizers Eq. (E12). By using the same method, one can check that $F_\xi^{FC}$ commutes with the domain wall stabilizers as well. We conclude that the Hamiltonian Eq. (E12) corresponds to the domain wall Hamiltonian representing the $C \leftrightarrow F$ domain wall.

## Appendix F: Domain walls in the 3d toric code: calculation details

In this section, we provide calculation details for the domain walls discussed in Section VI. In all the cases, we will start from a trivial SPT, apply the SPT entanglers, and then gauge the entire system. We discuss the type-I, II, and III domain walls in Section F 1, F 2, and F 3.

Let us remind the readers of our convention. We shall employ a lattice structure in Fig. 7. The vertices of this lattice can be partitioned into three sets $R$, $G$, and $B$. Vertices in these sets and the orientation of the edges are described as follows:

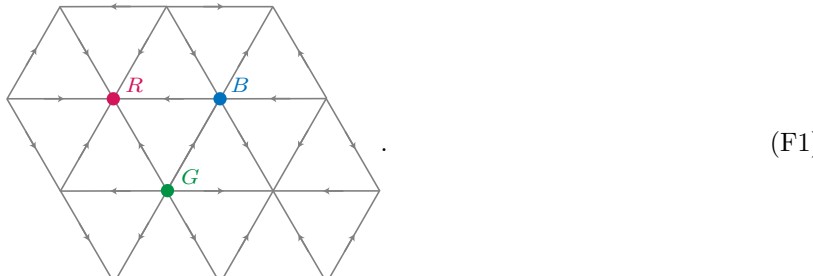

$$\text{(F1)}$$

With respect to this lattice, the SPT entanglers are constructed using the method of Ref. [61, 90], which amounts to assigning the following phase factor to each triangle:

$$\text{(F2)}$$

where $\omega$ is an appropriate cocycle.

### 1. Type-I domain wall

We begin by constructing a nontrivial $\mathbb{Z}_2$ SPT on a codimension-1 submanifold. The SPT on this submanifold is called the type-I SPT, which is equivalent to the Levin-Gu SPT [91]. The corresponding type-I 3-cocycle is given by [61, 72]

$$\omega_I(g_1, g_2, g_3) = \exp\left(\frac{p\pi i}{2} g_1(g_2 + g_3 - [g_2 + g_3])\right), \tag{F3}$$

where $g_1, g_2, g_3 \in \{0, 1\}$ correspond to the states at three vertices of a triangle, and $[g_2 + g_3] = g_2 + g_3$ mod 2. Equivalently, the nontrivial 3-cocycle can be written in the following form

$$\omega_I(g_1, g_2, g_3) = (-1)^{g_1 g_2 g_3}, \tag{F4}$$

which will be more convenient for us.

Letting $\omega = \omega_I$, we obtain the SPT entangler. For the purpose of obtaining the stabilizers of the SPT Hamiltonian, it suffices to obtain the entangler acting nontrivially on a single site, such as the following:

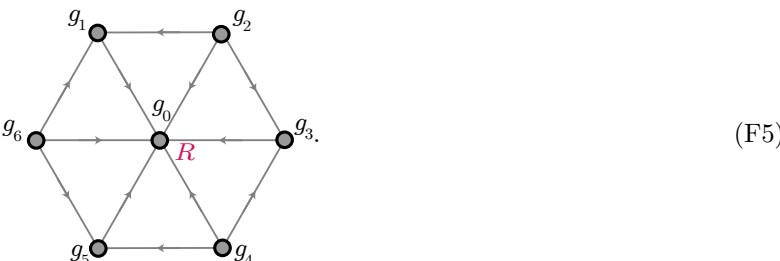

$$\text{(F5)}$$

With respect to these sites, we obtain the following SPT entangler (specifically, consisting only of gate acting nontrivially on $g_0$)

$$\frac{\omega_I(g_0, g_0^{-1}g_1, g_1^{-1}g_2)\omega_I(g_0, g_0^{-1}g_3, g_3^{-1}g_4)\omega_I(g_0, g_0^{-1}g_5, g_5^{-1}g_6)}{\omega_I(g_0, g_0^{-1}g_3, g_3^{-1}g_2)\omega_I(g_0, g_0^{-1}g_5, g_5^{-1}g_4)\omega_I(g_0, g_0^{-1}g_1, g_1^{-1}g_6)}$$
$$= \exp\left(\pi i g_0(g_1 g_2 - g_2 g_3 + g_3 g_4 - g_4 g_5 + g_5 g_6 - g_6 g_1)\right)$$
$$= \text{CCZ}_{0,1,2}\text{CCZ}_{0,2,3}\text{CCZ}_{0,3,4}\text{CCZ}_{0,4,5}\text{CCZ}_{0,5,6}\text{CCZ}_{0,6,1}, \tag{F6}$$

in which $\mathrm{CCZ}_{i,j,k}$ is the CCZ gate between qubits $i$, $j$, and $k$. The same result applies to every vertex, independent of their type $(R, G, \text{and } B)$. Thus the SPT entangler is a product of CCZ on every plaquette. Therefore, the fixed-point SPT wavefunction is given by the following,

$$|\psi\rangle = \sum_{\{g_v\}} \prod_{\triangle_{123}} \omega_I^{s(\triangle_{123})}(g_3, g_3^{-1}g_2, g_2^{-1}g_1) \bigotimes_v |g_v\rangle, \tag{F7}$$

in which $g_v$ is the state at site $v$, $\triangle_{123}$ is a 2-simplex (triangular face), and $s(\triangle_{123}) = \pm 1$ denotes the orientation of the triangle.

Conjugating the local stabilizers of the trivial SPT by the SPT entangler, we obtain the Hamiltonian of the SPT:

$$H = -\sum_v \quad \text{} \quad , \tag{F8}$$

After applying the gauging map [Section III], we obtain stabilizers of 3d toric code in the bulk. However, on the domain wall, we obtain the Hamiltonian of the type-I $\mathbb{Z}_2$ twisted quantum double, which is similar to the Hamiltonian of the quantum double model but includes an extra phase factor $W_v^g$ associated with each vertex [34, 106]. The Hamiltonian on the domain wall is given by

$$H = -\sum_v \sum_{g \in G} A_v^g W_v^g - \sum_p B_p. \tag{F9}$$

Here $A_v$ and $B_p$ are the vertex operator and the plaquette operator. (We use the convention in which raising by the power of 0 yields an identity operator.)

We now describe the remaining operator $W_v$. To that end, recall that the gauging maps the vertex degrees of freedom to the edge degrees of freedom [Section III]. Specifically, it gives the map:

$$\Gamma : |\{g_i\}\rangle_v \to |\{g_i g_j^{-1}\}\rangle_e. \tag{F10}$$

When $g_i \in \mathbb{Z}_n$, we have $g_i g_j^{-1} = g_i - g_j$. Thus we get $|g_{ij}\rangle = |g_i - g_j\rangle$ on the edges after gauging. Applying this relation to every CZ gate in Eq. (F8), we obtain

$$W_v^1 = \quad \text{} \quad , \tag{F11}$$

where the red wavy lines represent CZ gates between the connected two qubits.

While Eq. (F9) is not known in the literature in its exact form, we note that it can be related to a known exactly solvable model of double semions. More specifically, if we remove the edges perpendicular to the triangular lattice, its ground state becomes exactly the ground state of the model in Ref. [99]. To see this, note the following relation between a CZ gate and a product of $S$ and $S^\dagger$ gates on a flux-free triangle:

$$\text{} \tag{F12}$$

Using these relations, we can change the star operator as follows:

$$A_v^1 W_v^1 = \quad\longrightarrow\quad - \qquad\qquad. \tag{F13}$$

Here we conjugated each site on the triangular lattice by $S$ and $S^\dagger$ to remove Pauli $Z$ operators. At each edge connected to $v$, a factor of $i$ is introduced. The product of these factors gives rise to a minus sign. If we only consider the qubits on the triangular lattice, Eq. (F13) is exactly the generalized star term defined in Ref. [91, 99] as the stabilizer of the double semion model.

Similar to quantum doubles, ribbon operators can be defined for twisted quantum doubles as well. A detailed discussion can be found in Ref. [107]. Here, we present the relevant results.

An open ribbon $\xi$ can be defined on the triangular lattice with endpoints $(i_A, t_A)$ and $(i_B, t_B)$, as displayed below

$$\tag{F14}$$

A ribbon operator $F_\xi^{(h,g)}$ acting on the ribbon $\xi$ can be defined as follows:

$$\langle \{h'_{ij}\}|F_\xi^{(h \cdot g)}|\{h_{ij}\}\rangle = f_A \cdot f_B \cdot f_{AB} \cdot w_h^\xi(g). \tag{F15}$$

in which $|\{h_{ij}\}\rangle$ is the set of states inside the ribbon and $h'_{ij} = hh_{ij}$ when the group is Abelian. The right hand side of Eq. (F15) is a phase factor. The definitions of each components can be found in Ref. [107]. Here, instead of writing down the explicit form of the ribbon operator, we analyze the structure of the ribbon operator, and discuss the phenomenology of anyons.

The anyons of the TQD can be labeled by the magnetic fluxes, $A \in \mathbb{Z}_2$, and the associated projective representation corresponding to a 2-cocycle $c_A$, which is given by the slant product of the 3-cocycle.

$$c_A(B, C) := i_A \omega(B, C) = \frac{\omega_I(A, B, C)\omega_I(B, C, A)}{\omega_I(B, A, C)}. \tag{F16}$$

When $A = 0$, $c_A$ vanishes. Therefore, anyons without magnetic fluxes are classified by the irreducible representations of $\mathbb{Z}_2$, which corresponds to the the vacuum $1$ and the boson $s\bar{s}$. The ribbon operator creating a pair of $s\bar{s}$ can be represented as follows,

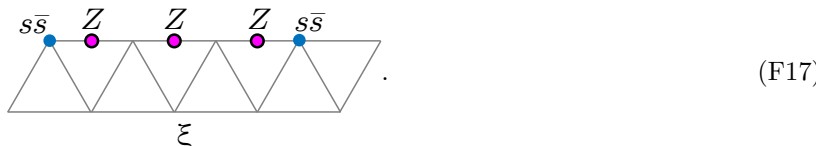

$$\tag{F17}$$

By applying the operator above, the star terms at the endpoints are violated, creating a pair of $s\bar{s}$. One can then apply Pauli $Z$ operators on the neighboring edges in the bulk to move the $s\bar{s}$ pairs into the bulk, where they become two electric charges.

When $A = 1$, $C_A = (-1)^{BC}$. The corresponding projective representations give rise to the semion $s$ and the anti-semion $\bar{s}$. The ribbon operators can be written in the following form:

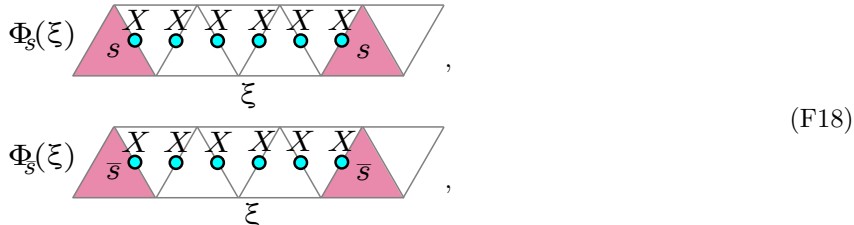

$$\text{(F18)}$$

in which $\Phi_s(\xi)$ and $\Phi_{\bar{s}}(\xi)$ are phase gates satisfying Eq. (F15). On the domain wall, a pair of $s$ or $\bar{s}$ anyons can be created by the above ribbon operator. At the same time, this operator also violates the connected plaquette stabilizers in the bulk, generating semi-loops of magnetic fluxes on both sides. Therefore, the excitation it creates is a magnetic loop anchored on the domain wall, with intersections attached to a pair of semions or anti-semions.

## 2. Type-II domain wall

In this subsection, we discuss the $2+1$D $\mathbb{Z}_2 \times \mathbb{Z}_2$ SPT-sewn domain wall. Restricted to the domain wall, the model is effectively a $\mathbb{Z}_2 \times \mathbb{Z}_2$ twisted quantum double (TQD), which is also equivalent to the $\mathbb{Z}_4$ quantum double (QD) [94]. We consider a lattice construction similar to that shown in Eq. (F1). In the bulks on both sides of the domain wall, there are 3d toric codes, and the domain wall itself is the $\mathbb{Z}_2 \times \mathbb{Z}_2$ SPT-sewn domain wall. Therefore, there are two qubits per site on the domain wall, each belonging to a different bulk on either side.

Let us first discuss the ungauged model. The nontrivial SPT that couples two $\mathbb{Z}_2$ symmetries together is called the type-II SPT [61, 72, 90]. The corresponding 3-cocycle can be written as,

$$\omega_{II}(g_1, g_2, g_3) = \exp\left(\frac{p\pi i}{2} g_1^{(1)}(g_2^{(2)} + g_3^{(2)} - \left[g_2^{(2)} + g_3^{(2)}\right])\right), \tag{F19}$$

in which $g_1$, $g_2$, and $g_3$ are $\mathbb{Z}_2 \times \mathbb{Z}_2$ states at each vertex of a triangle on the domain wall, and we have $g_1 = (g_1^{(1)}, g_1^{(2)})$, $g_2 = (g_2^{(1)}, g_2^{(2)})$, $g_3 = (g_3^{(1)}, g_3^{(2)})$, $p, g_1^{(i)}, g_2^{(j)}, g_3^{(k)} \in \{0, 1\}$, and $\left[g_2^{(2)} + g_3^{(2)}\right]$ represents the sum of $g_2^{(2)}$ and $g_3^{(2)}$ modulo 2. Equivalently, the nontrivial 3-cocycle can be written as the following form,

$$\omega_{II}(g_1, g_2, g_3) = (-1)^{g_1^{(1)} g_2^{(2)} g_3^{(2)}}. \tag{F20}$$

On the domain wall, the lattice is the same as Eq. (F1), but with two qubits per site. The branching structure near site $G$ is given as follows,

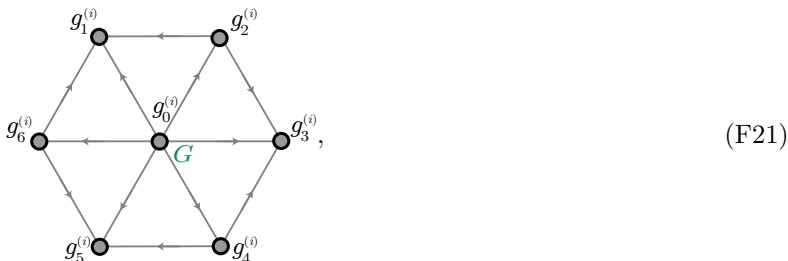

$$\text{(F21)}$$

in which $i \in \{1, 2\}$ labels different qubits at each site. The SPT entangler acting nontrivially on $g_0$ is

$$
\frac{\omega_{II}(g_1, g_1^{-1}g_2, g_2^{-1}g_0)\omega_{II}(g_3, g_3^{-1}g_4, g_4^{-1}g_0)\omega_{II}(g_5, g_5^{-1}g_6, g_6^{-1}g_0)}{\omega_{II}(g_3, g_3^{-1}g_2, g_2^{-1}g_0)\omega_{II}(g_5, g_5^{-1}g_4, g_4^{-1}g_0)\omega_{II}(g_1, g_1^{-1}g_6, g_6^{-1}g_0)}
$$
$$
= \exp\left(\pi i g_0^{(2)}(g_2^{(2)}g_1^{(1)} - g_2^{(2)}g_3^{(1)} + g_4^{(2)}g_3^{(1)} - g_4^{(2)}g_5^{(1)} + g_6^{(2)}g_5^{(1)} - g_6^{(2)}g_1^{(1)})\right)
$$
$$
\times \exp\left(\pi i \left(g_1^{(1)}g_1^{(2)}(g_2^{(2)} - g_6^{(2)}) + g_3^{(1)}g_3^{(2)}(g_4^{(2)} - g_2^{(2)}) + g_5^{(1)}g_5^{(2)}(g_6^{(2)} - g_4^{(2)})\right)\right)
$$
$$
\times \exp\left(\pi i \left(g_1^{(1)}g_6^{(2)} - g_1^{(1)}g_2^{(2)} + g_3^{(1)}g_2^{(2)} - g_3^{(1)}g_4^{(2)} + g_5^{(1)}g_4^{(2)} - g_5^{(1)}g_6^{(2)}\right)\right) \quad\text{(F22)}
$$
$$
= \text{CCZ}_{0^{(2)},1^{(1)},2^{(2)}} \text{CCZ}_{0^{(2)},2^{(2)},3^{(1)}} \text{CCZ}_{0^{(2)},4^{(2)},3^{(1)}} \text{CCZ}_{0^{(2)},4^{(2)},5^{(1)}} \text{CCZ}_{0^{(2)},6^{(2)},5^{(1)}} \text{CCZ}_{0^{(2)},6^{(2)},1^{(1)}}
$$
$$
\times \text{CCZ}_{1^{(1)},1^{(2)},2^{(2)}} \text{CCZ}_{1^{(1)},1^{(2)},6^{(2)}} \text{CCZ}_{3^{(1)},2^{(2)},3^{(2)}} \text{CCZ}_{3^{(1)},3^{(2)},4^{(2)}} \text{CCZ}_{5^{(1)},4^{(2)},5^{(2)}} \text{CCZ}_{5^{(1)},5^{(2)},6^{(2)}}
$$
$$
\times \text{CZ}_{1^{(1)},2^{(2)}} \text{CZ}_{1^{(1)},6^{(2)}} \text{CZ}_{3^{(1)},2^{(2)}} \text{CZ}_{3^{(1)},4^{(2)}} \text{CZ}_{5^{(1)},4^{(2)}} \text{CZ}_{5^{(1)},6^{(2)}}.
$$

Here $\text{CCZ}_{0^{(2)},1^{(1)},2^{(2)}}$ represents a CCZ gate acting on the second qubit at site 0, the first qubit at site 1, and the second qubit at site 2. Note that, when conjugating the single-qubit operator on a site of type $G$, only the first line in the last equation contributes.

Similarly, one can get the SPT entangler near site of type $R$, which is given by

$$
\text{CCZ}_{0^{(1)},1^{(2)},2^{(2)}} \text{CCZ}_{0^{(1)},2^{(2)},3^{(2)}} \text{CCZ}_{0^{(1)},4^{(2)},2^{(2)}} \text{CCZ}_{0^{(1)},4^{(2)},5^{(2)}} \text{CCZ}_{0^{(1)},6^{(2)},5^{(2)}} \text{CCZ}_{0^{(1)},6^{(2)},1^{(2)}}. \quad\text{(F23)}
$$

And we also have the SPT entangler near site of type $B$, which is

$$
\text{CCZ}_{0^{(2)},2^{(1)},1^{(2)}} \text{CCZ}_{0^{(2)},6^{(1)},1^{(2)}} \text{CCZ}_{0^{(2)},2^{(1)},3^{(2)}} \text{CCZ}_{0^{(2)},4^{(1)},3^{(2)}} \text{CCZ}_{0^{(2)},6^{(1)},5^{(2)}} \text{CCZ}_{0^{(2)},4^{(1)},5^{(2)}}
$$
$$
\times \text{CCZ}_{2^{(1)},2^{(2)},1^{(2)}} \text{CCZ}_{2^{(1)},2^{(2)},3^{(2)}} \text{CCZ}_{4^{(1)},4^{(2)},3^{(2)}} \text{CCZ}_{4^{(1)},4^{(2)},5^{(2)}} \text{CCZ}_{6^{(1)},6^{(2)},5^{(2)}} \text{CCZ}_{6^{(1)},6^{(2)},1^{(2)}}. \quad\text{(F24)}
$$

Note again only the first line contributes when conjugating an operator at the center site by the SPT entangler.

Combining all the contributions together, the Hamiltonian of the SPT is given by the following:

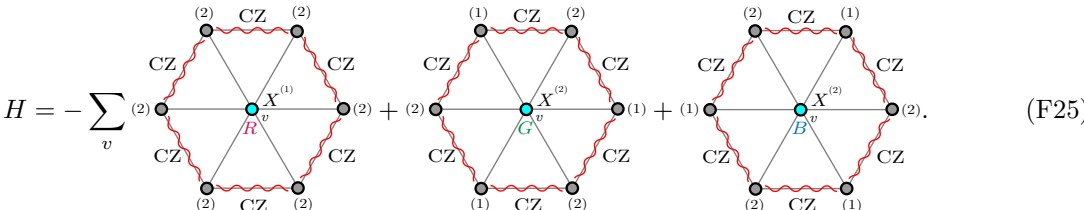

$$
H = -\sum_v \quad \text{(F25)}
$$

The number labels associated with each site represent which qubit the CZ gates apply on.

After applying the gauging map to Eq. (F25), we obtain 3d toric codes in the bulk on both sides of the domain wall. On the domain wall, we get a model which is effectively a $\mathbb{Z}_2 \times \mathbb{Z}_2$ twisted quantum double, but with extra edges connected to the bulk. The Hamiltonian can be written in the following form:

$$
H = -\sum_v \sum_{g \in G} A_v^g W_v^g - \sum_p B_p, \quad\text{(F26)}
$$

where $A_v^g$, $B_p$ and $W_v^g$ are shown in Fig. 12.

Before we discuss the properties of the domain wall in terms of bulk excitations, we first discuss the anyon theory of the $\mathbb{Z}_2 \times \mathbb{Z}_2$ twisted quantum double [72]. Anyons of $\mathbb{Z}_2 \times \mathbb{Z}_2$ TQD can be labelled as $(A, n^{(1)}n^{(2)})$, in which $A$ labels the magnetic fluxes and $n^{(1)}n^{(2)}$ labels the electric charges. For simplicity, and to draw an analogy to the $\mathbb{Z}_2$ toric code model, we will label the anyon $(A, n^{(1)}n^{(2)})$ as $m^{(a^{(1)},a^{(2)})}e^{(n^{(1)},n^{(2)})}$. For the full anyon content, see Table II. The exchange statistics are given by

$$
\theta(m^{(0,1)}e^{(0,1)}) = \theta(m^{(1,0)}e^{(1,0)}) = \theta(m^{(1,0)}e^{(1,1)}) = \theta(m^{(0,1)}e^{(1,1)}) = -1,
$$
$$
\theta(m^{(1,1)}) = \theta(m^{(1,1)}e^{(1,1)}) = i, \quad \theta(m^{(1,1)}e^{(0,1)}) = \theta(m^{(1,1)}e^{(1,0)}) = -i. \quad\text{(F27)}
$$

All other anyons are bosons.

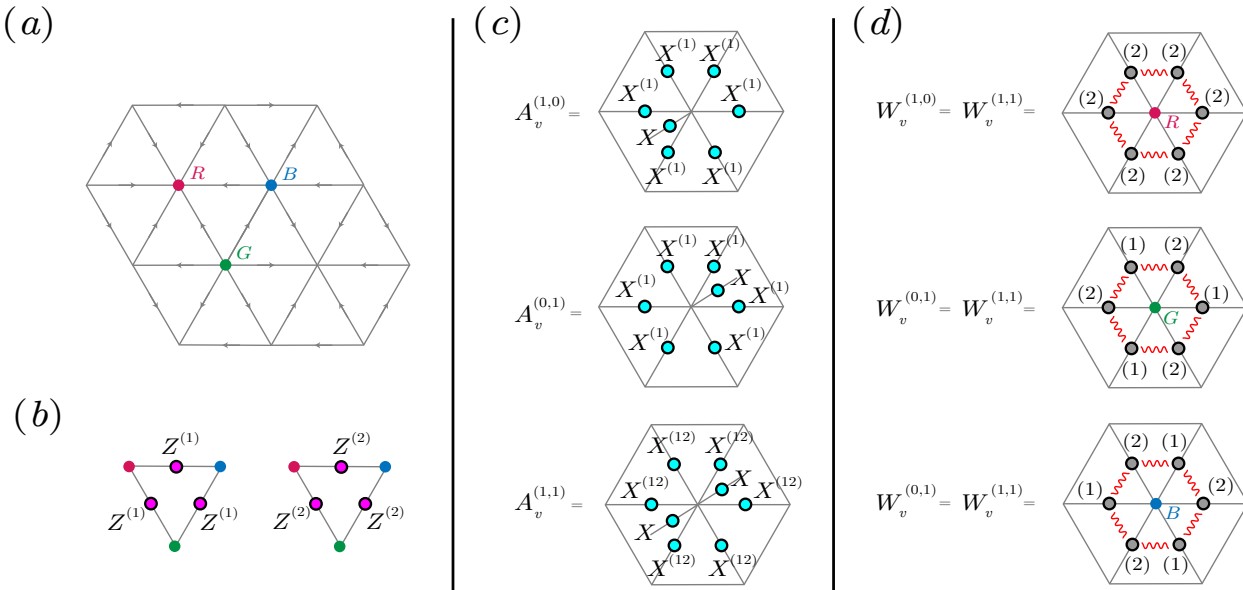

Figure 12. (a) Lattice used in the construction of type-II domain wall. We assign two qubits per site, each one being labeled by the bulk on different sides. (b) Plaquette $Z$ stabilizers $B_p$. (c) The star terms $A_v^g$. $X^{(1)}$ represents the Pauli $X$ operator acting on the first qubit, $X^{(2)}$ represents the Pauli $X$ operator acting on the second qubit, and $X^{(12)}$ represents a product of Pauli $X$ operators acting on both qubits. (d) The phase operator $W_v^g$. The red wavy lines represent CZ gates between qubits on different sites. The numbers label the qubits on each site. For example, a wavy line connects (1) and (2) represents a CZ gate between the first qubit on the first site and the second qubit on the second site. The operators not specified in these diagrams are identities.

The ribbon operators are given as follows. Pairs of $e^{(1,0)}$ and pairs of $e^{(0,1)}$ anyons can be created by applying Pauli $Z$ strings on the corresponding qubits.

$$\text{(F28)}$$

in which the subscripts represents which side of the 3d toric code the qubit belongs to. At the endpoints, the star terms are violated, creating excitations. By applying Pauli $Z$ operators on the neighboring edges in the bulk, one can move the anyons $e^{(1,0)}$ and $e^{(0,1)}$ into the bulk, where they become $e_L$ and $e_R$, respectively.

A Pair of $m^{(1,0)}$ and $m^{(1,0)}e^{(0,1)}$ can be created by

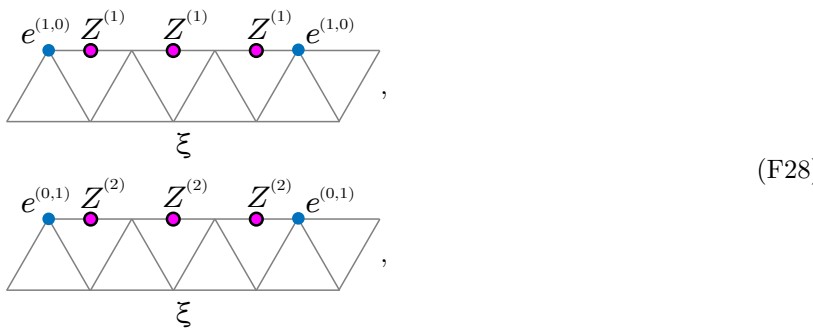

$$\text{(F29)}$$

And similarly, a pair of $m^{(0,1)}$ and $m^{(0,1)}e^{(1,0)}$ can be created by

$$\Phi_2(\xi) \quad \begin{array}{c} X^{(2)}X^{(2)}X^{(2)}X^{(2)}X^{(2)}X^{(2)} \\ \\ m^{(0,1)} \qquad \xi \qquad m^{(0,1)}e^{(1,0)} \end{array} \quad , \tag{F30}$$

in which $\Phi_1(\xi)$ and $\Phi_2(\xi)$ are phase gates satisfying Eq. (F15). On one side of the bulk, $e^{(1,0)}$ or $e^{(0,1)}$ can move out of the domain wall and become $e_L$ or $e_R$ in the bulk. On the other side, string of $X^{(i)}$ violates the plaquette stabilizers in the bulk, generating a semi-loop of magnetic flux with endpoints attached with a pair of $m^{(1,0)}$ or $m^{(0,1)}$.

With these ribbon operators, we can study the interplay between the excitations in the domain wall and the excitations of the 3d bulk. Consider a system with two $(3+1)$d toric codes and a $(2+1)$d $\mathbb{Z}_2 \times \mathbb{Z}_2$ TQD as a gapped domain wall, as depicted in Fig. 9(a). We label the excitations in the left as $\{1_L, e_L, m_L, m_L e_L\}$, and the excitations in the right as $\{1_R, e_R, m_R, m_R e_R\}$.

The endpoints of $e_L$ and $e_R$ lines can terminate on this domain wall and become bound states with $e^{(1,0)}$ and $e^{(0,1)}$, respectively. The endpoints of $m_L$ and $m_R$ can terminate on this domain wall as well, and become bound states with $m^{(1,0)}$ and $m^{(0,1)}$, respectively. This condensation property can enable a nontrivial domain wall. Namely, a pair of endpoints of magnetic fluxes become a single electric charge on the other side. This process is depicted in Fig. 9(a).

Also, a semi-circle of magnetic flux terminate on this domain wall and becomes a pair of $m^{(1,0)}$ or $m^{(0,1)}$ anyons and their anti-particles. For example, we can choose the anyon pair to be $m^{(1,0)}$ and $m^{(1,0)}e^{(0,1)}$. Since $e^{(0,1)}$ forms a bound state with $e_R$ from the other side, it can come out of the domain wall and become a $e_R$ particle in the bulk. Therefore, near the domain wall, there is one magnetic semi-loop attached to a pair of $m^{(1,0)}$ particles, and a string of $e_R$ particle.

### 3. Type-III domain wall

In this subsection, we discuss the $\mathbb{Z}_2 \times \mathbb{Z}_2 \times \mathbb{Z}_2$ SPT-sewn domain wall between two 3+1D toric codes and one 3+1D toric code. The procedure is briefly summarized as follows: Consider that we have three copies of the lattice displayed in Eq. (F1). For each copy, we assign one qubit at each vertex. We stack two copies on one side of the domain wall and one copy on the other side. At the domain wall, the lattice is similar to Eq. (F1), but with three qubits per site, each belonging to one of the copies. We initialize the state of each qubit to be the $|+\rangle$ state and apply the SPT entangler on the domain wall. After gauging the entire $\mathbb{Z}_2 \times \mathbb{Z}_2 \times \mathbb{Z}_2$ symmetry, we obtain the $\mathbb{Z}_2 \times \mathbb{Z}_2 \times \mathbb{Z}_2$ SPT-sewn domain wall.

Let us first discuss the ungauged model. The corresponding type-III 3-cocycle is given by

$$\omega_{III}(g_1, g_2, g_3) = \exp\left(p\pi i g_1^{(1)} g_2^{(2)} g_3^{(3)}\right), \tag{F31}$$

in which $g_1 = (g_1^{(1)}, g_1^{(2)}, g_1^{(3)})$, $g_2 = (g_2^{(1)}, g_2^{(2)}, g_2^{(3)})$, $g_3 = (g_3^{(1)}, g_3^{(2)}, g_3^{(3)})$, $p, g_1^{(i)}, g_2^{(j)}, g_3^{(k)} \in \{0,1\}$. The nontrivial 3-cocycle can be equivalently written as,

$$\omega_{III} = (-1)^{g_1^{(1)} g_2^{(2)} g_3^{(3)}}. \tag{F32}$$

Following calculations similar to Appendix F 1 and F 2, the SPT entangler near a site of type $R$ is given by

$$\mathrm{CCZ}_{0^{(1)}, 1^{(2)}, 2^{(3)}} \mathrm{CCZ}_{0^{(1)}, 3^{(2)}, 2^{(3)}} \mathrm{CCZ}_{0^{(1)}, 3^{(2)}, 4^{(3)}} \mathrm{CCZ}_{0^{(1)}, 5^{(2)}, 4^{(3)}} \mathrm{CCZ}_{0^{(1)}, 5^{(2)}, 6^{(3)}} \mathrm{CCZ}_{0^{(1)}, 1^{(2)}, 6^{(3)}}. \tag{F33}$$

Similarly, one can get the SPT entangler near site $G$ and site $B$. The Hamiltonian of the SPT on the domain

wall is given by

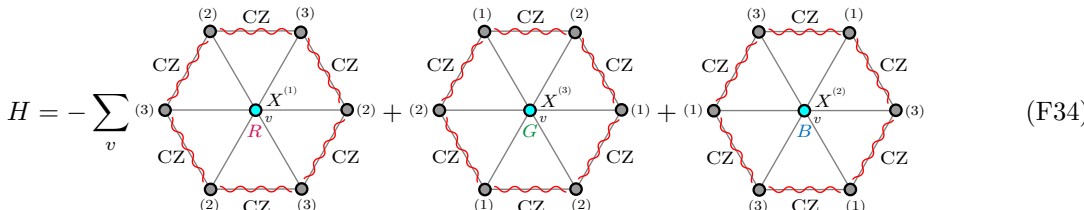

$$H = -\sum_v \quad (F34)$$

After applying the gauging map to Eq. (F10), we get the $\mathbb{Z}_2^3$ twisted quantum double. The Hamiltonian can be written in the follow form

$$H = -\sum_v \sum_{g \in G} A_v^g W_v^g - \sum_p B_p, \qquad (F35)$$

in which $g \in G = \mathbb{Z}_2^3$, $A_v$ is the star term, $B_p$ is the plaquette term, while the phase terms $W_v^g$ are displayed below,

$$W_v^{(1,b,c)} = \quad , W_v^{(a,b,1)} = \quad , W_v^{(a,1,c)} = \quad , \qquad (F36)$$

where $a, b, c \in \{0, 1\}$.

The excitations of the $\mathbb{Z}_2^3$ TQD can be labelled by the magnetic fluxes, $A \in \mathbb{Z}_2^3$, and the associated projective representation corresponding to a 2-cocycle $c_A$, which is given by the slant product of the 3-cocycle.

$$c_A(B, C) := i_A \omega(B, C) = \frac{\omega_{III}(A, B, C)\omega_{III}(B, C, A)}{\omega_{III}(B, A, C)}. \qquad (F37)$$

The anyons are summarized as follows [72]:

- $A = (0, 0, 0)$. There are 1 vacuum and 7 electric charges in this class. We label them as $1, e^{(100)}, e^{(010)}$, $e^{(001)}, e^{(110)}, e^{(011)}, e^{(101)}$, and $e^{(111)}$, in which $e^{(110)} = e^{(100)} \times e^{(010)}$.

- $A = (1, 0, 0)$. There are two two-dimensional irreducible representations for $c_A$, and the self-statistic for one is bosonic and the other is fermionic. Therefore, we label them as $m^{(100)}$ and $f^{(100)}$. Similarly, we have $m^{(010)}$ and $f^{(010)}$ when $A = (0, 1, 0)$, and $m^{(001)}$ and $f^{(001)}$ when $A = (0, 0, 1)$.

- $A = (1, 1, 0)$. Similar to the previous case, there are two two-dimensional irreducible representations for $c_A$, and the self-statistic for one is bosonic and the other is fermionic. We use the same nomenclature to label the anyons in this class as $m^{(110)}, m^{(011)}, m^{(101)}, f^{(110)}, f^{(011)}$, and $f^{(101)}$.

- $A = (1, 1, 1)$. In this case, there are two two-dimensional irreducible representations for $c_A$, and the self-statistics for one is semionic while for the other one is anti-semionic. Therefore, we label them as $s$ and $\bar{s}$.

In summary, there are 22 different types of anyons in $\mathbb{Z}_2^3$ TQD, including 1 vacuum, 7 electric charges, and 14 dyons.

Similar to the type-I, and type-II cases, electric charges can be created by strings of Pauli $Z$ operators, and magnetic fluxes can be created by strings of Pauli $X$ operators with a sequence of phase gates along the ribbon. When braiding two magnetic fluxes with different types, for example, $m^{(100)}$ and $m^{(010)}$, an eletric charge of the third type, $e^{(001)}$, can be generated at the intersection, which can be moved to the bulk and becomes a electric charge on the other side of the domain wall.