# Peer review of "Domain walls from SPT-sewing"

_SciPost Physics_

## Round 1 · Referee Report · Ryohei Kobayashi (Referee 1) · 2025-6-27

Report

This paper introduces a method termed SPT-sewing to construct gapped domain walls in topologically ordered systems by gauging lower-dimensional SPT phases. The method generalizes earlier constructions of domain walls by gauged SPT defects. The authors show that the SPT sewing can construct generic invertible defects in finite Abelian gauge theory in (2+1)D. The authors also find a non-invertible gapped domain walls in (3+1)D Z2 toric code which has not been discussed in literature. This paper contains a sufficiently new set of results, and I would accept after the authors address the comments listed below.

Before listing my comments, let me summarize what the SPT sewing is doing in G gauge theory. My understanding is as follows (correct me if I’m wrong):

The gapped domain walls of G gauge theories are understood as a gapped boundary of G x G gauge theory under folding, and then a generic gapped boundary condition is specified by a pair (K, omega), where K is a subgroup of G (unbroken gauge group at boundary), and omega is an element of H^2(K,U(1)). - The gauged SPT defect in a previous literature takes the unbroken gauge group K = diag(G x G) = G, with omega in H^2(G,U(1)). - Their G x G-SPT sewing in Section IVB corresponds to the Neumann (m condensed) boundary with K = G x G, with omega in H^2(G x G, U(1)). - Their G x G x G-SPT sewing in Section IVC would correspond to first taking a Neumann boundary condition for G x G gauge theory, and then adding a new dynamical G gauge field support at the boundary. One can then consider the gapped boundary labeled by H^2((G x G) x G, U(1)). By considering the mixed SPT action between (G x G) and new G at the boundary, one can enforce Dirichlet boundary condition for (G x G) by integrating out the boundary gauge field. The boundary condition therefore ends up with a partial Dirichlet boundary condition G x G -> K, depending on the choice of the mixed SPT action described above. The authors show that the invertible domain walls for finite gauge group G can be constructed in this manner.

I have a number of questions and comments:

  1. Given that the authors aim to construct generic gapped domain walls of finite gauge theory, I feel it would be easier to follow if the authors can mention generic correspondence to gapped boundary by folding labeled by (K in G x G, omega), and explain how their construction resides in this general framework. So far I can find some discussions in appendix, but it would be better to highlight this perspective to add clarity. This would clarify why we want to consider G x G SPT in Section IVB, or why we want to introduce additional G symmetry to study G x G x G SPT.

  2. The authors show that generic invertible domain walls can be obtained by G x G x G SPT sewing when G is finite Abelian, but I wonder if the authors can say something about non-invertible domain walls. I feel by considering generic mixed SPT action between (G x G) and G, it would realize generic symmetry breaking pattern G x G -> K, so I would expect one can get generic gapped domain walls which is not necessarily invertible. Can the authors comment or elaborate on that?

  3. The authors conjecture that G x RepG x G SPT sewing gives all invertible domain walls of G gauge theory, but it is hard to imagine how it is true. Is there a rationale or expectation why it is expected to be true for generic finite group G? For instance, is the (G x G) x RepG SPT expected to lead to generic symmetry breaking pattern G x G -> K, or only some special cases of them?

  4. I believe the anchoring domain wall for type II cocycle is non-invertible, since the magnetic surfaces can end at the domain wall. (I think it would be better to clarify it if it’s the case.) Can the authors comment on the fusion rule of this domain wall? I also believe that it would be nice to point out that a pair of magnetic defects fuse into an electric particle on this anchored domain wall, since the fusion rule is extended to Z4 at the domain wall.

  5. Similar comment applies for the type III domain wall; it would be interesting if the authors can comment on fusion rules or other algebraic structure of these defects.

  6. Let me point out missing citations. The paper omits references to prior works of gauged SPTs such as 2211.11764, 2311.06917. Both of which studies generic structure of the gauged SPT defects in Z2 gauge theories. Given the authors study a variant of gauged SPT defects, and also investigate the defects of (3+1)D Z2 toric code based on a similar construction, it would be appropriate and fair to cite these articles.

In summary, this is an interesting and well-executed paper with a lot of new results about domain walls and gauging procedures. I recommend acceptance after the authors address the points raised above.

Recommendation

Publish (easily meets expectations and criteria for this Journal; among top 50%)

---

## Round 1 · Referee Report · Anonymous (Referee 2) · 2025-7-2

Report

This paper introduces a method for creating defects by gauging. In particular, unlike the usual method of adding an SPT before gauging, they introduce a 1D chain coupled to one bulk phase on each side. By gauging this composite system, they realize defects such as the $e$-$m$ permutation symmetry in the toric code, which cannot be obtained by gauging an SPT defect in a bosonic system.

Generalizing this, they prove that for Abelian symmetry, invertible domain walls can be obtained by gauging some SPT with $G\times G\times G$ symmetry, and in the non-Abelian case, they conjecture this is true if the SPT has $G\times Rep(G)\times G$ symmetry.

Lastly, they discussed the construction in 3+1d, where they find domain walls which have a peculiar property. Namely, pushing a point charge through realizes a flux loop which is anchored to the domain wall on the other side. Thus the charge cannot really "pass" through this domain wall.

My general comments are that in the 1+1d setup, it seems that the SPT inserted in the middle can be blown up into a 2+1d patch (at least in the abelian case). Thus the SPT sowing construction seems to be related to gauging three patches of a trivial phase with G symmetry connected by two SPTs: one for each interface. I'm wondering if this picture can offer more insight into the types of domain walls that can be created (including the non-invertible ones.)

I believe this construction is interesting and provides new insights both abstractly and in terms of concrete lattice models. I will be happy to accept the paper after the authors address my concerns below.

Requested changes

  1. Figure on page 18: it seems the plus/minus signs in the subscript after gauging $L^g_-$ wasn't resolved properly. Also, it would be helpful to review the gauging of the non-abelian $Z$ operators in Section 3, since they were used in this figure.

  2. I am confused by the discussion at the bottom of page 18. Is the state given an SPT or the symmetry is broken to $(\mathbb Z_3 \times \mathbb Z_3) \rtimes \mathbb Z_2$? I think readers would also appreciate some more discussion about the property of this SPT. Is it somehow related to the $S_3 \times Rep(S_3)$ SPT discussed in Ref. 55?

  3. I regrettably cannot understand what the stabilizers in Figure 5 are. Many symbols don't seem to be defined in Eq. 35, such as

  4. $\tilde Z$ without the L,R subscript
  5. $Z_{a_l}$ and why it is in the c layer instead of the a layer -$Z_\pi$
  6. Why certain lattices have a branching structure while others don't. Also some sites have opposite branching structure? One should be able to invert the group elements to reverse the arrow on that edge to make the notation uniform across odd/even sites?
  7. What is the expression 2 Tr (...) acting on

  8. I think the claim that the anchoring domain wall generalizes the $e$-$m$ defect is slightly a red herring (precisely because of this anchoring property). This domain wall is non-invertible, which means that the $e$ excitation cannot pass through. It ends as an excitation on the domain wall. I also think the reverse process is crucial to discuss: what happens if one tries to push the $m$ loop into this domain wall.

  9. The anchoring domain wall is obtained from unfolding a gapped boundary of two toric codes which is labeled by a type-II cocycle of $\mathbb Z_2 \times \mathbb Z_2$. Thus, I also do not understand the comment that the authors claim it is beyond the Lagrangian subgroup formalism. Just as we understand all defects in 2+1d finite group gauge theories by constructing the Lagrangian algebra of their folded theories in terms of subgroups and 2-cocycles, the same can be said for this domain wall. In fact, it is a minimal boundary in the sense of arXiv:2502.20440

  10. I think Figure 10a is confusing/misleading. What does it mean to add the four $e$'s on the left? In 2+1d, the fusion $m \times m = 1+ e_1 + e_2 + e_1e_2 $ means that depending on the internal state of $m$, either of these fusion outcomes is possible. I think what is intended is that the $m$ loop anchored on the right boundary can allow a string of either one of the four charges to come out on the left, rather than all four of them "together".

  11. Related to the previous question, I'm actually wondering if the case of the type-III cocycle is really anchoring. In the type-II, I think it is anchoring because the only fusion outcome of the end point of m loops is a non-trivial anyon $e$. But for type-III, the identity channel exists. If I understand correctly, this seems to suggest that depending on the fusion outcome, the flux loop can be pulled off completely?

Recommendation

Ask for minor revision

---

## Round 1 · Referee Report · Anonymous (Referee 3) · 2025-7-4

Report

The manuscript provides construction of a family of invertible domain walls (also examples of non-invertible domain walls) in untwisted finite group gauge theory in 2+1d and 3+1d. The manuscript provides systematic discussion and explicit construction in Hamiltonian model, which can be useful.

I recommend the manuscript for publication after the following comments/questions are addressed (also the questions in the other referee report):

  • Can the author comment on whether the method can say anything about other categorical structures of the domain walls such as 10j symbol or other associator?

  • Can the author compare the construction with e.g. arxiv:2407.07964
    which also provide lattice construction for general gapped domain walls in finite group gauge theory in all dimensions? e.g. the construction of S_psi using SPT is also in (4.9) of 2407.07964

Recommendation

Ask for minor revision

---

## Editorial Decision

awaiting_resubmission